# ON THE IMPORTANCE OF SAMPLING IN TRAINING GCNS: CONVERGENCE ANALYSIS AND VARIANCE REDUCTION

## ABSTRACT

Graph Convolutional Networks (GCNs) have achieved impressive empirical advancement across a wide variety of graph-related applications. Despite their great success, training GCNs on large graphs suffers from computational and memory issues. A potential path to circumvent these obstacles is sampling-based methods, where at each layer a subset of nodes is sampled. Although recent studies have empirically demonstrated the effectiveness of sampling-based methods, these works lack theoretical convergence guarantees under realistic settings and cannot fully leverage the information of evolving parameters during optimization. In this paper, we describe and analyze a general ***doubly variance reduction*** schema that can accelerate any sampling method under the memory budget. The motivating impetus for the proposed schema is a careful analysis for the variance of sampling methods where it is shown that the induced variance can be decomposed into node embedding approximation variance (*zeroth-order variance*) during forward propagation and layerwise-gradient variance (*first-order variance*) during backward propagation. We theoretically analyze the convergence of the proposed schema and show that it enjoys an $\mathcal{O}(1/T)$ convergence rate. We complement our theoretical results by integrating the proposed schema in different sampling methods and applying them to different large real-world graphs.

## 1 INTRODUCTION

In the past few years, graph convolutional networks (GCNs) have achieved great success in many graph-related applications, such as semi-supervised node classification (Kipf & Welling, 2016), supervised graph classification (Xu et al., 2018), protein interface prediction (Fout et al., 2017), and knowledge graph (Schlichtkrull et al., 2018; Wang et al., 2017). However, most works on GCNs focus on relatively small graphs, and scaling GCNs for large-scale graphs is not straight forward. Due to the dependency of the nodes in the graph, we need to consider a large receptive-field to calculate the representation of each node in the mini-batch, while the receptive field grows exponentially with respect to the number of layers. To alleviate this issue, sampling-based methods, such as node-wise sampling (Hamilton et al., 2017; Ying et al., 2018; Chen et al., 2017), layer-wise sampling (Chen et al., 2018; Zou et al., 2019), and subgraph sampling (Chiang et al., 2019; Zeng et al., 2019) are proposed for mini-batch GCN training.

Although empirical results show that sampling-based methods can scale GCN training to large graphs, these methods suffer from a few key issues. First, the theoretical understanding of sampling-based methods is still lacking. Second, the aforementioned sampling strategies are only based on the structure of the graph. Although most recent works (Huang et al., 2018; Cong et al., 2020) propose to utilize adaptive importance sampling strategies to constantly re-evaluate the relative importance of nodes during training (e.g., current gradient or representation of nodes), finding the optimal adaptive sampling distribution is computationally inadmissible, as it requires to calculate the full gradient or node representations in each iteration. This necessitates developing alternative solutions that can efficiently be computed and that come with theoretical guarantees.

In this paper, we develop a novel variance reduction schema that can be applied to any sampling strategy to significantly reduce the induced variance. The key idea is to use the historical node

Figure 1: The effect of doubly variance reduction on training loss, validation loss, and mean-square error (MSE) of gradient on `Flickr` dataset using `LADIES` proposed in Zou et al. (2019).

embeddings and the historical layerwise gradient of each graph convolution layer as control variants. The main motivation behind the proposed schema stems from our theoretical analysis of the sampling methods' variance in training GCNs. Specifically, we show that due to the composite structure of training objective, any sampling strategy introduces two types of variance in estimating the stochastic gradients: node embedding approximation variance (*zeroth-order variance*) which results from embeddings approximation during forward propagation, and layerwise-gradient variance (*first-order variance*) which results from gradient estimation during backward propagation. In Figure 1, we exhibit the performance of proposed schema when utilized in the sampling strategy introduced in (Zou et al., 2019). The plots show that applying our proposal can lead to a significant reduction in variance; hence faster convergence rate and better test accuracy. We can also see that both zeroth-order and first-order methods are equally important and demonstrate significant improvement when applied jointly (i.e, *doubly variance reduction*).

**Contributions.** We summarize the contributions of this paper as follows:

- We provide the theoretical analysis for sampling-based GCN training (`SGCN`) with a non-asymptotic convergence rate. We show that due to the node embedding approximation variance, `SGCN`s suffer from *residual error* that hinders their convergence.
- We mathematically show that the aforementioned *residual error* can be resolved by employing zeroth-order variance reduction to node embedding approximation (dubbed as `SGCN+`), which explains why `VRGCN` (Chen et al., 2017) enjoys a better convergence than `GraphSAGE` (Hamilton et al., 2017), even with less sampled neighbors.
- We extend the algorithm from node embedding approximation to stochastic gradient approximation, and propose a generic and efficient doubly variance reduction schema (`SGCN++`). `SGCN++` can be integrated with different sampling-based methods to significantly reduce both zeroth- and first-order variance, and resulting in a faster convergence rate and better generalization.
- We theoretically analyze the convergence of `SGCN++` and obtain an $\mathcal{O}(1/T)$ rate, which significantly improves the best known bound $\mathcal{O}(1/\sqrt{T})$. We empirically verify `SGCN++` through various experiments on several real-world datasets and different sampling methods, where it demonstrates significant improvements over the original sampling methods.

## 2 RELATED WORKS

**Training GCNs via sampling.** The full-batch training of a typical GCN is employed in Kipf & Welling (2016) which necessities keeping the whole graph data and intermediate nodes' representations in the memory. This is the key bottleneck that hinders the scalability of full-batch GCN training. To overcome this issues, sampling-based GCN training methods (Hamilton et al., 2017; Chen et al., 2017; Chiang et al., 2019; Chen et al., 2018; Huang et al., 2018) are proposed to train GCNs based on mini-batch of nodes, and only aggregate the embeddings of a sampled subset of neighbors of nodes in the mini-batch. For example, `GraphSAGE` (Hamilton et al., 2017) restricts the computation complexity by uniformly sampling a fixed number of neighbors from the previous layer nodes. However, a significant computational overhead is introduced when GCN goes deep. `VRGCN` (Chen et al., 2017) further reduces the neighborhood size and uses history activation of the previous layer to reduce variance. However, they require to perform a full-batch graph convolutional operation on history activation during each forward propagation, which is computationally expensive. Another direction applies layerwise importance sampling to reduce variance. For example, `FastGCN` (Chen et al., 2018) independently sample a constant number of nodes in all layers using importance sampling. However, the sampled nodes are too sparse to achieve high accuracy.

LADIES (Zou et al., 2019) further restrict the candidate nodes in the union of the neighborhoods of the sampled nodes in the upper layer. However, significant overhead may be incurred due to the expensive sampling algorithm. In addition, subgraph sampling methods such as GraphSAINT (Zeng et al., 2019) constructs mini-batches by importance sampling, and apply normalization techniques to eliminate bias and reduce variance. However, the sampled subgraphs are usually sparse and requires a large sampling size to guarantee the performance.

**Theoretical analysis.** Despite many algorithmic progresses over the years, the theoretical understanding of the convergence for SGCNs training method is still limited. VRGCN provides a convergence analysis under a strong assumption that the stochastic gradient due to sampling is unbiased and achieved a convergence rate of $\mathcal{O}(1/\sqrt{T})$. However, the convergence analysis is limited to VRGCN, and the assumption is not true due to the composite structure of training objective as will be elaborated. Chen & Luss (2018) provides another convergence analysis for FastGCN under a strong assumption that the stochastic gradient of GCN converges to the consistent gradient exponentially fast with respect to the sample size, and results in the same convergence rate as unbiased ones, i.e., $\mathcal{O}(1/\sqrt{T})$. Most recently, Sato et al. (2020) provides PAC learning-style bounds on the node embedding and gradient estimation for SGCNs training. Another direction of theoretical research focuses on analyzing the expressive power of GCN (Garg et al., 2020; Chen et al., 2019; Zhang et al., 2020), which is not the focus of this paper and omitted for brevity.

**Connection to composite optimization.** The proposed *doubly variance reduction* algorithm shares the same spirit with the variance reduced composite optimization problem considered in Zhang & Xiao (2019a); Hu et al. (2020); Tran-Dinh et al. (2020); Zhang & Xiao (2019c;b), but with two main differences. Firstly, the objective function is different. In composite optimization, an objective function that only the first composite layer has the trainable parameters is considered, where the output of the lower-level function is acting as the parameter for the higher-level function. However, in GCN model, the output of one graph convolutional layer is the input node embedding matrix for the next layer. As a result, when analyzing the convergence of variance reduced neural network model, we have to explicitly handle both the evolving node embedding matrices and the trainable parameters at all layers. Secondly, the data points in these works are sampled independently, but the data points (nodes) in SGCN are sampled node- or layer-dependent according to the graph structure. In our analysis, we provide a sampled-graph structure dependent convergence rate by bridging the connection between the convergence rate of GCN to the graph Laplacian matrices.

## 3  SGCN: A TIGHIT ANALYSIS OF SGD FOR GCN TRAINING

**Full-batch GCN training.** We begin by introducing the basic mathematical formulation of training GCNs. In this paper, we consider training GCNs in semi-supervised multi-class classification setting. Given an undirected graph $\mathcal{G} = (\mathcal{V}, \mathcal{E})$ with $N = |\mathcal{V}|$ and $|\mathcal{E}|$ edges and the adjacency matrix $\mathbf{A} \in \{0,1\}^{N \times N}$, we assume that each node is associated with a feature vector $\boldsymbol{x}_i \in \mathbb{R}^d$ and label $y_i$. We use $\mathbf{X} = [\boldsymbol{x}_1, \ldots, \boldsymbol{x}_N] \in \mathbb{R}^{N \times d}$ and $\boldsymbol{y} = [y_1, \ldots, y_N] \in \mathbb{R}^N$ to denote the node feature matrix and label vector, respectively. The Laplacian matrix is calculated as $\mathbf{L} = \mathbf{D}^{-1/2} \mathbf{A} \mathbf{D}^{-1/2}$ or $\mathbf{L} = \mathbf{D}^{-1} \mathbf{A}$ where $\mathbf{D} \in \mathbb{R}^{N \times N}$ is the degree matrix. We use $\boldsymbol{\theta} = \{\mathbf{W}^{(1)}, \ldots, \mathbf{W}^{(L)}\}$ to denote the stacked weight parameters of a $L$-layer GCN. The training of full-batch GCN (FullGCN) as an empirical risk minimization problem aims at minimizing the loss $\mathcal{L}(\boldsymbol{\theta})$ over all training data

$$\mathcal{L}(\boldsymbol{\theta}) = \frac{1}{N} \sum_{i=1}^{N} \text{Loss}(\boldsymbol{h}_i^{(L)}, y_i), \ \mathbf{H}^{(L)} = \sigma\Big(\mathbf{L} \ldots \sigma\Big(\mathbf{L}\sigma(\underbrace{\mathbf{L}\mathbf{X}\mathbf{W}^{(1)}}_{\mathbf{Z}^{(1)}})\mathbf{W}^{(2)}\Big) \ldots \mathbf{W}^{(L)}\Big)$$

where $\boldsymbol{h}_i^{(\ell)}$ is the $i$th row of node embedding matrix $\mathbf{H}^{(\ell)} = \sigma(\mathbf{Z}^{(\ell)}), \mathbf{Z}^{(\ell)} = \mathbf{L}\mathbf{H}^{(\ell-1)}\mathbf{W}^{(\ell)}$ that corresponds to embedding of $i$th node at $\ell$th layer (hop), $\text{Loss}(\cdot, \cdot)$ is the loss function (e.g., cross-entropy loss) to measure the discrepancy between the prediction of the GCN and its ground truth label, and $\sigma(\cdot)$ is the activation function (e.g., ReLU function).

**Sampling-based GCN training.** When the graph is large, the computational complexity of forward and backward propagation could be very high. One practical solution to alleviate this issue is to sample a subset of nodes and construct a sparser normalized Laplacian matrix $\widetilde{\mathbf{L}}^{(\ell)}$ for each layer with $\text{supp}(\widetilde{\mathbf{L}}^{(\ell)}) \ll \text{supp}(\mathbf{L})$, and perform forward and backward propagation only based on the sampled Laplacian matrices. The sparse Laplacian matrix construction algorithms can be roughly

classified as *nodewise* sampling, *layerwise* sampling, and *subgraph* sampling. A detailed discussion of different sampling strategies can be found in Appendix D. To apply Stochastic Gradient Descent (SGD) (Bottou et al., 2018) to train GCN (SGCN), we sample a mini-batch of nodes $\mathcal{V}_\mathcal{B} \subseteq \mathcal{V}$ from all nodes with size $B = |\mathcal{V}_\mathcal{B}|$, and construct the set of sparser Laplacian matrices $\{\widetilde{\mathbf{L}}^{(\ell)}\}_{\ell=1}^L$ based on nodes sampled at each layer and compute the stochastic gradient to update parameters as [1]

$$\nabla\widetilde{\mathcal{L}}(\boldsymbol{\theta}) = \tfrac{1}{B}\sum_{i\in\mathcal{V}_\mathcal{B}}\nabla\text{Loss}(\widetilde{\boldsymbol{h}}_i^{(L)}, y_i),\ \widetilde{\mathbf{H}}^{(L)} = \sigma\Big(\widetilde{\mathbf{L}}^{(L)}\ldots\sigma\big(\widetilde{\mathbf{L}}^{(2)}\sigma(\underbrace{\widetilde{\mathbf{L}}^{(1)}\mathbf{X}\mathbf{W}^{(1)}}_{\widetilde{\mathbf{Z}}^{(1)}})\mathbf{W}^{(2)}\big)\ldots\mathbf{W}^{(L)}\Big)$$

**Key challenges.** Compared to vanilla SGD, the key challenge of theoretical understanding for SGCN training is the *biasedness* of stochastic gradient due to sampling of nodes at inner layers. Let denote FullGCN's full-batch gradient as $\nabla\mathcal{L}(\boldsymbol{\theta}) = \{\mathbf{G}^{(\ell)} = \frac{\partial\mathcal{L}(\boldsymbol{\theta})}{\partial\mathbf{W}^{(\ell)}}\}_{\ell=1}^L$ and SGCN's stochastic gradient as $\nabla\widetilde{\mathcal{L}}(\boldsymbol{\theta}) = \{\widetilde{\mathbf{G}}^{(\ell)} = \frac{\partial\widetilde{\mathcal{L}}(\boldsymbol{\theta})}{\partial\mathbf{W}^{(\ell)}}\}_{\ell=1}^L$. By the chain rule, we can compute the full-batch gradient $\mathbf{G}_t^{(\ell)}$ w.r.t. the $\ell$th layer weight matrix $\mathbf{W}^{(\ell)}$ as

$$\mathbf{G}_t^{(\ell)} = [\mathbf{L}\mathbf{H}_t^{(\ell-1)}]^\top\big(\mathbf{D}_t^{(\ell+1)}\circ\sigma'(\mathbf{Z}_t^{(\ell)})\big),\quad \mathbf{D}_t^{(\ell)} = \mathbf{L}^\top\big(\mathbf{D}_t^{(\ell+1)}\circ\sigma'(\mathbf{Z}_t^{(\ell)})\big)\mathbf{W}_t^{(\ell)},\quad \mathbf{D}_t^{(L+1)} = \frac{\partial\mathcal{L}(\boldsymbol{\theta}_t)}{\partial\mathbf{H}^{(L)}} \quad (1)$$

and compute stochastic gradient $\widetilde{\mathbf{G}}_t^{(\ell)}$ utilized in SGCN for the $\ell$th layer w.r.t. $\mathbf{W}^{(\ell)}$ as

$$\widetilde{\mathbf{G}}_t^{(\ell)} = [\widetilde{\mathbf{L}}^{(\ell)}\widetilde{\mathbf{H}}_t^{(\ell-1)}]^\top\big(\widetilde{\mathbf{D}}_t^{(\ell+1)}\circ\sigma'(\widetilde{\mathbf{Z}}_t^{(\ell)})\big),\quad \widetilde{\mathbf{D}}_t^{(\ell)} = [\widetilde{\mathbf{L}}^{(\ell)}]^\top\big(\widetilde{\mathbf{D}}_t^{(\ell+1)}\circ\sigma'(\widetilde{\mathbf{Z}}_t^{(\ell)})\big)\mathbf{W}_t^{(\ell)},\quad \widetilde{\mathbf{D}}_t^{(L+1)} = \frac{\partial\widetilde{\mathcal{L}}(\boldsymbol{\theta}_t)}{\partial\widetilde{\mathbf{H}}^{(L)}} \quad (2)$$

For any layer $\ell \in [L]$, the stochastic gradient $\widetilde{\mathbf{G}}_t^{(\ell)}$ is a biased estimator of full-batch gradient $\mathbf{G}_t^{(\ell)}$, as it is computed from $\mathbf{H}_t^{(L)}$ and $\mathbf{Z}_t^{(\ell)}$, which are not available in SGCN since $\widetilde{\mathbf{H}}_t^{(L)}$ and $\widetilde{\mathbf{Z}}_t^{(\ell)}$ are used as an approximation during training. Recently, Chen et al. (2017) established a convergence rate under the strong assumption that the stochastic gradient of SGCN is unbiased and Chen & Luss (2018) provided another analysis under the strong assumption that the stochastic gradient converges to the consistent gradient exponentially fast as the number of sampled nodes increases. While both studies establish the same convergence rate of $\mathcal{O}(1/\sqrt{T})$, however, these assumptions do not hold in reality due to the composite structure of the training objectives and sampling of nodes at inner layers. Motivated by this, we aim at providing a tight analysis without the aforementioned strong assumptions on the stochastic gradient. Our analysis is inspired by the bias and variance decomposition of the mean-square error of stochastic gradient, which has been previously used in Cong et al. (2020) to analysis the stochastic gradient in GCN. Formally, we can decompose mean-square error of stochastic gradient as

$$\mathbb{E}[\|\nabla\widetilde{\mathcal{L}}(\boldsymbol{\theta}) - \nabla\mathcal{L}(\boldsymbol{\theta})\|_\text{F}^2] = \underbrace{\mathbb{E}[\|\mathbb{E}[\nabla\widetilde{\mathcal{L}}(\boldsymbol{\theta})] - \nabla\mathcal{L}(\boldsymbol{\theta})\|_\text{F}^2]}_{\text{Bias }\mathbb{E}[\|\mathbf{b}\|_\text{F}^2]} + \underbrace{\mathbb{E}[\|\nabla\widetilde{\mathcal{L}}(\boldsymbol{\theta}) - \mathbb{E}[\nabla\widetilde{\mathcal{L}}(\boldsymbol{\theta})]\|_\text{F}^2]}_{\text{Variance }\mathbb{E}[\|\mathbf{n}\|_\text{F}^2]} \quad (3)$$

where the bias terms $\mathbb{E}[\|\mathbf{b}\|_\text{F}^2]$ is mainly due to the node embedding approximation variance (*zeroth-order variance*) during forward propagation and the variance term $\mathbb{E}[\|\mathbf{n}\|_\text{F}^2]$ is mainly due to the layerwise gradient variance (*first-order variance*) during backward propagation. Before proceeding to analysis, we make the following standard assumptions on the Lipschitz-continuity and smoothness of the loss function $\text{Loss}(\cdot, \cdot)$ and activation function $\sigma(\cdot)$.

**Assumption 1.** *The loss function $\text{Loss}(\cdot, \cdot)$ is $C_{loss}$-Lipschitz continuous and $L_{loss}$-smoothness w.r.t. to the input node embedding vector, i.e., $\|\text{Loss}(\boldsymbol{h}^{(L)}, y) - \text{Loss}(\boldsymbol{h'}^{(L)}, y)\|_2 \leq C_{loss}\|\boldsymbol{h}^{(L)} - \boldsymbol{h'}^{(L)}\|_2$ and $\|\nabla\text{Loss}(\boldsymbol{h}^{(L)}, y) - \nabla\text{Loss}(\boldsymbol{h'}^{(L)}, y)\|_2 \leq L_{loss}\|\boldsymbol{h}^{(L)} - \boldsymbol{h'}^{(L)}\|_2$.*

**Assumption 2.** *The activation function $\sigma(\cdot)$ is $C_\sigma$-Lipschitz continuous and $L_\sigma$-smoothness, i.e., $\|\sigma(\boldsymbol{z}^{(\ell)}) - \sigma(\boldsymbol{z'}^{(\ell)})\|_2 \leq C_\sigma\|\boldsymbol{z}^{(\ell)} - \boldsymbol{z'}^{(\ell)}\|_2$ and $\|\sigma'(\boldsymbol{z}^{(\ell)}) - \sigma'(\boldsymbol{z'}^{(\ell)})\|_2 \leq L_\sigma\|\boldsymbol{z}^{(\ell)} - \boldsymbol{z'}^{(\ell)}\|_2$.*

We also make the following assumptions on the norm of weight matrices, Laplacian matrices, and node feature matrix, which are used in the generalization analysis of GNNs Garg et al. (2020).

**Assumption 3.** *For any $\ell \in [L]$, the norm of weight matrices, Laplacian matrices, input node feature matrix are bounded: $\|\mathbf{W}^{(\ell)}\|_\text{F} \leq B_W$, $\|\widetilde{\mathbf{L}}^{(\ell)}\|_\text{F} \leq B_{LA}$, $\|\mathbf{L}\|_\text{F} \leq B_{LA}$, and $\|\mathbf{X}\|_\text{F} \leq B_H$.*

---

[1]We use a tilde symbol $\widetilde{\square}$ for their stochastic form

**Proposition 1.** *For any $\ell \in [L]$, there exist constants $B_H$ and $B_D$ such that the norm of node embedding matrices and the gradient with respect to the input node embedding matrices satisfy* $\|\mathbf{H}^{(\ell)}\|_{\mathrm{F}} \leq B_H$, $\|\widetilde{\mathbf{H}}^{(\ell)}\|_{\mathrm{F}} \leq B_H$, $\|\frac{\partial \sigma(\mathbf{L}\mathbf{H}^{(\ell-1)}\mathbf{W}^{(\ell)})}{\partial \mathbf{H}^{(\ell-1)}}\|_{\mathrm{F}} \leq B_D$, *and* $\|\frac{\partial \sigma(\widetilde{\mathbf{L}}^{(\ell)}\widetilde{\mathbf{H}}^{(\ell-1)}\mathbf{W}^{(\ell)})}{\partial \widetilde{\mathbf{H}}^{(\ell-1)}}\|_{\mathrm{F}} \leq B_D$.

Before presenting the convergence of SGCN, we introduce the notation of *propagation matrices* $\{\mathbf{P}^{(\ell)}\}_{\ell=1}^{L}$, which are defined as the column-wise expectation of the sparser Laplacian matrices. Note that this notation is only for presenting the theoretical results, and are not used in the practical training algorithms. By doing so, we can decompose the difference between $\widetilde{\mathbf{L}}^{(\ell)}$ and $\mathbf{L}$ as the summation of column-wise difference $\|\widetilde{\mathbf{L}}^{(\ell)} - \mathbf{P}^{(\ell)}\|_{\mathrm{F}}^2$ and row-wise difference $\|\mathbf{P}^{(\ell)} - \mathbf{L}\|_{\mathrm{F}}^2$.

In the following theorem, we show that the upper bound of the bias and variance of stochastic gradient is closely related to the expectation of column-wise difference $\mathbb{E}[\|\widetilde{\mathbf{L}}^{(\ell)} - \mathbf{P}^{(\ell)}\|_{\mathrm{F}}^2]$ and row-wise difference $\mathbb{E}[\|\mathbf{P}^{(\ell)} - \mathbf{L}\|_{\mathrm{F}}^2]$ which can significantly impact the convergence of SGCN.

**Theorem 1** (Convergence of SGCN). *Suppose Assumptions 1, 2, 3 hold and apply SGCN with learning rate chosen as $\eta = \min\{1/L_{\mathrm{F}}, 1/\sqrt{T}\}$ where $L_{\mathrm{F}}$ is the smoothness constant. Let $\Delta_{\mathbf{n}}$ and $\Delta_{\mathbf{b}}$ denote the upper bound on the variance and bias of stochastic gradients as:*

$$\Delta_{\mathbf{n}} = \sum_{\ell=1}^{L} \mathcal{O}(\mathbb{E}[\|\widetilde{\mathbf{L}}^{(\ell)} - \mathbf{P}^{(\ell)}\|_{\mathrm{F}}^2]) + \mathcal{O}(\mathbb{E}[\|\mathbf{P}^{(\ell)} - \mathbf{L}\|_{\mathrm{F}}^2]), \quad \Delta_{\mathbf{b}} = \sum_{\ell=1}^{L} \mathcal{O}(\mathbb{E}[\|\mathbf{P}^{(\ell)} - \mathbf{L}\|_{\mathrm{F}}^2]) \quad (4)$$

*Then, the output of SGCN satisfies*

$$\min_{t \in [T]} \mathbb{E}[\|\nabla \mathcal{L}(\boldsymbol{\theta}_t)\|_{\mathrm{F}}^2] \leq \frac{2(\mathcal{L}(\boldsymbol{\theta}_1) - \mathcal{L}(\boldsymbol{\theta}^\star))}{\sqrt{T}} + \frac{L_{\mathrm{F}}\Delta_{\mathbf{n}}}{\sqrt{T}} + \Delta_{\mathbf{b}}. \quad (5)$$

The exact value of key parameters $L_{\mathrm{F}}$, $\Delta_{\mathbf{n}}$, and $\Delta_{\mathbf{b}}$ are computed in Lemma 1, Lemma 2, and Lemma 3 respectively and can be found in Appendix G. Theorem 1 implies that after $T$ iterations the gradient norm of SGCN is at most $\mathcal{O}(\Delta_{\mathbf{n}}/\sqrt{T}) + \Delta_{\mathbf{b}}$, which suffers from a constant residual error $\Delta_{\mathbf{b}}$ that is not decreasing as the number of iterations $T$ increases. Without the bias[2] we recover the convergence of vanilla SGD. Of course, this type of convergence is only useful if $\Delta_{\mathbf{b}}$ and $\Delta_{\mathbf{n}}$ are small enough. We note that existing SGCN algorithms propose to reduce $\Delta_{\mathbf{b}}$ by increasing the number of neighbors sampled at each layer (e.g., GraphSAGE), or applying importance sampling (e.g., FastGCN, LADIES and GraphSAINT).

## 4 SGCN+: ZEROTH-ORDER VARIANCE REDUCTION

An important question to answer is: *can we eliminate the residual error without using all neighbors during forward-propagation?* A remarkable attempt to answer this question has been recently made in VRGCN (Chen et al., 2017) where they propose to use historical node embeddings as an approximation to estimate the true node embeddings. More specifically, the graph convolution in VRGCN is defined as $\widetilde{\mathbf{H}}_t^{(\ell)} = \sigma\big(\mathbf{L}\widetilde{\mathbf{H}}_{t-1}^{(\ell-1)}\mathbf{W}^{(\ell)} + \widetilde{\mathbf{L}}^{(\ell)}(\widetilde{\mathbf{H}}_t^{(\ell-1)} - \widetilde{\mathbf{H}}_{t-1}^{(\ell-1)})\mathbf{W}^{(\ell)}\big)$. Taking advantage of historical node embeddings, VRGCN requires less sampled neighbors and results in significant less computation overhead during gradient computation. Although VRGCN achieves significant speed up and better performance compared to other SGCNs, it involves using the full Laplacian matrix at each iteration, which can be computationally prohibitive. Moreover, since both SGCNs and VRGCN are approximating the exact node embeddings calculated using all neighbors, it is still not clear why VRGCN achieves a better convergence result than SGCNs using historical node embeddings.

To fill in these gaps, we introduce *zeroth-order variance reduced* sampling-based GCN training method dubbed as SGCN+. As shown in Algorithm 1, SGCN+ has two types of forward propagation: the forward propagation at the *snapshot steps* and the forward propagation at the *regular steps*. At the snapshot step ($t \mod K = 0$), a full Laplacian matrix is utilized:

$$\mathbf{Z}_t^{(\ell)} = \mathbf{L}\mathbf{H}_t^{(\ell-1)}\mathbf{W}_t^{(\ell)}, \quad \mathbf{H}_t^{(\ell)} = \sigma(\mathbf{Z}_t^{(\ell)}), \quad \widetilde{\mathbf{Z}}_t^{(\ell)} \leftarrow \mathbf{Z}_t^{(\ell)} \quad (6)$$

During the regular steps ($t \mod K \neq 0$), the sampled Laplacian matrix is utilized:

$$\widetilde{\mathbf{Z}}_t^{(\ell)} = \widetilde{\mathbf{Z}}_{t-1}^{(\ell)} + \widetilde{\mathbf{L}}^{(\ell)}\widetilde{\mathbf{H}}_t^{(\ell-1)}\mathbf{W}_t^{(\ell)} - \widetilde{\mathbf{L}}^{(\ell)}\widetilde{\mathbf{H}}_{t-1}^{(\ell-1)}\mathbf{W}_{t-1}^{(\ell)}, \quad \widetilde{\mathbf{H}}_t^{(\ell)} = \sigma(\widetilde{\mathbf{Z}}_t^{(\ell)}) \quad (7)$$

---

[2]We have $\Delta_{\mathbf{b}} = 0$ if all neighbor are used to calculate the exact node embeddings, i.e., $\mathbf{P}^{(\ell)} = \mathbf{L}$, $\forall \ell \in [L]$.

---

**Algorithm 1** SGCN+: Zeroth-order variance reduction (Detailed version in Algorithm 4)

---
1: **Input:** Learning rate $\eta > 0$, snapshot gap $K > 0$
2: **for** $t = 1, \ldots, T$ **do**
3:     **if** $t \bmod K = 0$ **then**
4:         Calculate node embeddings using Eq. 6
5:         Calculate full-batch gradient $\nabla\mathcal{L}(\boldsymbol{\theta}_t)$ as Eq. 1 and update as $\boldsymbol{\theta}_{t+1} = \boldsymbol{\theta}_t - \eta\nabla\mathcal{L}(\boldsymbol{\theta}_t)$
6:     **else**
7:         Calculate node embeddings using Eq. 7
8:         Calculate stochastic gradient $\nabla\widetilde{\mathcal{L}}(\boldsymbol{\theta}_t)$ as Eq. 2 and update as $\boldsymbol{\theta}_{t+1} = \boldsymbol{\theta}_t - \eta\nabla\widetilde{\mathcal{L}}(\boldsymbol{\theta}_t)$
9:     **end if**
10: **end for**
11: **Output:** Model with parameter $\boldsymbol{\theta}_{T+1}$

---

**Algorithm 2** SGCN++: Doubly variance reduction (Detailed version in Algorithm 5)

---
1: **Input:** Learning rate $\eta > 0$, snapshot gap $K > 0$
2: **for** $t = 1, \ldots, T$ **do**
3:     **if** $t \bmod K = 0$ **then**
4:         Calculate node embeddings using Eq. 6
5:         Calculate full-batch gradient $\nabla\mathcal{L}(\boldsymbol{\theta}_t)$ usng Eq. 1 and update as $\boldsymbol{\theta}_{t+1} = \boldsymbol{\theta}_t - \eta\nabla\mathcal{L}(\boldsymbol{\theta}_t)$
6:         Save the per layerwise gradient $\widetilde{\mathbf{G}}_t^{(\ell)} \leftarrow \mathbf{G}_t^{(\ell)}$, $\widetilde{\mathbf{D}}_t^{(\ell)} \leftarrow \mathbf{D}_t^{(\ell)}$, $\forall \ell \in [L]$
7:     **else**
8:         Calculate node embeddings using Eq. 7
9:         Calculate stochastic gradient $\nabla\widetilde{\mathcal{L}}(\boldsymbol{\theta}_t)$ usng Eq. 10 and update as $\boldsymbol{\theta}_{t+1} = \boldsymbol{\theta}_t - \eta\nabla\widetilde{\mathcal{L}}(\boldsymbol{\theta}_t)$
10:     **end if**
11: **end for**
12: **Output:** Model with parameter $\boldsymbol{\theta}_{T+1}$

---

Comparing with VRGCN, the proposed SGCN+ only requires one full Laplacian graph convolution operation every $K$ iterations, where $K > 0$ is an additional parameter to be tuned.

In the following theorem, we introduce the convergence result of SGCN+. Recall that the node embedding approximation variance (*zeroth-order variance*) determines the bias of stochastic gradient $\mathbb{E}[\|\mathbf{b}\|_{\mathrm{F}}^2]$. Applying SGCN+ can significantly reduce the bias of stochastic gradient, such that its value is small enough that will not deteriorate the convergence.

**Theorem 2** (Convergence of SGCN+). *Suppose Assumptions 1, 2, 3 hold and apply SGCN+ with learning rate chosen as $\eta = \min\{1/L_{\mathrm{F}}, 1/\sqrt{T}\}$ where $L_{\mathrm{F}}$ is the smoothness constant. Let $\Delta_{\mathbf{n}}$ and $\Delta_{\mathbf{b}}^+$ denote the upper bound for the variance and bias of stochastic gradient as:*

$$
\begin{aligned}
\Delta_{\mathbf{n}} &= \sum_{\ell=1}^{L} \mathcal{O}(\mathbb{E}[\|\widetilde{\mathbf{L}}^{(\ell)} - \mathbf{P}^{(\ell)}\|_{\mathrm{F}}^2]) + \mathcal{O}(\mathbb{E}[\|\mathbf{P}^{(\ell)} - \mathbf{L}\|_{\mathrm{F}}^2]), \\
\Delta_{\mathbf{b}}^+ &= \eta^2 \Delta_{\mathbf{b}}^{+\prime} \ \text{where} \ \Delta_{\mathbf{b}}^{+\prime} = \mathcal{O}\Big( K \sum_{\ell=1}^{L} |\mathbb{E}[\|\mathbf{P}^{(\ell)}\|_{\mathrm{F}}^2] - \|\mathbf{L}\|_{\mathrm{F}}^2| \Big)
\end{aligned}
\tag{8}
$$

*Then, the output of SGCN+ satisfies*

$$
\min_{t \in [T]} \mathbb{E}[\|\nabla\mathcal{L}(\boldsymbol{\theta}_t)\|_{\mathrm{F}}^2] \leq \frac{2(\mathcal{L}(\boldsymbol{\theta}_1) - \mathcal{L}(\boldsymbol{\theta}^\star))}{\sqrt{T}} + \frac{L_{\mathrm{F}}\Delta_{\mathbf{n}}}{\sqrt{T}} + \frac{\Delta_{\mathbf{b}}^{+\prime}}{T}.
\tag{9}
$$

The exact value of key parameters $L_{\mathrm{F}}$, $\Delta_{\mathbf{n}}$, and $\Delta_{\mathbf{b}}^+$ are computed in Lemma 1, Lemma 2, and Lemma 5 respectively, and can be found in Appendix G, H. Theorem 2 implies that after $T$ iterations the gradient norm of SGCN+ is at most $\mathcal{O}(\Delta_{\mathbf{n}}/\sqrt{T}) + \mathcal{O}(\Delta_{\mathbf{b}}^{+\prime}/T)$. When using all neighbors for calculating the exact node embeddings, we have $\mathbf{P}^{(\ell)} = \mathbf{L}$ such that $\Delta_{\mathbf{b}}^{+\prime} = 0$, which leads to convergence rate of SGD. Comparing with vallina SGCN, the bias of SGCN+ is scaled by learning rate $\eta$. Therefore, we can reduce the negative effect of bias by choose learning rate as $\eta = \mathcal{O}(1/\sqrt{T})$. This also explains why SGCN+ achieves a significantly better convergence rate compared to SGCN.

## 5 SGCN++: DOUBLY VARIANCE REDUCTION

Algorithm 1 applies zeroth-order variance reduction on node embedding matrices and results in a faster convergence. However, both SGCN and SGCN+ suffer from the same stochastic gradient variance $\Delta_{\mathbf{n}}$, which can be only reduced either by increasing the mini-batch size of SGCN or applying variance reduction on stochastic gradient. An interesting question that arises is: *can we further accelerate the convergence by simultaneously employing zeroth-order variance reduction on node embeddings and first-order variance reduction on layerwise gradient?* To answer this question, we propose *doubly variance reduction* algorithm SGCN++, that extends the variance reduction algorithm from node embedding approximation to layerwise gradient estimation.

As shown in Algorithm 2, the main idea of SGCN++ is to use the historical gradient as control variants for current layerwise gradient estimation. More specifically, similar to SGCN+ that has two types of forward propagation steps, SGCN++ also has two types of backward propagation: at the *snapshot steps* and at the *regular steps*. The snapshot steps ($t \mod K = 0$) backward propagation are full-batch gradient computation as is defined in Eq. 1, and the computed full-batch gradient are saved as control variants for the following regular steps. The backward propagation ($t \mod K \neq 0$) at the regular steps are defined as

$$\widetilde{\mathbf{G}}_t^{(\ell)} = \widetilde{\mathbf{G}}_{t-1}^{(\ell)} + [\widetilde{\mathbf{L}}^{(\ell)}\widetilde{\mathbf{H}}_t^{(\ell-1)}]^\top \left( \widetilde{\mathbf{D}}_t^{(\ell+1)} \circ \sigma'(\widetilde{\mathbf{Z}}_t) \right) - [\widetilde{\mathbf{L}}^{(\ell)}\widetilde{\mathbf{H}}_{t-1}^{(\ell-1)}]^\top \left( \widetilde{\mathbf{D}}_{t-1}^{(\ell+1)} \circ \sigma'(\widetilde{\mathbf{Z}}_{t-1}) \right)$$
$$\widetilde{\mathbf{D}}_t^{(\ell)} = \widetilde{\mathbf{D}}_{t-1}^{(\ell)} + [\widetilde{\mathbf{L}}^{(\ell)}]^\top \left( \widetilde{\mathbf{D}}_t^{(\ell+1)} \circ \sigma'(\widetilde{\mathbf{Z}}_t) \right) [\mathbf{W}_t^{(\ell)}] - [\widetilde{\mathbf{L}}^{(\ell)}]^\top \left( \widetilde{\mathbf{D}}_{t-1}^{(\ell+1)} \circ \sigma'(\widetilde{\mathbf{Z}}_{t-1}) \right) [\mathbf{W}_{t-1}^{(\ell)}]$$

(10)

Next, in the following theorem, we establish the convergence rate of SGCN++. Recall that the mean-square error of the stochastic gradient can be decomposed into bias $\mathbb{E}[\|\mathbf{b}\|_{\mathrm{F}}^2]$ that is due to node embedding approximation and variance $\mathbb{E}[\|\mathbf{n}\|_{\mathrm{F}}^2]$ that is due to layerwise gradient estimation. Applying doubly variance reduction on node embedding and layerwise gradient simultaneously can significantly reduce mean-square error of stochastic gradient and speed up convergence.

**Theorem 3** (Convergence of SGCN++). *Suppose Assumptions 1, 2, 3 hold, and denote $L_{\mathrm{F}}$ as the smoothness constant and $\Delta_{\mathbf{n+b}}^{++}$ as the upper-bound of mean-square error of stochastic gradient*

$$\Delta_{\mathbf{n+b}}^{++} = \eta^2 \Delta_{\mathbf{n+b}}^{++\prime} = \eta^2 \mathcal{O}\left( K \sum_{\ell=1}^{L} |\mathbb{E}[\|\widetilde{\mathbf{L}}^{(\ell)}\|_{\mathrm{F}}^2] - \|\mathbf{L}\|_{\mathrm{F}}^2| \right)$$

(11)

*Apply SGCN++ in Algorithm 2 with learning rate as $\eta = \frac{2}{L_{\mathrm{F}} + \sqrt{L_{\mathrm{F}}^2 + 4\Delta_{\mathbf{n+b}}^{++\prime}}}$. Then it holds that*

$$\frac{1}{T} \sum_{t=1}^{T} \mathbb{E}[\|\nabla\mathcal{L}(\boldsymbol{\theta}_t)\|^2] \leq \frac{1}{T}\left( L_{\mathrm{F}} + \sqrt{L_{\mathrm{F}}^2 + 4\Delta_{\mathbf{n+b}}^{++\prime}} \right)\left( \mathcal{L}(\boldsymbol{\theta}_1) - \mathcal{L}(\boldsymbol{\theta}^\star) \right).$$

(12)

The exact value of key parameter $L_{\mathrm{F}}$ and $\Delta_{\mathbf{n+b}}^{++\prime}$ are computed in Lemma 1 and Lemma 12 respectively, and can be found in Appendix G, I. Theorem 2 implies that applying doubly variance reduction can scale the mean-square error $\mathcal{O}(\eta^2 K)$ times smaller. As a result, after $T$ iterations the norm of gradient of solution obtained by SGCN++ is at most $\mathcal{O}(\Delta_{\mathbf{n+b}}^{++\prime}/T)$, which enjoys the same rate as vanilla variance reduced SGD (Reddi et al., 2016; Fang et al., 2018).

**Scalability of SGCN++.** One might doubt *whether the computation at the snapshot step with full-batch gradient will hinder the scalability of SGCN++ for extremely large graphs*? Heuristically, we can approximate the full-batch gradient with the gradient calculated on a large-batch using all neighbors. The intuition stems from matrix Bernstein inequality (Gross, 2011), where the probability of the approximation error violating the desired accuracy decreases exponentially as the number of samples increase. Please refer to Algorithm 6 for the full-batch free SGCN++ and explanation on why large-batch approximation is feasible using tools from matrix concentration. Moreover, we provide the empirical evaluation on the large-batch size instead of full-batch in Appendix C. We remark that large-batch approximation can also be utilized in SGCN+ to further reduce the memory requirement for historical node embeddings.

**Connection to composite optimization.** Although we formulate sampling-based GCNs as a special case of the composite optimization problem, it is worth noting that compared to the classical composite optimization, there are a few key differences that make the utilization of variance reduction methods for composite optimization non-trivial: (a) different objective function that makes the GCN analysis challenging; (b) different gradient computation, analysis, and algorithm which make

Table 1: Comparison of the accuracy (F1-score) of SGCN, SGCN+, and SGCN++.

| METHOD / VR | | PPI | PPI-LARGE | FLICKR | REDDIT | YELP |
|---|---|---|---|---|---|---|
| EXACT | SGCN | 77.61 | 78.10 | 52.30 | 95.07 | 59.99 |
| | SGCN++ (DOUBLY) | **82.18** | **88.98** | **52.88** | **95.17** | **62.09** |
| VRGCN | SGCN+ (ZEROTH) | 77.65 | 77.82 | 52.57 | 95.17 | 61.29 |
| | SGCN++ (DOUBLY) | **82.50** | **88.65** | 52.53 | **95.17** | **62.64** |
| GRAPHSAGE | SGCN | 71.40 | 69.51 | 51.13 | 94.73 | 59.58 |
| | SGCN+ (ZEROTH) | 72.16 | 69.67 | 51.13 | 94.89 | 58.62 |
| | SGCN++ (DOUBLY) | **79.63** | **85.41** | **52.65** | **95.18** | **61.75** |
| FASTGCN | SGCN | 63.51 | 59.60 | 50.74 | 87.36 | 55.75 |
| | SGCN+ (ZEROTH) | 72.32 | 72.30 | 51.07 | 94.54 | 56.86 |
| | SGCN++ (DOUBLY) | **81.92** | **85.04** | **52.57** | **94.99** | **60.63** |
| LADIES | SGCN | 62.46 | 60.74 | 50.29 | 94.11 | 59.84 |
| | SGCN+ (ZEROTH) | 72.16 | 74.03 | 51.83 | 94.39 | 57.06 |
| | SGCN++ (DOUBLY) | **81.58** | **84.18** | **52.09** | **95.05** | **60.96** |
| GRAPHSAINT | SGCN | 61.51 | 38.68 | 50.10 | 93.68 | 54.65 |
| | SGCN+ (ZEROTH) | 66.94 | 41.40 | 50.66 | 84.61 | 55.42 |
| | SGCN++ (DOUBLY) | **79.63** | **79.71** | **50.94** | **94.18** | **57.03** |
| FULLGCN | N/A | 82.14 | 90.62 | 52.99 | 95.15 | 62.77 |

directly applying multi-level variance reduction methods such as SPIDER (Zhang & Xiao, 2019b) nontrivial; (c) different theoretical results and novel intuition for sampling-based GCN training. Due to the space limit, we defer the detail discussion to the Appendix A.

## 6 EXPERIMENTS

**Experimental setup.** We evaluate our proposed methods in semi-supervised learning setting on various classification datasets, summarized in Appendix B. In addition to different sampling mechanisms, we introduce Exact sampling that takes all neighbors for node embedding computation during mini-batch training (no zeroth-order variance), which can be used to explicitly test the importance of first-order variance reduction. We add SGCN+(Zeroth) and SGCN++(Doubly) on top of each sampling method to illustrate how zeroth-order and doubly variance reduction affect GCN training. All implementation details are deferred to Appendix B. By default, we train 2-layer GCNs with hidden dimension of 256 and snapshot gap $K = 10$. We use all nodes for snapshot computation on Flickr, PPI, PPI-large datasets. To scale variance reduced algorithm on large-graph, we employ snapshot large-batch approximation for both SGCN+ and SGCN++ by randomly selecting 50% of nodes for Reddit and 15% of nodes for Yelp dataset. We update the model with a mini-batch size of $B = 512$ and Adam optimizer with a learning rate of $\eta = 0.01$. We conduct training 3 times for 200 epochs and report the average results. We choose the model with the lowest validation error as the convergence point. A summary of experiment configurations and data statistic can be found in Table 2 and Table 3 in Appendix B. Due to the space limit, more experiments can be found in Appendix C.

**Experiment results.** In Table 1 and Figure 2, we show the accuracy and convergence comparison of SGCN, SGCN+, and SGCN++. We remark that multi-class classification tasks prefer a more stable node embedding and gradient than single-class classification tasks. Therefore, even the vanilla Exact, GraphSAGE and VRGCN already outperforms other baseline methods on PPI, PPI-large, and Yelp. Applying variance reductions can further improve its performance. In addition, we observe that the effect of variance reduction depends on its base sampling algorithms. Even though the performance of base sampling algorithm various significantly, the doubly variance reduction can bring their performance to a similar level. Moreover, we can observe from the loss curves that SGCNs suffers an residual error as discussed in Theorem 1, and the residual error is proportional to node embedding approximation variance (zeroth-order variance), where VRGCN has less variance than GraphSAGE because of its zeroth-order variance reduction, and GraphSAGE has less variance than LADIES because more nodes are sampled for node embedding approximation.

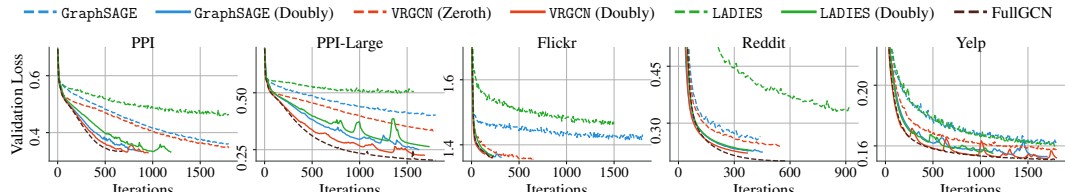

Figure 2: Comparing the validation loss of SGCN and SGCN++ on real world datasets.

## 7 CONCLUSION

In this work, we develop a theoretical framework for analyzing the convergence of sampling based mini-batch GCNs training. We show that the node embedding approximation variance and layerwise gradient variance are two key factors that slow down the convergence of these methods. Furthermore, we propose doubly variance reduction schema and theoretically analyzed its convergence. Experimental results on benchmark datasets demonstrate the effectiveness of proposed schema to significantly reduce the variance of different sampling strategies to achieve better generalization.

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

# Appendix

## Table of Contents

# A CONNECTION TO COMPOSITE OPTIMIZATION

In this section, we formally compare the optimization problem in training GCNs to the standard composite optimization and highlight the key differences that necessitates developing a completely different variance reduction schema and convergence analysis compared to the composite optimization counterparts (e.g., see Fang et al. (2018)).

**Different objective function.** In composite optimization, the output of the lower-level function is treated as the *parameter* of the outer-level function. However in GCN, the output of the lower-level function is used as the *input* of the outer-level function, and the parameter of the outer-level function is independent of the output of the inner-layer result.

More specifically, a two-level composite optimization problem can be formulated as

$$F(\boldsymbol{\theta}) = \frac{1}{N}\sum_{i=1}^{N} f_i\Big(\frac{1}{M}\sum_{j=1}^{M} g_j(\boldsymbol{w})\Big),\ \boldsymbol{\theta} = \{\boldsymbol{w}\}, \tag{13}$$

where $f_i(\cdot)$ is the outer-level function computed on the $i$th data point, $g_j(\cdot)$ is the inner-level function computed on the $j$th data point, and $\boldsymbol{w}$ is the parameter. We denote $\nabla f_i(\cdot)$ and $\nabla g_j(\cdot)$ as the gradient. Then, the gradient for Eq. 13 is computed as

$$\nabla F(\boldsymbol{\theta}) = \Big[\frac{1}{N}\sum_{i=1}^{N} \nabla f_i\Big(\frac{1}{M}\sum_{j=1}^{M} g_j(\boldsymbol{w})\Big)\Big]\Big(\frac{1}{M}\sum_{j=1}^{M} \nabla g_j(\boldsymbol{w})\Big),\ \boldsymbol{\theta} = \{\boldsymbol{w}\}, \tag{14}$$

where the dependency between inner- and outer-level sampling are not considered. One can independently sample inner layer data to estimate $\widetilde{g} \approx \frac{1}{M}\sum_{j=1}^{M} g_j(\boldsymbol{w})$ and $\nabla\widetilde{g} \approx \frac{1}{M}\sum_{j=1}^{M} \nabla g_j(\boldsymbol{w})$, sample outer layer data to estimate $\nabla\widetilde{f} \approx \frac{1}{N}\sum_{i=1}^{N} \nabla f_i(\widetilde{g})$, then estimate $\nabla F(\boldsymbol{\theta})$ by using $[\nabla\widetilde{f}]^{\top}\nabla\widetilde{g}$.

By casting the optimizaion problem in GCN as composite optimization problem in Eq. 13, we have

$$\mathcal{L}(\boldsymbol{\theta}) = \frac{1}{B}\sum_{i\in\mathcal{V}_{\mathcal{B}}} \mathrm{Loss}(\boldsymbol{h}_i^{(L)}, y_i),\ \boldsymbol{\theta} = \{\mathbf{W}^{(1)}\}$$
$$\mathbf{H}^{(L)} = \sigma(\widetilde{\mathbf{L}}^{(L)}\mathbf{X}\widetilde{\mathbf{W}}^{(L)}),\ \widetilde{\mathbf{W}}^{(L)} = \sigma\Big(\widetilde{\mathbf{L}}^{(L-1)}\mathbf{X}\sigma\Big(\widetilde{\mathbf{L}}^{(L-2)}\mathbf{X}\dots\underbrace{\sigma\Big(\widetilde{\mathbf{L}}^{(1)}\mathbf{X}\mathbf{W}^{(1)}\Big)}_{\widetilde{\mathbf{W}}^{(2)}}\dots\Big), \tag{15}$$

which is different from the vanilla GCN model. To see this, we note that in vanilla GCNs, since the sampled nodes at the $\ell$th layer are dependent from the nodes sampled at the $(\ell+1)$th layer, we have $\mathbb{E}[\widetilde{\mathbf{L}}^{(\ell)}] = \mathbf{P}^{(\ell)} \neq \mathbf{L}$. However in Eq. 15, since the sampled nodes have no dependency on the weight matrices or nodes sampled at other layers, we can easily obtain $\mathbb{E}[\widetilde{\mathbf{L}}^{(\ell)}] = \mathbf{L}$. These key differences makes the analysis more involved and are reflected in all three theorems, that give us different results.

**Different gradient computation and algorithm.** The stochastic gradients to update the parameters in Eq. 15 are computed as

$$\frac{\partial\mathcal{L}(\boldsymbol{\theta})}{\partial\widetilde{\mathbf{W}}^{(\ell)}} = \frac{\partial\mathcal{L}(\boldsymbol{\theta})}{\partial\widetilde{\mathbf{W}}^{(L)}}\Big(\prod_{j=\ell+1}^{L} \frac{\partial\widetilde{\mathbf{W}}^{(j)}}{\partial\widetilde{\mathbf{W}}^{(j-1)}}\Big). \tag{16}$$

However in GCN, there are two types of gradient at each layer (i.e., $\widetilde{\mathbf{D}}^{(\ell)}$ and $\widetilde{\mathbf{G}}^{(\ell)}$) that are fused with each other (i.e., $\widetilde{\mathbf{D}}^{(\ell)}$ is a part of $\widetilde{\mathbf{G}}^{(\ell-1)}$ and $\widetilde{\mathbf{D}}^{(\ell)}$ is a part of $\widetilde{\mathbf{D}}^{(\ell-1)}$) but with different functionality. $\widetilde{\mathbf{D}}^{(\ell)}$ is passing gradient between different layers, $\widetilde{\mathbf{G}}^{(\ell)}$ is passing gradient to weight matrices.

These two types of gradient and their coupled relation make both algorithm and analysis different from Zhang & Xiao (2019b). For example in Zhang & Xiao (2019b), the zeroth-order variance reduction is applied to $\widetilde{\mathbf{W}}_t^{(\ell)}$ in Eq. 13 (please refer to Algorithm 3 in Zhang & Xiao (2019b)),

where $\widetilde{\mathbf{W}}_{t-1}^{(\ell)}$ is used as a control variant to reduce the variance of $\widetilde{\mathbf{W}}_t^{(\ell)}$, i.e.,

$$\widetilde{\mathbf{W}}_t^{(\ell+1)} = \widetilde{\mathbf{W}}_{t-1}^{(\ell+1)} + \sigma(\widetilde{\mathbf{L}}_t^{(\ell)}\mathbf{X}\widetilde{\mathbf{W}}_t^{(\ell)}) - \sigma(\widetilde{\mathbf{L}}_t^{(\ell)}\mathbf{X}\widetilde{\mathbf{W}}_{t-1}^{(\ell)}). \tag{17}$$

However in SGCN++, the zeroth-order variance reduction is applied to $\widetilde{\mathbf{H}}_t^{(\ell)}$. Because the node sampled at the $t$th and $(t-1)$th iteration are unlikely the same, we cannot directly use $\mathbf{H}_{t-1}^{(\ell)}$ to reduce the variance of $\mathbf{H}_t^{(\ell)}$. Instead, the control variant in SGCN++ is computed by applying historical weight $\mathbf{W}_{t-1}^{(\ell)}$ on the historical node embedding from previous layer $\mathbf{H}_{t-1}^{(\ell-1)}$, i.e.,

$$\widetilde{\mathbf{H}}_t^{(\ell)} = \widetilde{\mathbf{H}}_{t-1}^{(\ell)} + \sigma(\widetilde{\mathbf{L}}_t^{(\ell)}\mathbf{H}_t^{(\ell-1)}\mathbf{W}_t^{(\ell)}) - \sigma(\widetilde{\mathbf{L}}_t^{(\ell)}\mathbf{H}_{t-1}^{(\ell-1)}\mathbf{W}_{t-1}^{(\ell)}). \tag{18}$$

These changes are not simply heuristic modifications, but all reflected in the analysis and the result.

**Different theoretical results and intuition.** The aforementioned differences further result in a novel analysis of Theorem 1, where we show that the vanilla sampling-based GCNs suffer a residual error $\Delta_\mathbf{b}$ that is not decreasing as the number of iterations $T$ increases, and this residual error is strongly connected to the difference between sampled and full Laplacian matrices. This is one of our novel observations for GCNs, when compared to (1) multi-level composite optimization with layerwise changing learning rate Yang et al. (2019); Chen et al. (2020), (2) variance reduction based methods Zhang & Xiao (2019b), and (3) the previous analysis on the convergence of GCNs Chen et al. (2018); Chen & Luss (2018). Our observation can be used as a theoretical motivation on using first-order and doubly variance reduction, and can mathematically explain why VRGCN outperform GraphSAGE, even with fewer nodes during training. Furthermore, as the algorithm and gradient computation are different, the theoretical results in Theorems 2 and 3 are also different.

# B    EXPERIMENT CONFIGURATIONS

**Hardware specification and environment**. We run our experiments on a single machine with Intel i5-7500, NVIDIA GTX1080 GPU (8GB memory) and, 32GB RAM memory. The code is written in Python 3.7 and we use PyTorch 1.4 on CUDA 10.1 to train the model on GPU. During each epoch, we randomly construct 10 mini-batches in parallel.

**Implementation details.** To demonstrate the effectiveness of doubly variance reduction, we modified the PyTorch implementation of GCN (Kipf & Welling, 2016)[3] to add LADIES (Zou et al., 2019), FastGCN (Chen et al., 2018), GraphSAGE (Hamilton et al., 2017), GraphSAINT (Zeng et al., 2019), VRGCN (Chen et al., 2017), and Exact sampling mechanism. Then, we implement SGCN+ and SGCN++ on the top of each sampling method to illustrate how zeroth-order variance reduction and doubly variance reduction help for GCN training.

By default, we train 2-layer GCNs with hidden state dimension of 256, element-wise ELU as the activation function and symmetric normalized Laplacian matrix $\mathbf{L} = \mathbf{D}^{-1/2}\mathbf{A}\mathbf{D}^{-1/2}$. We use *mean-aggregation* for single-class classification task and *concatenate-aggregation* for multi-class classification. The default mini-batch batch size and sampled node size are summarized in Table 2. We update the model using Adam optimizer with a learning rate of 0.01. For SGCN++, historical node embeddings are first calculated on GPUs and transfer to CPU memory using PyTorch command Tensor.to(device). Therefore, no extra GPU memory is required when training with SGCN++. To balance the staleness of snapshot model and the computational efficiency, as default we choose snapshot gap $K = 10$ and early stop inner-loop if the Euclidean distance between current step gradient to snapshot gradient is larger than 0.002 times the norm of snapshot gradient.

Table 2: Configuration of different sampling algorithms during training

|  | GraphSAGE | VRGCN | Exact | FastGCN | LADIES | GraphSAINT |
|---|---|---|---|---|---|---|
| **Mini-batch size** | 512 | 512 | 512 | 512 | 512 | 2048 |
| **Sampled neighbors** | 5 | 2 | All | - | - | - |
| **Samples in layerwise** | - | - | - | 4096 | 512 | 2048 |

[3]https://github.com/tkipf/pygcn

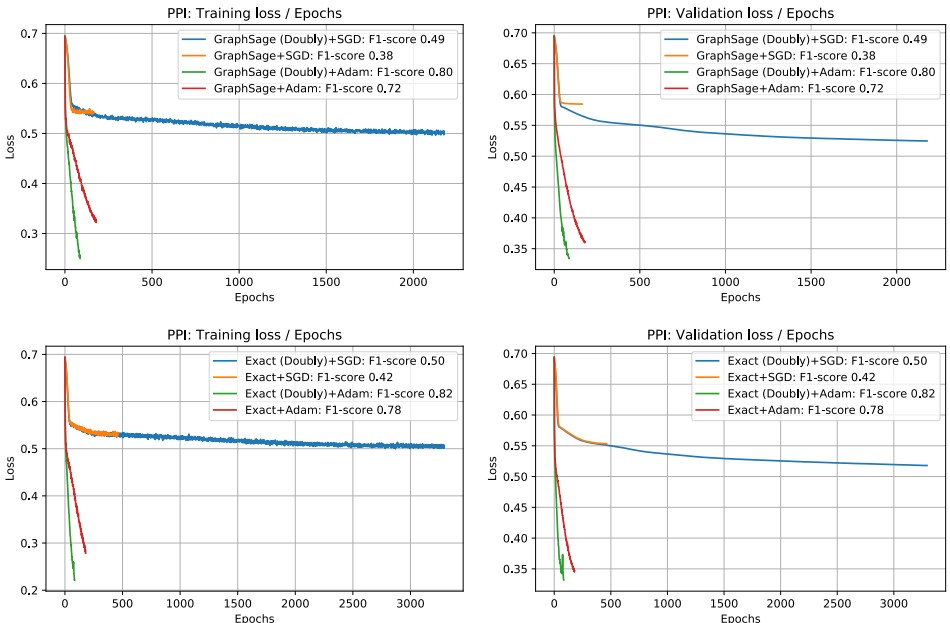

Figure 3: Comparison of doubly variance reduction and vanilla sampling-based GCN training on PPI dataset with SGD (learning rate 0.1) and Adam optimizer (learning rate 0.01). All other configurations are as default.

During training, for each epoch we construct 10 mini-batches in parallel using Python package `multiprocessing` and perform training on the sampled 10 mini-batches. To achieves a fair comparison of different sampling strategies in terms of sampling complexity, we implement all sampling algorithms using `numpy.random` and `scipy.sparse` package.

We have to emphasize that, in order to better observe the impact of sampling on convergence, we have not use any augmentation methods (e.g., "layer normalization", "skip-connection", and "attention"), which have been proven to impact the GCN performance in Cai et al. (2020); Dwivedi et al. (2020). Notice that we are not criticizing the usage of these augmentations. Instead, we use the most primitive network structure to better explore the impact of sampling and variance reduction on convergence.

**Comparison of SGD and Adam.** It is worth noting that Adam optimizer is used as the default optimizer during training. We choose Adam optimizer over SGD optimizer for the following reasons:

(a) Baseline methods training with SGD cannot converge when using a constant learning rate due to the bias and variance in stochastic gradient (Adam has some implicit variance reduction effect, which can alleviate the issue). The empirical result of SGD trained baseline models has a huge performance gap to the one trained with Adam, which makes the comparison meaningless. For example in Figure 3, we compare Adam and SGD optimizer on PPI dataset. For Adam optimizer we use PyTorch's default learning rate 0.01, and for SGD optimizer we choose learning rate as 0.1, which is selected as the most stable learning rate from range $[0.01, 1]$ for this dataset. Although the SGD is using a learning rate 10 times larger than Adam, it requires 100 times more iterations than Adam to reach the early stop point (valid loss do not decrease for 200 iterations), and suffers a giant performance gap when comparing to Adam optimizer.

(b) Most public implementation of GCNs, including all implementations in PyTorch Geometric and DGL packages, use Adam optimizer instead of SGD optimizer.

(c) In this paper, we mainly focus on how to estimate a stabilized stochastic gradient, instead of how to take the existing gradient for weight update. We employ Adam optimizer for all algorithms during experiment, which lead to a fair comparison.

**Dataset statistics.** We summarize the dataset statistics in Table 3.

Table 3: Summary of dataset statistics. **m** stands for **m**ulti-class classification, and **s** stands for **s**ingle-class.

| Dataset | Nodes | Edges | Degree | Feature | Classes | Train/Val/Test |
|---|---|---|---|---|---|---|
| **PPI** | $14,755$ | $225,270$ | $15$ | $50$ | $121(\mathbf{m})$ | $66\%/12\%/22\%$ |
| **PPI-Large** | $56,944$ | $818,716$ | $14$ | $50$ | $121(\mathbf{m})$ | $79\%/11\%/10\%$ |
| **Flickr** | $89,250$ | $899,756$ | $10$ | $500$ | $7(\mathbf{s})$ | $50\%/25\%/25\%$ |
| **Reddit** | $232,965$ | $11,606,919$ | $50$ | $602$ | $41(\mathbf{s})$ | $66\%/10\%/24\%$ |
| **Yelp** | $716,847$ | $6,977,410$ | $10$ | $300$ | $100(\mathbf{m})$ | $75\%/10\%/15\%$ |

# C    ADDITIONAL EXPERIMENTS

**Effective of variance reduction.** In Figure 4, we empirically evaluate the effectiveness of variance reduction by comparing the mean-square error of the stochastic gradient and training loss curve of different sampling strategies on `Reddit` dataset.

Because `Exact` sampling is using all neighbors for the node embedding approximation, it is only affected by layerwise gradient variance (first-order variance). Therefore, employing first-order variance reduction on `Exact` sampling can significantly reduce the mean-square error of stochastic gradient to full-gradient, and speed up the convergence.

Different from `Exact` sampling that the exact node embeddings are available during training, layer-wise sampling algorithm `LADIES`, and nodewise sampling `GraphSAGE` are approximating the true node embeddings using a subset of nodes (neighbors). Therefore, these methods both suffer from node embedding approximation variance (zeroth-order variance) and layerwise gradient variance (first-order variance). As a result, applying zeroth-order variance reduction and first-order variance reduction simultaneously is necessary to reduce the mean-square error of the stochastic gradient and speed up the convergence.

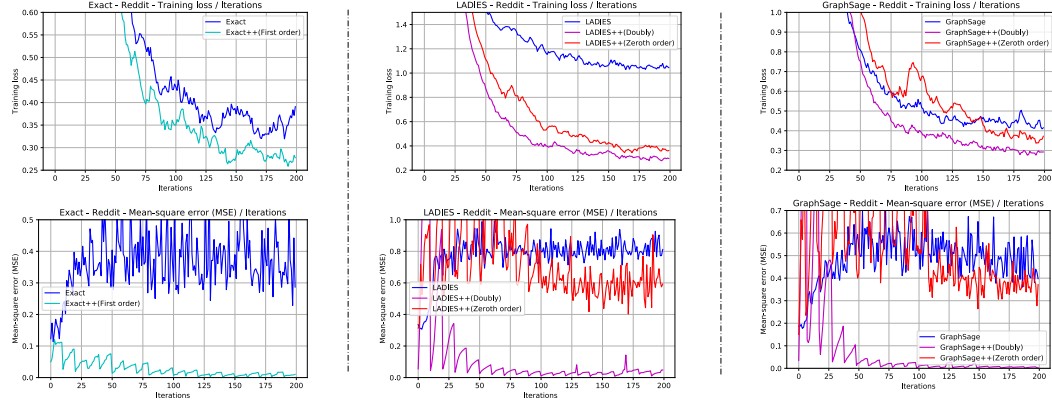

Figure 4: Comparing the mean-square error of stochastic gradient to full gradient and training loss of `SGCN`, `SGCN+`, `SGCN++` in the first 200 iterations of training process on `Reddit` dataset.

**GPU memory usage.** In Figure 5, We compare the GPU memory usage of `SGCN` and `SGCN++`. We calculate the allocated memory by `torch.cuda.memory_allocated`, which is the current GPU memory occupied by tensors in bytes for a given device. We calculate the maximum allocated memory by `torch.cuda.max_memory_allocated`, which is the maximum GPU memory occupied by tensors in bytes for a given device.

From Figure 5, we observe that neither running full-batch GCN nor saving historical node embeddings and gradients will significantly increase the computation overhead during training. Besides, since all historical activations are stored outside GPU, we see that `SGCN++` only requires several megabytes to transfer data between GPU memory to the host, which can be ignored compared to the memory usage of calculation itself.

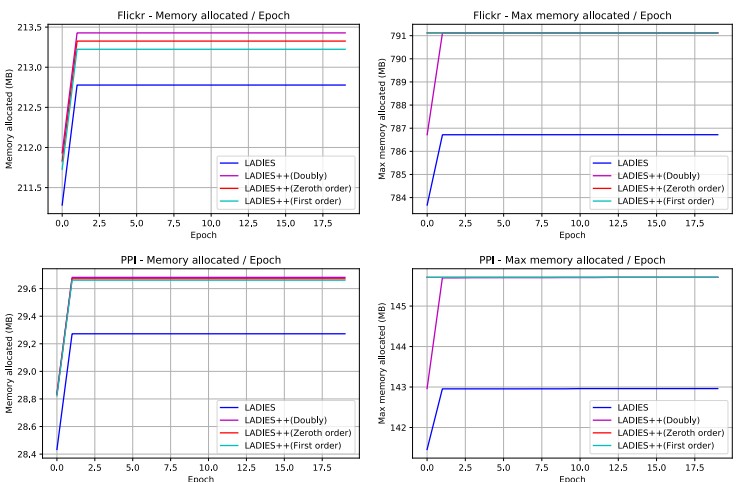

Figure 5: Comparison of GPU memory usage of `SGCN` and `SGCN++` on Flickr and PPI dataset.

**Evaluation of total time**. In Table 4 and Table 5, we report the average time of doubly variance reduced `LADIES++` and vanilla `LADIES`. We classify the wall clock time during the training process into five categories:

- **Snapshot step sampling time**: The time used to construct the snapshot full-batch or the snapshot large-batch. In practice, we directly use full-batch training for the smaller datasets (e.g., PPI, PPI-large, and Flickr) and use sampled snapshot large-batch for large datasets (e.g., Reddit and Yelp). When constructing snapshot large-batch, the `Exact` sampler has to go through all neighbors of each node using for-loops based on the graph structure, such that it is time-consuming.
- **Snapshot step transfer time**: The time required to transfer the sampled snapshot batch nodes and Laplacian matrices to the GPUs.
- **Regular step sampling time**: The time used to construct the mini-batches using layerwise `LADIES` sampler.
- **Regular step transfer time**: The time required to transfer the sampled mini-batch nodes and Laplacian matrices to GPUs, and the time to transfer the historical node embeddings and the stochastic gradient between GPUs and CPUs.
- **Computation time**: The time used for forward- and backward-propagation.

Notice that we are reporting the total time per iteration because the vanilla sampling-based method cannot reach the same accuracy as the doubly variance reduced algorithm (due to the residual error as shown in Theorem 1).

From Table 4 and Table 5, we can observe that the most time-consuming process in sampling-based GCN training is data sampling and data transfer. The extra computation time introduces by employing the snapshot step is negligible when comparing to the mini-batch sampling time during each regular step. Therefore, a promising future direction for large-scale graph training is developing a provable sampling algorithm with low sampling complexity.

**Evaluation of snapshot gap for SGCN+ and SGCN++.** Doubly variance reduced `SGCN++` requires performing full-batch (large-batch) training periodically to calculate the snapshot node embeddings and gradients. A larger snapshot gap $K$ can make training faster, but also might make the snapshot node embeddings and gradients too stale for variance reduction. In this experiment, we evaluate the effect of snapshot grap on training by choosing mini-batch size as $B = 512$ and change the inner-loop intervals from $K = 5$ mini-batches to $K = 20$ mini-batches. In Figure 6 and Figure 7, we show

Table 4: Comparison of average time (1 snapshot step and 10 regular steps) of doubly variance reduced `LADIES++` with regular step batch size as 512. Full-batch is used for snapshot step on PPI, PPI-Large, and Flickr. 50% training set nodes are sampled for the snapshot step on Reddit, and 15% training set nodes are sampled for the snapshot step on Yelp.

| Time (second) | PPI | PPI-Large | Flickr | Reddit | Yelp |
|---|---|---|---|---|---|
| Snapshot step sampling | 0.182 | 0.355 | 0.221 | 18.446 | 21.909 |
| Snapshot step transfer | 0.035 | 0.070 | 0.036 | 0.427 | 0.176 |
| Regular step sampling | 1.128 | 1.322 | 0.899 | 9.499 | 9.102 |
| Regular step transfer | 0.393 | 0.459 | 0.250 | 0.550 | 0.372 |
| Computation | 0.215 | 0.196 | 0.136 | 0.399 | 0.139 |
| Total time | 1.954 | 2.377 | 1.442 | 29.321 | 31.697 |

Table 5: Comparison of average time (10 regular steps) of `LADIES` with regular step batch size as 512

| Time (second) | PPI | PPI-Large | Flickr | Reddit | Yelp |
|---|---|---|---|---|---|
| Regular step sampling | 1.042 | 1.077 | 0.977 | 9.856 | 9.155 |
| Regular step transfer | 0.036 | 0.047 | 0.016 | 0.496 | 0.041 |
| Computation | 0.077 | 0.068 | 0.034 | 0.029 | 0.082 |
| Total time | 1.156 | 1.192 | 1.028 | 10.381 | 9.278 |

the comparison of training loss and validation loss with different number of inner-loop intervals for `SGCN++` and `SGCN+` on `Reddit` dataset, respectively. We can observe that the model with a smaller snapshot gap requires less number of iterations to reach the same training and validation loss, and gives us a better generalization performance (F1-score).

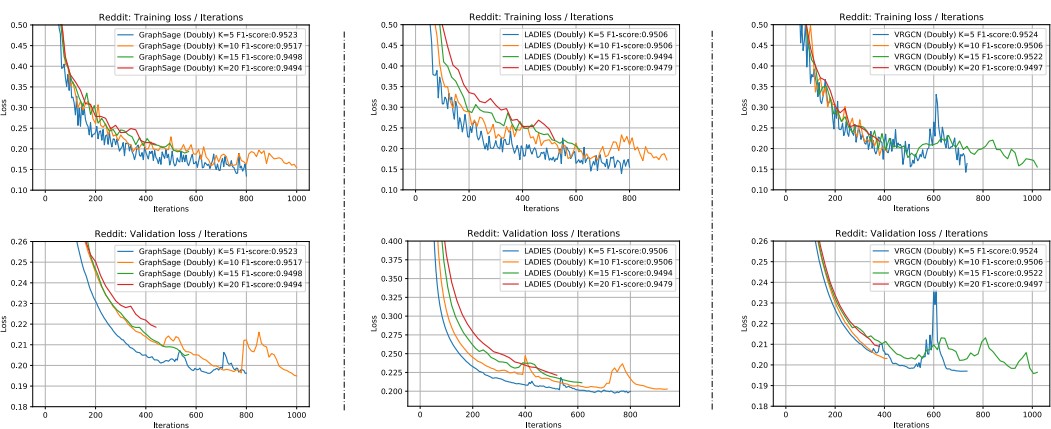

Figure 6: Comparison of training loss, validation loss, and F1-score of `SGCN++` with different snapshot gap on `Reddit` dataset.

**Evaluation of large-batch size for `SGCN+` and `SGCN++`.** The full-batch gradient calculation at each snapshot step is computationally expensive. Heuristically, we can approximate the full-batch gradient by using the gradient computed on a large-batch of nodes. Besides, it is worth noting that large-batch approximation can be also used for the node embedding approximation in zeroth-order variance reduction. In `SGCN+`, saving the historical node embeddings for all nodes in an extreme large graph can be computationally prohibitive. An alternative strategy is sampling a large-batch during the snapshot step, computing the node embeddings for all nodes in the large-batch, and saving the freshly computed node embeddings on the storage. After that, mini-batch nodes are sampled from the large-batch during the regular steps. Let denote $B'$ as the snapshot step large-batch size and $B$ denote the regular step mini-batch size. By default, we choose snapshot gap as

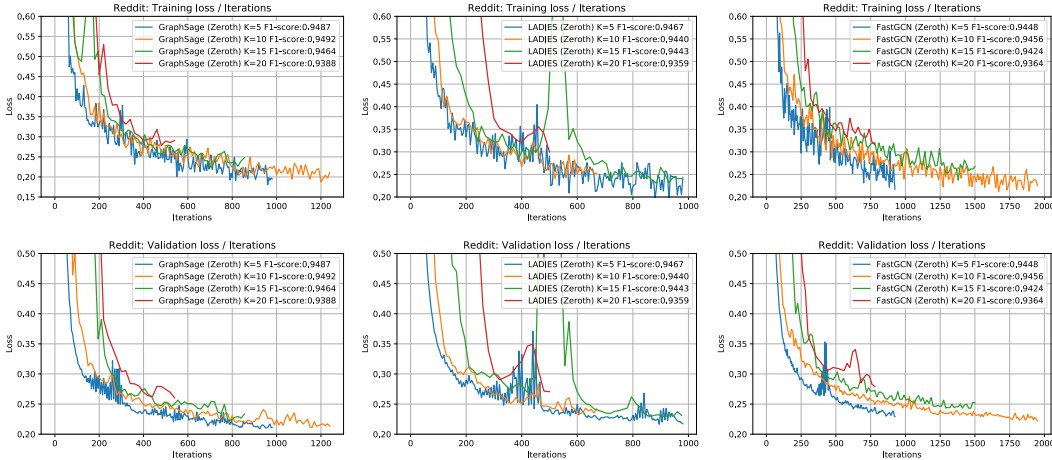

Figure 7: Comparison of training loss, validation loss, and F1-score of `SGCN+` with different snapshot gap on `Reddit` dataset.

$K = 10$, fix the regular step batch size as $B = 512$, and change the snapshot step batch size $B'$ from $20,000$ (20K) to $80,000$ (80K). In Figure 8 and Figure 9, we show the comparison of training loss and validation loss with different snapshot step large-batch size $B'$ for `SGCN++` and `SGCN+`, respectively.

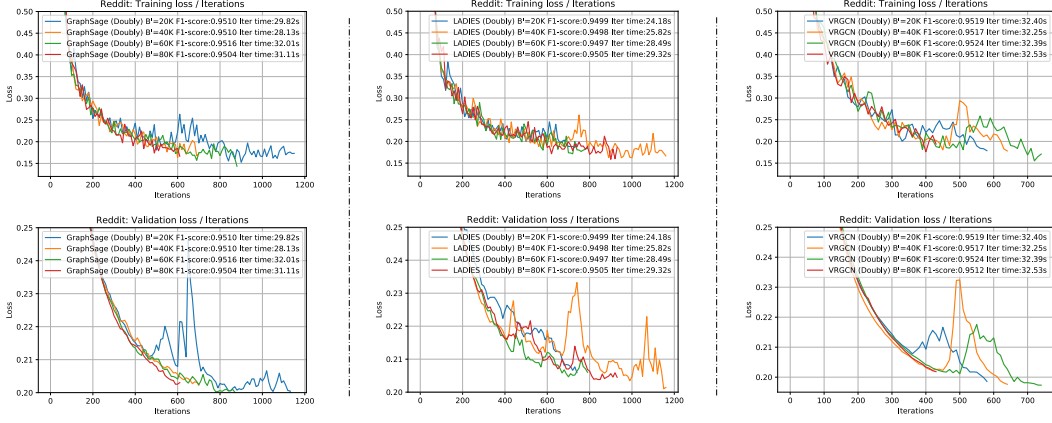

Figure 8: Comparison of training loss, validation loss, and F1-score of `SGCN++` with different snapshot large-batch size on `Reddit` dataset.

**The effect of mini-batch size.** In Figure 10, we show the comparsion of training loss and validation loss with different regular step mini-batch size. By default, we choose the snapshot gap as $K = 10$, fix the snapshot step batch size as $B' = 80,000$, and change the regular step mini-batch size $B$ from $256$ to $2,048$. Besides, we note that subgraph sampling algorithm `GraphSAINT` requires an extreme large mini-batch size every iterations. In Figure 11, we explicitly compare the effectiveness of mini-batch size on doubly variance reduced `GraphSAINT++` and vanilla `GraphSAINT`, and show that a smaller mini-batch is required by `GraphSAINT++`.

**Evaluation of increasing snapshot gap.** Snapshot gap size $K$ serves as a budge hyper-parameter that balances between training speed and the quality of variance reduction. During training, as the number of iterations increases, the GCN models convergences to a saddle point. Therefore, it is interesting to explore whether increasing the snapshot gap $K$ during the training process can obtain

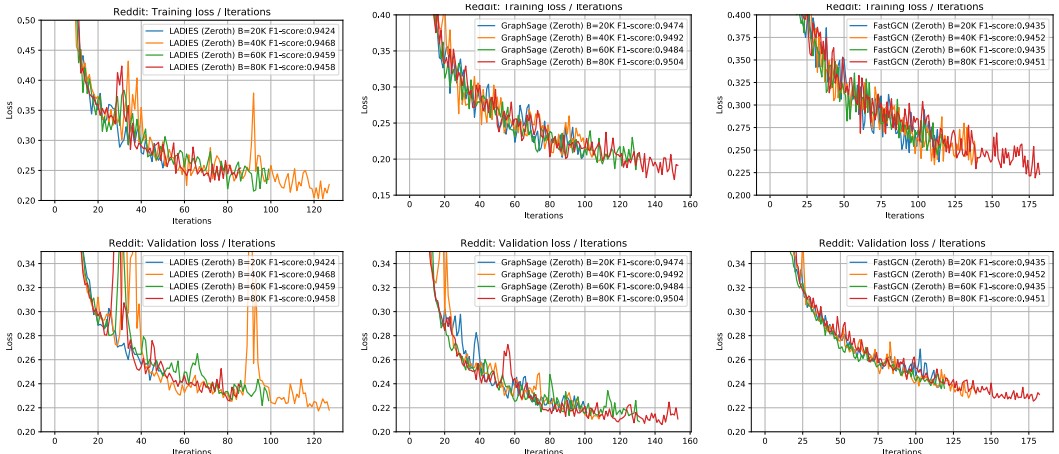

Figure 9: Comparison of training loss, validation loss, and F1-score of `SGCN+` with different snapshot large-batch size on `Reddit` dataset.

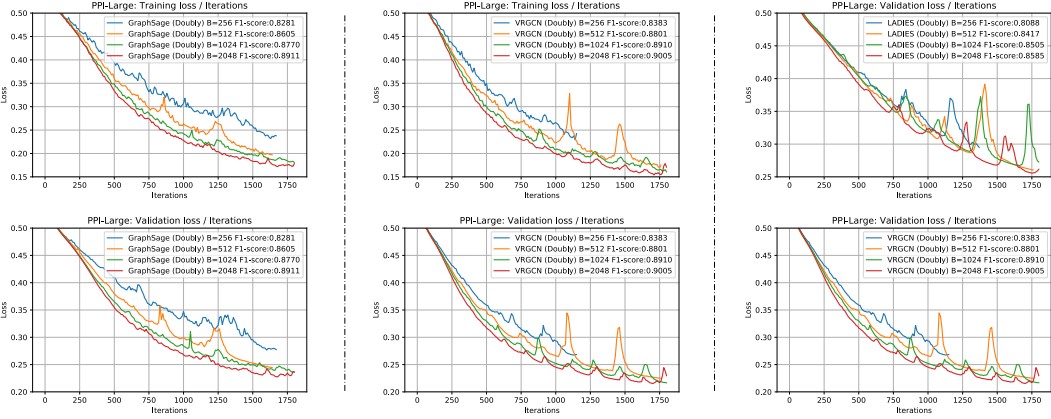

Figure 10: Comparison of training loss, validation loss, and F1-score of `SGCN++` with different mini-batch size on `Reddit` dataset.

a speed boost. In Figure 12, we show the comparison of validation loss of fixed snapshot gap $K = 10$ and gradually increasing snapshot gap $K = 10 + 0.1 \times s, s = 1, 2, \ldots$, where $s$ is the number of snapshot steps has been computed. Recall that the key bottleneck for `SGCN++` is memory budget and sampling complexity, rather than snapshot computing. Dynamically increasing snapshot gap can reduce the number of snapshot steps, but cannot significantly reduce the training time but might lead to a performance drop.

## D    DIFFERENT SAMPLING STRATEGIES

In this section, we highlight the difference between node-wise sampling, layer-wise sampling, and subgraph sampling algorithms.

**Node-wide sampling.**    The main idea of node-wise sampling is to first sample all the nodes needed for the computation using neighbor sampling (NS), then train the GCN based on the sampled nodes. For each node in the $\ell$th GCN layer, NS randomly samples $s$ of its neighbors at the $(\ell-1)$th GCN layer and formulate $\widetilde{\mathbf{L}}^{(\ell)}$ by

$$\widetilde{L}_{i,j}^{(\ell)} = \begin{cases} \frac{|\mathcal{N}(i)|}{s} \times L_{i,j}, & \text{if } j \in \widetilde{\mathcal{N}}^{(\ell)}(i) \\ 0, & \text{otherwise} \end{cases} \tag{19}$$

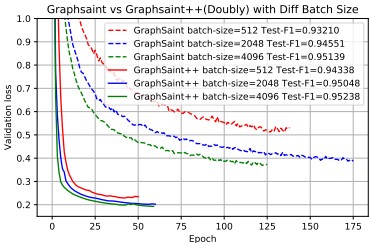

Figure 11: Comparing the validation loss and F1-score of `GraphSAINT` and `GraphSAINT++` with different mini-batch size on `Reddit` dataset

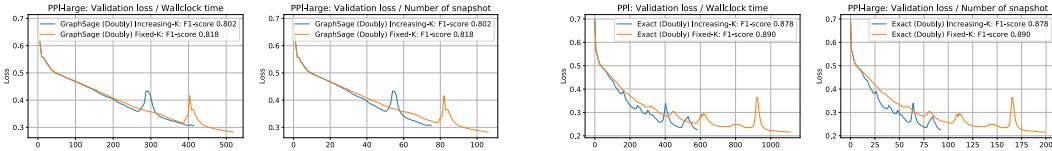

Figure 12: Effectiveness of gradually increasing snapshot gap $K$ during training on wallclock time (second) and accuracy on `PPI` dataset. We choose snapshot gap $K = 10$ for fixed-$K$. For increasing-$K$, we choose snapshot gap $K = 10 + 0.1 \times s, s = 1, 2, \ldots$, where $s$ is the number of snapshot steps.

where $\mathcal{N}(i)$ is the full set of $i$th node neighbor, $\widetilde{\mathcal{N}}^{(\ell)}(i)$ is the sampled neighbors of node $i$ for $\ell$th GCN layer. `GraphSAGE` Hamilton et al. (2017) follows the spirit of node-wise sampling where it performs uniform node sampling on the previous layer neighbors for a fixed number of nodes to bound the mini-batch computation complexity.

**Layer-wise sampling.** To avoid the neighbor explosion issue, layer-wise sampling is introduced to controls the size of sampled neighborhoods in each layer. For the $\ell$th GCN layer, layer-wise sampling methods sample a set of nodes $\mathcal{B}^{(\ell)} \subseteq \mathcal{V}$ of size $s$ under the distribution $\boldsymbol{p}$ to approximate the Laplacian by

$$\widetilde{L}_{i,j}^{(\ell)} = \begin{cases} \frac{1}{s \times p_j} \times L_{i,j}, & \text{if } j \in \mathcal{B}^{(\ell)} \\ 0, & \text{otherwise} \end{cases} \tag{20}$$

Existing work `FastGCN` Chen et al. (2018) and `LADIES` Zou et al. (2019) follows the spirit of layer-wise sampling. `FastGCN` performs independently node sampling for each layer and applies important sampling to reduce variance and results in a constant sample size in all layers. However, mini-batches potentially become too sparse to achieve high accuracy. `LADIES` improves `FastGCN` by layer-dependent sampling. Based on the sampled nodes in the upper layer, it selects their neighborhood nodes, constructs a bipartite subgraph, and computes the importance probability accordingly. Then, it samples a fixed number of nodes based on the calculated probability, and recursively conducts such a procedure per layer to construct the whole computation graph.

**Subgraph sampling.** Subgraph sampling is similar to layer-wise sampling by restricting the sampled Laplacian matrices at each layer are identical

$$\widetilde{L}_{i,j}^{(1)} = \ldots = \widetilde{L}_{i,j}^{(L)} = \begin{cases} \frac{1}{s \times p_j} \times L_{i,j}, & \text{if } j \in \mathcal{B}^{(\ell)} \\ 0, & \text{otherwise} \end{cases} \tag{21}$$

For example, `GraphSAINT` Zeng et al. (2019) can be viewed as a special case of layer-wise sampling algorithm `FastGCN` by restricting the nodes sampled at the 1-st to $(L-1)$th layer the same as the nodes sampled at the $L$th layer. However, `GraphSAINT` requires a significant large mini-batch size compared to other layer-wise sampling methods. We leave this as a potential future direction to explore.

# E DETAILED ALGORITHMS

## E.1 DESCRIPTION

In order to help readers better compare the difference of different algorithms, we summarize the vanilla sampling-based GCN training algorithm SGCN in Algorithm 3, zeroth-order variance reduced algorithm SGCN+ in Algorithm 4, and doubly variance reduced algorithm SGCN++ in Algorithm 5. In addition, we illustrate in Figure 13 the relationship between node embedding approximation variance and layerwise gradient variance to the forward- and backward-propagation. We remark that zeroth-order variance reduction SGCN+ is only applied during forward-propagation, and doubly (zeroth- and first-order) variance reduction SGCN++ is applied during forward- and backward-propagation, simultaneously.

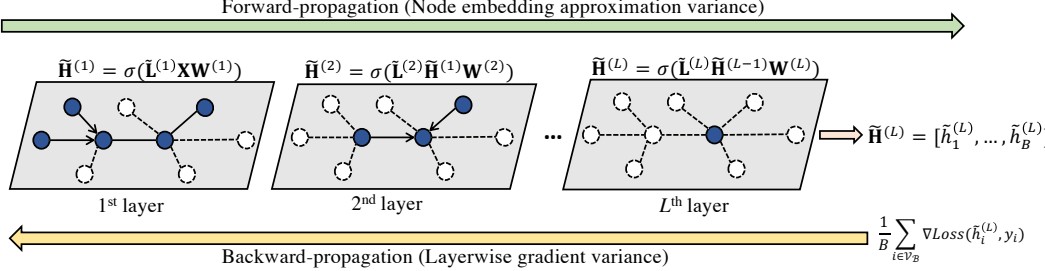

Figure 13: Relationship between the two types of variance with the training process, where embedding approximation variance (zeroth-order variance) happens during forward-propagation and layerwise gradient variance (first-order variance) happens during backward-propagation.

## E.2 SGCN

**Algorithm 3** SGCN: Vanilla sampling-based GCN training method

1: **Input:** Learning rate $\eta > 0$
2: **for** $t = 1, \ldots, T$ **do**
3:     Sample mini-batch $\mathcal{V}_{\mathcal{B}} \subset \mathcal{V}$
4:     Calculate node embeddings using

$$\widetilde{\mathbf{H}}^{(\ell)} = \sigma(\widetilde{\mathbf{L}}^{(\ell)}\widetilde{\mathbf{H}}^{(\ell-1)}\mathbf{W}^{(\ell)}), \text{ where } \widetilde{\mathbf{H}}^{(0)} = \mathbf{X}, \tag{22}$$

5:     Calculate loss as $\widetilde{\mathcal{L}}(\boldsymbol{\theta}_t) = \frac{1}{B}\sum_{i \in \mathcal{V}_{\mathcal{B}}} \text{Loss}(\widetilde{\boldsymbol{h}}_i^{(L)}, y_i)$
6:     Calculate stochastic gradient $\nabla\widetilde{\mathcal{L}}(\boldsymbol{\theta}_t) = \{\widetilde{\mathbf{G}}^{(\ell)}\}_{\ell=1}^L$ as

$$\widetilde{\mathbf{G}}_t^{(\ell)} := [\widetilde{\mathbf{L}}^{(\ell)}\widetilde{\mathbf{H}}_t^{(\ell-1)}]^\top\left(\widetilde{\mathbf{D}}_t^{(\ell+1)} \circ \nabla\sigma(\widetilde{\mathbf{Z}}_t^{(\ell)})\right),$$

$$\widetilde{\mathbf{D}}_t^{(\ell)} := [\widetilde{\mathbf{L}}^{(\ell)}]^\top\left(\widetilde{\mathbf{D}}_t^{(\ell+1)} \circ \nabla\sigma(\widetilde{\mathbf{Z}}_t^{(\ell)})\right)\mathbf{W}_t^{(\ell)}, \ \widetilde{\mathbf{D}}_t^{(L+1)} = \frac{\partial\widetilde{\mathcal{L}}(\boldsymbol{\theta}_t)}{\partial\widetilde{\mathbf{H}}^{(L)}} \tag{23}$$

7:     Update parameters as $\boldsymbol{\theta}_{t+1} = \boldsymbol{\theta}_t - \eta\nabla\widetilde{\mathcal{L}}(\boldsymbol{\theta}_t)$
8: **end for**
9: **Output:** Model with parameter $\boldsymbol{\theta}_{T+1}$

## E.3 SGCN+

**Algorithm 4** SGCN+: Zeroth-order variance reduction (Detailed version of Algorithm 4)

1: **Input:** Learning rate $\eta > 0$, snapshot gap $K > 0$
2: **for** $t = 1, \ldots, T$ **do**
3:     **if** $t \mod K = 0$ **then**
4:         % Snapshot steps

5:        Calculate node embeddings and update historical node embeddings using

$$\mathbf{Z}_t^{(\ell)} = \mathbf{LH}_t^{(\ell-1)}\mathbf{W}_t^{(\ell)}, \; \mathbf{H}_t^{(\ell)} = \sigma(\mathbf{Z}_t^{(\ell)}), \; \widetilde{\mathbf{Z}}_t^{(\ell)} \leftarrow \mathbf{Z}_t^{(\ell)} \tag{24}$$

6:        Calculate loss as $\mathcal{L}(\boldsymbol{\theta}_t) = \frac{1}{N}\sum_{i=1}^{N}\text{Loss}(\boldsymbol{h}_i^{(L)}, y_i)$

7:        Calculate full-batch gradient $\nabla\mathcal{L}(\boldsymbol{\theta}_t) = \{\mathbf{G}^{(\ell)}\}_{\ell=1}^{L}$ as

$$\mathbf{G}_t^{(\ell)} := [\mathbf{LH}_t^{(\ell-1)}]^\top\Big(\mathbf{D}_t^{(\ell+1)} \circ \nabla\sigma(\mathbf{Z}_t^{(\ell)})\Big),$$

$$\mathbf{D}_t^{(\ell)} := \mathbf{L}^\top\Big(\mathbf{D}_t^{(\ell+1)} \circ \nabla\sigma(\mathbf{Z}_t^{(\ell)})\Big)\mathbf{W}_t^{(\ell)}, \; \mathbf{D}_t^{(L+1)} = \frac{\partial\mathcal{L}(\boldsymbol{\theta}_t)}{\partial\mathbf{H}^{(L)}} \tag{25}$$

8:        Update parameters as $\boldsymbol{\theta}_{t+1} = \boldsymbol{\theta}_t - \eta\nabla\mathcal{L}(\boldsymbol{\theta}_t)$

9:    **else**

10:      % Regular steps

11:      Sample mini-batch $\mathcal{V}_\mathcal{B} \subset \mathcal{V}$

12:      Calculate node embeddings using

$$\widetilde{\mathbf{Z}}_t^{(\ell)} = \widetilde{\mathbf{Z}}_{t-1}^{(\ell)} + \widetilde{\mathbf{L}}^{(\ell)}\widetilde{\mathbf{H}}_t^{(\ell-1)}\mathbf{W}_t^{(\ell)} - \widetilde{\mathbf{L}}^{(\ell)}\widetilde{\mathbf{H}}_{t-1}^{(\ell-1)}\mathbf{W}_{t-1}^{(\ell)}, \; \widetilde{\mathbf{H}}_t^{(\ell)} = \sigma(\widetilde{\mathbf{Z}}^{(\ell)}) \tag{26}$$

13:      Calculate loss as $\widetilde{\mathcal{L}}(\boldsymbol{\theta}_t) = \frac{1}{B}\sum_{i\in\mathcal{V}_\mathcal{B}}\text{Loss}(\widetilde{\boldsymbol{h}}_i^{(L)}, y_i)$

14:      Calculate the stochastic gradient $\nabla\widetilde{\mathcal{L}}(\boldsymbol{\theta}_t) = \{\widetilde{\mathbf{G}}^{(\ell)}\}_{\ell=1}^{L}$ as

$$\widetilde{\mathbf{G}}_t^{(\ell)} := [\widetilde{\mathbf{L}}^{(\ell)}\widetilde{\mathbf{H}}_t^{(\ell-1)}]^\top\Big(\widetilde{\mathbf{D}}_t^{(\ell+1)} \circ \nabla\sigma(\widetilde{\mathbf{Z}}_t^{(\ell)})\Big),$$

$$\widetilde{\mathbf{D}}_t^{(\ell)} := [\widetilde{\mathbf{L}}^{(\ell)}]^\top\Big(\widetilde{\mathbf{D}}_t^{(\ell+1)} \circ \nabla\sigma(\widetilde{\mathbf{Z}}_t^{(\ell)})\Big)\mathbf{W}_t^{(\ell)}, \; \widetilde{\mathbf{D}}_t^{(L+1)} = \frac{\partial\widetilde{\mathcal{L}}(\boldsymbol{\theta}_t)}{\partial\widetilde{\mathbf{H}}^{(L)}} \tag{27}$$

15:      Update parameters as $\boldsymbol{\theta}_{t+1} = \boldsymbol{\theta}_t - \eta\nabla\widetilde{\mathcal{L}}(\boldsymbol{\theta}_t)$

16:    **end if**

17: **end for**

18: **Output:** Model with parameter $\boldsymbol{\theta}_{T+1}$

### E.4   `SGCN++`

---

**Algorithm 5** `SGCN++`: Doubly variance reduction (Detailed version of Algorithm 5)

---

1: **Input:** Learning rate $\eta > 0$, snapshot gap $K > 0$

2: **for** $t = 1, \ldots, T$ **do**

3:    **if** $t \bmod K = 0$ **then**

4:      % Snapshot steps

5:      Calculate node embeddings and update historical node embeddings using

$$\mathbf{Z}_t^{(\ell)} = \mathbf{LH}_t^{(\ell-1)}\mathbf{W}_t^{(\ell)}, \; \mathbf{H}_t^{(\ell)} = \sigma(\mathbf{Z}_t^{(\ell)}), \; \widetilde{\mathbf{Z}}_t^{(\ell)} \leftarrow \mathbf{Z}_t^{(\ell)} \tag{28}$$

6:      Calculate loss as $\mathcal{L}(\boldsymbol{\theta}_t) = \frac{1}{N}\sum_{i=1}^{N}\text{Loss}(\boldsymbol{h}_i^{(L)}, y_i)$

7:      Calculate the full-batch gradient $\nabla\mathcal{L}(\boldsymbol{\theta}_t) = \{\mathbf{G}^{(\ell)}\}_{\ell=1}^{L}$ as

$$\mathbf{G}_t^{(\ell)} := [\mathbf{LH}_t^{(\ell-1)}]^\top\Big(\mathbf{D}_t^{(\ell)} \circ \nabla\sigma(\mathbf{Z}_t^{(\ell)})\Big),$$

$$\mathbf{D}_t^{(\ell)} := \mathbf{L}^\top\Big(\mathbf{D}_t^{(\ell+1)} \circ \nabla\sigma(\mathbf{Z}_t^{(\ell)})\Big)\mathbf{W}_t^{(\ell)}, \; \mathbf{D}_t^{(L+1)} = \frac{\partial\mathcal{L}(\boldsymbol{\theta}_t)}{\partial\mathbf{H}^{(L)}} \tag{29}$$

8:      Save the per layerwise gradient $\widetilde{\mathbf{G}}_t^{(\ell)} \leftarrow \mathbf{G}_t^{(\ell)}, \; \widetilde{\mathbf{D}}_t^{(\ell)} \leftarrow \mathbf{D}_t^{(\ell)}$ for all $\ell \in [L]$

9:      Update parameters as $\boldsymbol{\theta}_{t+1} = \boldsymbol{\theta}_t - \eta\nabla\mathcal{L}(\boldsymbol{\theta}_t)$

10:   **else**

11:     % Regular steps

12:     Sample mini-batch $\mathcal{V}_\mathcal{B} \subset \mathcal{V}$

13:    Calculate node embeddings using

$$\widetilde{\mathbf{Z}}_t^{(\ell)} = \widetilde{\mathbf{Z}}_{t-1}^{(\ell)} + \widetilde{\mathbf{L}}^{(\ell)}\widetilde{\mathbf{H}}_t^{(\ell-1)}\mathbf{W}_t^{(\ell)} - \widetilde{\mathbf{L}}^{(\ell)}\widetilde{\mathbf{H}}_{t-1}^{(\ell-1)}\mathbf{W}_{t-1}^{(\ell)}, \ \widetilde{\mathbf{H}}_t^{(\ell)} = \sigma(\widetilde{\mathbf{Z}}^{(\ell)}) \tag{30}$$

14:    Calculate loss as $\widetilde{\mathcal{L}}(\boldsymbol{\theta}_t) = \frac{1}{B}\sum_{i\in\mathcal{V}_\mathcal{B}}\mathrm{Loss}(\widetilde{\boldsymbol{h}}_i^{(L)}, y_i)$

15:    Calculate the stochastic gradient $\nabla\widetilde{\mathcal{L}}(\boldsymbol{\theta}_t) = \{\widetilde{\mathbf{G}}^{(\ell)}\}_{\ell=1}^L$ as

$$\begin{aligned}
\widetilde{\mathbf{G}}_t^{(\ell)} &= \widetilde{\mathbf{G}}_{t-1}^{(\ell)} + [\widetilde{\mathbf{L}}^{(\ell)}\widetilde{\mathbf{H}}_t^{(\ell-1)}]^\top\Big(\widetilde{\mathbf{D}}_t^{(\ell+1)}\circ\nabla\sigma(\widetilde{\mathbf{Z}}_t)\Big) \\
&\quad - [\widetilde{\mathbf{L}}^{(\ell)}\widetilde{\mathbf{H}}_{t-1}^{(\ell-1)}]^\top\Big(\widetilde{\mathbf{D}}_{t-1}^{(\ell+1)}\circ\nabla\sigma(\widetilde{\mathbf{Z}}_{t-1})\Big) \\
\widetilde{\mathbf{D}}_t^{(\ell)} &= \widetilde{\mathbf{D}}_{t-1}^{(\ell)} + [\widetilde{\mathbf{L}}^{(\ell)}]^\top\Big(\widetilde{\mathbf{D}}_t^{(\ell+1)}\circ\nabla\sigma(\widetilde{\mathbf{Z}}_t)\Big)[\mathbf{W}_t^{(\ell)}] \\
&\quad - [\widetilde{\mathbf{L}}^{(\ell)}]^\top\Big(\widetilde{\mathbf{D}}_{t-1}^{(\ell+1)}\circ\nabla\sigma(\widetilde{\mathbf{Z}}_{t-1})\Big)[\mathbf{W}_{t-1}^{(\ell)}], \ \widetilde{\mathbf{D}}_t^{(L+1)} = \frac{\partial\widetilde{\mathcal{L}}(\boldsymbol{\theta}_t)}{\partial\widetilde{\mathbf{H}}_t^{(L)}}
\end{aligned} \tag{31}$$

16:    Update parameters as $\boldsymbol{\theta}_{t+1} = \boldsymbol{\theta}_t - \eta\nabla\widetilde{\mathcal{L}}(\boldsymbol{\theta}_t)$
17:  **end if**
18: **end for**
19: **Output:** Model with parameter $\boldsymbol{\theta}_{T+1}$

---

### E.5    SGCN++ WITHOUT FULL-BATCH

Furthermore, in Algorithm 6, we provide an alternative version of SGCN++ that does not require full-batch forward- and backward-propagation at the snapshot step. The basic idea is to approximate the full-batch gradient by sampling a large mini-batch $\mathcal{V}_\mathcal{B}'$ of size $B' = |\mathcal{V}_\mathcal{B}'|$ using Exact sampling, then compute the node embedding matrices and stochastic gradients on the sampled large-batch $\mathcal{V}_\mathcal{B}'$.

---

**Algorithm 6** SGCN++ (without full-batch): Doubly variance reduction

---

1: **Input:** Learning rate $\eta > 0$, snapshot gap $K > 0$
2: **for** $t = 1,\ldots,T$ **do**
3:    **if** $t \bmod K = 0$ **then**
4:      % Snapshot steps
5:      Sample a large-batch $\mathcal{V}_\mathcal{B}'$ of size $B'$ and construct the Laplacian matrices $\mathbf{L}^{(\ell)}$ for each layer using all neighbors, i.e.,

$$L_{i,j}^{(\ell)} = \begin{cases} L_{i,j}, & \text{if } j \in \mathcal{N}^{(\ell)}(i) \\ 0, & \text{otherwise} \end{cases} \tag{32}$$

6:      Calculate node embeddings and update historical node embeddings using

$$\mathbf{Z}_t^{(\ell)} = \mathbf{L}^{(\ell)}\mathbf{H}_t^{(\ell-1)}\mathbf{W}_t^{(\ell)}, \ \mathbf{H}_t^{(\ell)} = \sigma(\mathbf{Z}_t^{(\ell)}), \ \widetilde{\mathbf{Z}}_t^{(\ell)} \leftarrow \mathbf{Z}_t^{(\ell)} \tag{33}$$

7:      Calculate loss as $\mathcal{L}(\boldsymbol{\theta}_t) = \frac{1}{B'}\sum_{i\in\mathcal{V}_\mathcal{B}'}\mathrm{Loss}(\boldsymbol{h}_i^{(L)}, y_i)$
8:      Calculate the approximated snapshot gradient $\nabla\mathcal{L}(\boldsymbol{\theta}_t) = \{\mathbf{G}^{(\ell)}\}_{\ell=1}^L$ as

$$\begin{aligned}
\mathbf{G}_t^{(\ell)} &:= [\mathbf{L}^{(\ell)}\mathbf{H}_t^{(\ell-1)}]^\top\Big(\mathbf{D}_t^{(\ell+1)}\circ\nabla\sigma(\mathbf{Z}_t^{(\ell)})\Big), \\
\mathbf{D}_t^{(\ell)} &:= [\mathbf{L}^{(\ell)}]^\top\Big(\mathbf{D}_t^{(\ell+1)}\circ\nabla\sigma(\mathbf{Z}_t^{(\ell)})\Big)\mathbf{W}_t^{(\ell)}, \ \mathbf{D}_t^{(L+1)} = \frac{\partial\mathcal{L}(\boldsymbol{\theta}_t)}{\partial\mathbf{H}^{(L)}}
\end{aligned} \tag{34}$$

9:      Save the per layerwise gradient $\widetilde{\mathbf{G}}_t^{(\ell)} \leftarrow \mathbf{G}_t^{(\ell)}$, $\widetilde{\mathbf{D}}_t^{(\ell)} \leftarrow \mathbf{D}_t^{(\ell)}$, $\forall\ell\in[L]$
10:     Update parameters as $\boldsymbol{\theta}_{t+1} = \boldsymbol{\theta}_t - \eta\nabla\mathcal{L}(\boldsymbol{\theta}_t)$
11:   **else**
12:     % Regular steps
13:     Sample mini-batch $\mathcal{V}_\mathcal{B} \subset \mathcal{V}_\mathcal{B}'$
14:     Calculate node embeddings using

$$\widetilde{\mathbf{Z}}_t^{(\ell)} = \widetilde{\mathbf{Z}}_{t-1}^{(\ell)} + \widetilde{\mathbf{L}}^{(\ell)}\widetilde{\mathbf{H}}_t^{(\ell-1)}\mathbf{W}_t^{(\ell)} - \widetilde{\mathbf{L}}^{(\ell)}\widetilde{\mathbf{H}}_{t-1}^{(\ell-1)}\mathbf{W}_{t-1}^{(\ell)}, \ \widetilde{\mathbf{H}}_t^{(\ell)} = \sigma(\widetilde{\mathbf{Z}}_t^{(\ell)}) \tag{35}$$

15:     Calculate loss as $\widetilde{\mathcal{L}}(\boldsymbol{\theta}_t) = \frac{1}{B} \sum_{i \in \mathcal{V}_{\mathcal{B}}} \mathrm{Loss}(\widetilde{\boldsymbol{h}}_i^{(L)}, y_i)$

16:     Calculate the stochastic gradient $\nabla \widetilde{\mathcal{L}}(\boldsymbol{\theta}_t) = \{\widetilde{\mathbf{G}}^{(\ell)}\}_{\ell=1}^{L}$ as

$$
\begin{aligned}
\widetilde{\mathbf{G}}_t^{(\ell)} &= \widetilde{\mathbf{G}}_{t-1}^{(\ell)} + [\widetilde{\mathbf{L}}^{(\ell)}\widetilde{\mathbf{H}}_t^{(\ell-1)}]^\top \left( \widetilde{\mathbf{D}}_t^{(\ell+1)} \circ \nabla\sigma(\widetilde{\mathbf{Z}}_t) \right) \\
&\quad - [\widetilde{\mathbf{L}}^{(\ell)}\widetilde{\mathbf{H}}_{t-1}^{(\ell-1)}]^\top \left( \widetilde{\mathbf{D}}_{t-1}^{(\ell+1)} \circ \nabla\sigma(\widetilde{\mathbf{Z}}_{t-1}) \right) \\
\widetilde{\mathbf{D}}_t^{(\ell)} &= \widetilde{\mathbf{D}}_{t-1}^{(\ell)} + [\widetilde{\mathbf{L}}^{(\ell)}]^\top \left( \widetilde{\mathbf{D}}_t^{(\ell+1)} \circ \nabla\sigma(\widetilde{\mathbf{Z}}_t) \right) [\mathbf{W}_t^{(\ell)}] \\
&\quad - [\widetilde{\mathbf{L}}^{(\ell)}]^\top \left( \widetilde{\mathbf{D}}_{t-1}^{(\ell+1)} \circ \nabla\sigma(\widetilde{\mathbf{Z}}_{t-1}) \right) [\mathbf{W}_{t-1}^{(\ell)}],
\end{aligned}
\tag{36}
$$

$$
\widetilde{\mathbf{D}}_t^{(L+1)} = \frac{\partial \widetilde{\mathcal{L}}(\boldsymbol{\theta}_t)}{\partial \widetilde{\mathbf{H}}_t^{(L)}}
$$

17:     Update parameters as $\boldsymbol{\theta}_{t+1} = \boldsymbol{\theta}_t - \eta \nabla\widetilde{\mathcal{L}}(\boldsymbol{\theta}_t)$

18:   **end if**

19: **end for**

20: **Output:** Model with parameter $\boldsymbol{\theta}_{T+1}$

The intuition of snapshot step large-batch approximation stems from matrix Bernstein inequality Gross (2011). More specifically, suppose given $\widetilde{\mathbf{G}}_i \in \mathbb{R}^{d \times d}$ be the stochastic gradient computed by using the $i$th node with `Exact` sampling (all neighbors are used to calculate the exact node embeddings). Suppose the different between $\widetilde{\mathbf{G}}_i$ and full-gradient $\mathbb{E}[\widetilde{\mathbf{G}}_i]$ is uniformly bounded and the variance is bounded:

$$
\|\widetilde{\mathbf{G}}_i - \mathbb{E}[\widetilde{\mathbf{G}}_i]\|_F \leq \mu, \ \mathbb{E}[\|\widetilde{\mathbf{G}}_i - \mathbb{E}[\widetilde{\mathbf{G}}_i]\|_F^2] \leq \sigma^2
\tag{37}
$$

Let $\widetilde{\mathbf{G}}'$ as the snapshot step gradient computed on the sampled large batch

$$
\widetilde{\mathbf{G}}' = \frac{1}{B'} \sum_{i \in \mathcal{V}_{\mathcal{B}}'} \widetilde{\mathbf{G}}_i
\tag{38}
$$

By matrix Bernstein inequality, we know the probability of $\|\widetilde{\mathbf{G}}' - \mathbb{E}[\widetilde{\mathbf{G}}_i]\|_F$ larger than some constant $\epsilon$ decreases exponentially as the size of the sampled large-batch size $B'$ increase, i.e.,

$$
\Pr(\|\widetilde{\mathbf{G}}' - \mathbb{E}[\widetilde{\mathbf{G}}_i]\|_F \geq \epsilon) \leq 2d \exp\left( -n \cdot \min\left\{ \frac{\epsilon^2}{4\sigma^2}, \frac{\epsilon}{2\mu} \right\} \right)
\tag{39}
$$

Therefore, by choosing a large enough snapshot step batch size $B'$, we can obtain a good approximation of full-gradient.

## F    NOTATIONS, IMPORTANT PROPOSITIONS AND LEMMAS

### F.1    NOTATIONS FOR GRADIENT COMPUTATION

We introduce the following notations to simplify the representation and make it easier for readers to understand. Let formulate each GCN layer in `FullGCN` as a function

$$
\mathbf{H}^{(\ell)} = [f^{(\ell)}(\mathbf{H}^{(\ell-1)}, \mathbf{W}^{(\ell)}) := \sigma(\underbrace{\mathbf{L}\mathbf{H}^{(\ell-1)}\mathbf{W}^{(\ell)}}_{\mathbf{Z}^{(\ell)}})] \in \mathbb{R}^{N \times d_\ell}
\tag{40}
$$

and its gradient w.r.t. the input node embedding matrix $\mathbf{D}^{(\ell)} \in \mathbb{R}^{N \times d_{\ell-1}}$ is computed as

$$
\mathbf{D}^{(\ell)} = \left[ \nabla_H f^{(\ell)}(\mathbf{D}^{(\ell+1)}, \mathbf{H}^{(\ell-1)}, \mathbf{W}^{(\ell)}) := [\mathbf{L}]^\top \left( \mathbf{D}^{(\ell+1)} \circ \sigma'(\mathbf{L}\mathbf{H}^{(\ell-1)}\mathbf{W}^{(\ell)}) \right) [\mathbf{W}^{(\ell)}]^\top \right]
\tag{41}
$$

and its gradient w.r.t. the weight matrix $\mathbf{G}^{(\ell)} \in \mathbb{R}^{d_{\ell-1} \times d_\ell}$ is computed as

$$
\mathbf{G}^{(\ell)} = \left[ \nabla_W f^{(\ell)}(\mathbf{D}^{(\ell+1)}, \mathbf{H}^{(\ell-1)}, \mathbf{W}^{(\ell)}) := [\mathbf{L}\mathbf{H}^{(\ell-1)}]^\top \left( \mathbf{D}^{(\ell+1)} \circ \sigma'(\mathbf{L}^{(\ell)}\mathbf{H}^{(\ell-1)}\mathbf{W}^{(\ell)}) \right) \right]
\tag{42}
$$

Similarly, we can formulate the calculation of node embedding matrix $\widetilde{\mathbf{H}}^{(\ell)} \in \mathbb{R}^{N \times d_\ell}$ at each GCN layer in `SGCN` as

$$\widetilde{\mathbf{H}}^{(\ell)} = [\widetilde{f}^{(\ell)}(\widetilde{\mathbf{H}}^{(\ell-1)}, \mathbf{W}^{(\ell)}) := \sigma(\underbrace{\widetilde{\mathbf{L}}^{(\ell)}\widetilde{\mathbf{H}}^{(\ell-1)}\mathbf{W}^{(\ell)}}_{\widetilde{\mathbf{Z}}^{(\ell)}})] \tag{43}$$

and its gradient w.r.t. the input node embedding matrix $\widetilde{\mathbf{D}}^{(\ell)} \in \mathbb{R}^{N \times d_{\ell-1}}$ is computed as

$$\widetilde{\mathbf{D}}^{(\ell)} = \left[\nabla_H \widetilde{f}^{(\ell)}(\widetilde{\mathbf{D}}^{(\ell+1)}, \widetilde{\mathbf{H}}^{(\ell-1)}, \mathbf{W}^{(\ell)}) := [\widetilde{\mathbf{L}}^{(\ell)}]^\top \left(\widetilde{\mathbf{D}}^{(\ell+1)} \circ \sigma'(\widetilde{\mathbf{L}}^{(\ell)}\widetilde{\mathbf{H}}^{(\ell-1)}\mathbf{W}^{(\ell)})\right)[\mathbf{W}^{(\ell)}]^\top\right] \tag{44}$$

and its gradient w.r.t. the weight matrix $\widetilde{\mathbf{G}}^{(\ell)} \in \mathbb{R}^{d_{\ell-1} \times d_\ell}$ is computed as

$$\widetilde{\mathbf{G}}^{(\ell)} = \left[\nabla_W \widetilde{f}^{(\ell)}(\widetilde{\mathbf{D}}^{(\ell+1)}, \widetilde{\mathbf{H}}^{(\ell-1)}, \mathbf{W}^{(\ell)}) := [\widetilde{\mathbf{L}}^{(\ell)}\widetilde{\mathbf{H}}^{(\ell-1)}]^\top \left(\widetilde{\mathbf{D}}^{(\ell+1)} \circ \sigma'(\widetilde{\mathbf{L}}^{(\ell)}\widetilde{\mathbf{H}}^{(\ell-1)}\mathbf{W}^{(\ell)})\right)\right] \tag{45}$$

Let us denote the gradient of loss w.r.t. the final node embedding matrix as

$$\begin{aligned}
\mathbf{D}^{(L+1)} &= \frac{\partial \text{Loss}(\mathbf{H}^{(L)}, \boldsymbol{y})}{\partial \mathbf{H}^{(L)}} \in \mathbb{R}^{N \times d_L}, [\mathbf{D}^{(L+1)}]_i = \frac{1}{N} \frac{\partial \text{Loss}(\boldsymbol{h}_i^{(L)}, y_i)}{\partial \boldsymbol{h}_i^{(L)}} \in \mathbb{R}^{d_L} \\
\widetilde{\mathbf{D}}^{(L+1)} &= \frac{\partial \text{Loss}(\widetilde{\mathbf{H}}^{(L)}, \boldsymbol{y})}{\partial \widetilde{\mathbf{H}}^{(L)}} \in \mathbb{R}^{N \times d_L}, [\widetilde{\mathbf{D}}^{(L+1)}]_i = \frac{1}{B}\mathbf{1}_{\{i \in \mathcal{V}_\mathcal{B}\}} \frac{\partial \text{Loss}(\widetilde{\boldsymbol{h}}_i^{(L)}, y_i)}{\partial \widetilde{\boldsymbol{h}}_i^{(L)}} \in \mathbb{R}^{d_L}
\end{aligned} \tag{46}$$

Notice that $\widetilde{\mathbf{D}}^{(L+1)}$ is a $N \times d_L$ matrix with the row number $i \in \mathcal{V}_\mathcal{B}$ are non-zero vectors. Then we can write the gradient for the $\ell$th weight matrix in `FullGCN` and `SGCN` as

$$\begin{aligned}
\mathbf{G}^{(\ell)} &= \nabla_W f^{(\ell)}(\nabla_H f^{(\ell+1)}(\dots \nabla_H f^{(L)}(\mathbf{D}^{(L+1)}, \mathbf{H}^{(L-1)}, \mathbf{W}^{(L)})\dots, \mathbf{H}^{(\ell)}, \mathbf{W}^{(\ell+1)}), \mathbf{H}^{(\ell-1)}, \mathbf{W}^{(\ell)}) \\
\widetilde{\mathbf{G}}^{(\ell)} &= \nabla_W \widetilde{f}^{(\ell)}(\nabla_H \widetilde{f}^{(\ell+1)}(\dots \nabla_H \widetilde{f}^{(L)}(\widetilde{\mathbf{D}}^{(L+1)}, \widetilde{\mathbf{H}}^{(L-1)}, \mathbf{W}^{(L)})\dots, \widetilde{\mathbf{H}}^{(\ell)}, \mathbf{W}^{(\ell+1)}), \widetilde{\mathbf{H}}^{(\ell-1)}, \mathbf{W}^{(\ell)})
\end{aligned} \tag{47}$$

### F.2 UPPER-BOUNDED ON THE NODE EMBEDDING MATRICES AND LAYERWISE GRADIENTS

Based on the Assumption 3, we first derive the upper-bound on the node embedding matrices and the gradient passing from $\ell$th layer node embedding matrix to the $(\ell - 1)$th layer node embedding matrix.

**Proposition 2** (Detailed version of Proposition 1). *For any $\ell \in [L]$, the Frobenius norm of node embedding matrices, gradient passing from the $\ell$th layer node embeddings to the $(\ell - 1)$th are bounded*

$$\|\mathbf{H}^{(\ell)}\|_\text{F} \leq B_H, \|\widetilde{\mathbf{H}}^{(\ell)}\|_\text{F} \leq B_H, \|\frac{\partial \sigma(\mathbf{L}\mathbf{H}^{(\ell-1)}\mathbf{W}^{(\ell)})}{\partial \mathbf{H}^{(\ell-1)}}\|_\text{F} \leq B_D, \|\frac{\partial \sigma(\widetilde{\mathbf{L}}^{(\ell)}\widetilde{\mathbf{H}}^{(\ell-1)}\mathbf{W}^{(\ell)})}{\partial \widetilde{\mathbf{H}}^{(\ell-1)}}\|_\text{F} \leq B_D \tag{48}$$

*where*

$$B_H = \left(C_\sigma B_{LA} B_W\right)^\ell B_H, \; B_D = \left(B_{LA} C_\sigma B_W\right)^{L-\ell} C_{loss} \tag{49}$$

*Proof.*

$$\begin{aligned}
\|\mathbf{H}^{(\ell)}\|_\text{F} &= \|\sigma(\mathbf{L}\mathbf{H}^{(\ell-1)}\mathbf{W}^{(\ell)})\|_\text{F} \\
&\leq C_\sigma B_{LA} B_W \|\mathbf{H}^{(\ell-1)}\|_\text{F} \\
&\leq \left(C_\sigma B_{LA} B_W\right)^\ell \|\mathbf{X}\|_\text{F} \leq \left(C_\sigma B_{LA} B_W\right)^\ell B_H
\end{aligned} \tag{50}$$

$$\begin{aligned}
\|\widetilde{\mathbf{H}}^{(\ell)}\|_\text{F} &= \|\sigma(\widetilde{\mathbf{L}}^{(\ell)}\widetilde{\mathbf{H}}^{(\ell-1)}\mathbf{W}^{(\ell)})\|_\text{F} \\
&\leq C_\sigma B_{LA} B_W \|\widetilde{\mathbf{H}}^{(\ell-1)}\|_\text{F} \\
&\leq \left(C_\sigma B_{LA} B_W\right)^\ell \|\mathbf{X}\|_\text{F} \leq \left(C_\sigma B_{LA} B_W\right)^\ell B_H
\end{aligned} \tag{51}$$

$$\|\mathbf{D}^{(\ell)}\|_{\mathrm{F}} = \left\| [\mathbf{L}]^\top \Big( \mathbf{D}^{(\ell+1)} \circ \sigma'(\mathbf{L}\mathbf{H}^{(\ell-1)}\mathbf{W}^{(\ell)}) \Big) [\mathbf{W}^{(\ell)}]^\top \right\|_{\mathrm{F}}$$

$$\leq B_{LA} C_\sigma B_W \|\mathbf{D}^{(\ell+1)}\|_{\mathrm{F}} \tag{52}$$

$$\leq \Big( B_{LA} C_\sigma B_W \Big)^{L-\ell} \|\mathbf{D}^{(L+1)}\|_{\mathrm{F}} \leq \Big( B_{LA} C_\sigma B_W \Big)^{L-\ell} C_{\mathrm{loss}}$$

$$\|\widetilde{\mathbf{D}}^{(\ell)}\|_{\mathrm{F}} = \left\| [\widetilde{\mathbf{L}}^{(\ell)}]^\top \Big( \widetilde{\mathbf{D}}^{(\ell+1)} \circ \sigma'(\widetilde{\mathbf{L}}^{(\ell)}\widetilde{\mathbf{H}}^{(\ell-1)}\mathbf{W}^{(\ell)}) \Big) [\mathbf{W}^{(\ell)}]^\top \right\|_{\mathrm{F}}$$

$$\leq B_{LA} C_\sigma B_W \|\widetilde{\mathbf{D}}^{(\ell+1)}\|_{\mathrm{F}} \tag{53}$$

$$\leq \Big( B_{LA} C_\sigma B_W \Big)^{L-\ell} \|\widetilde{\mathbf{D}}^{(L+1)}\|_{\mathrm{F}} \leq \Big( B_{LA} C_\sigma B_W \Big)^{L-\ell} C_{\mathrm{loss}}$$

$$\square$$

### F.3 Lipschitz continuity and smoothness property of graph convolution layers

Then, we derive the Lipschitz continuity of $\widetilde{f}^{(\ell)}(\cdot, \cdot)$ and its gradient. Notice that the following result hold for deterministic function $f^{(\ell)}(\cdot, \cdot)$ as well.

**Proposition 3.** $\widetilde{f}^{(\ell)}(\cdot, \cdot)$ is $C_H$-Lipschitz continuous w.r.t. the input node embedding matrix where $C_H = C_\sigma B_{LA} B_W$

*Proof.*

$$\|\widetilde{f}^{(\ell)}(\mathbf{H}_1^{(\ell-1)}, \mathbf{W}^{(\ell)}) - \widetilde{f}^{(\ell)}(\mathbf{H}_2^{(\ell-1)}, \mathbf{W}^{(\ell)})\|_{\mathrm{F}}$$

$$= \|\sigma(\widetilde{\mathbf{L}}^{(\ell)}\mathbf{H}_1^{(\ell-1)}\mathbf{W}^{(\ell)}) - \sigma(\widetilde{\mathbf{L}}^{(\ell)}\mathbf{H}_2^{(\ell-1)}\mathbf{W}^{(\ell)})\|_{\mathrm{F}}$$

$$\leq C_\sigma \|\widetilde{\mathbf{L}}^{(\ell)}\|_{\mathrm{F}} \|\mathbf{H}_1^{(\ell-1)} - \mathbf{H}_2^{(\ell-1)}\|_{\mathrm{F}} \|\mathbf{W}^{(\ell)}\|_{\mathrm{F}} \tag{54}$$

$$\leq C_\sigma B_{LA} B_W \|\mathbf{H}_1^{(\ell-1)} - \mathbf{H}_2^{(\ell-1)}\|_{\mathrm{F}}$$

$$\square$$

**Proposition 4.** $\widetilde{f}^{(\ell)}(\cdot, \cdot)$ is $C_W$-Lipschitz continuous w.r.t. the weight matrix where $C_W = C_\sigma B_{LA} B_H$

*Proof.*

$$\|\widetilde{f}^{(\ell)}(\mathbf{H}^{(\ell-1)}, \mathbf{W}_1^{(\ell)}) - \widetilde{f}^{(\ell)}(\mathbf{H}^{(\ell-1)}, \mathbf{W}_2^{(\ell)})\|_{\mathrm{F}}$$

$$= \|\sigma(\widetilde{\mathbf{L}}^{(\ell)}\mathbf{H}^{(\ell-1)}\mathbf{W}_1^{(\ell)}) - \sigma(\widetilde{\mathbf{L}}^{(\ell)}\mathbf{H}^{(\ell-1)}\mathbf{W}_2^{(\ell)})\|_{\mathrm{F}}$$

$$\leq C_\sigma \|\widetilde{\mathbf{L}}^{(\ell)}\|_{\mathrm{F}} \|\mathbf{H}^{(\ell-1)}\|_{\mathrm{F}} \|\mathbf{W}_1^{(\ell)} - \mathbf{W}_2^{(\ell)}\|_{\mathrm{F}} \tag{55}$$

$$\leq C_\sigma B_{LA} B_H \|\mathbf{W}_1^{(\ell)} - \mathbf{W}_2^{(\ell)}\|_{\mathrm{F}}$$

$$\square$$

**Proposition 5.** $\nabla_H \widetilde{f}^{(\ell)}(\cdot, \cdot, \cdot)$ is $L_H$-Lipschitz continuous where

$$L_H = \max \left\{ B_{LA} C_\sigma B_W, B_{LA}^2 B_D B_W^2 L_\sigma, B_{LA} B_D C_\sigma + B_{LA}^2 B_D B_W L_\sigma B_H \right\} \tag{56}$$

*Proof.*

$$\|\nabla_H \widetilde{f}^{(\ell)}(\widetilde{\mathbf{D}}_1^{(\ell+1)}, \widetilde{\mathbf{H}}^{(\ell-1)}, \mathbf{W}^{(\ell)}) - \nabla_H \widetilde{f}^{(\ell)}(\widetilde{\mathbf{D}}_2^{(\ell+1)}, \widetilde{\mathbf{H}}^{(\ell-1)}, \mathbf{W}^{(\ell)})\|_{\mathrm{F}}$$

$$\leq \| [\widetilde{\mathbf{L}}^{(\ell)}]^\top \Big( \widetilde{\mathbf{D}}_1^{(\ell+1)} \circ \sigma'(\widetilde{\mathbf{L}}^{(\ell)}\widetilde{\mathbf{H}}^{(\ell-1)}\mathbf{W}^{(\ell)}) \Big) [\mathbf{W}^{(\ell)}]^\top$$

$$- [\widetilde{\mathbf{L}}^{(\ell)}]^\top \Big( \widetilde{\mathbf{D}}_2^{(\ell+1)} \circ \sigma'(\widetilde{\mathbf{L}}^{(\ell)}\widetilde{\mathbf{H}}^{(\ell-1)}\mathbf{W}^{(\ell)}) \Big) [\mathbf{W}^{(\ell)}]^\top \|_{\mathrm{F}} \tag{57}$$

$$\leq B_{LA} C_\sigma B_W \|\widetilde{\mathbf{D}}_1^{(\ell+1)} - \widetilde{\mathbf{D}}_2^{(\ell+1)}\|_{\mathrm{F}}$$

$$\|\nabla_H \widetilde{f}^{(\ell)}(\widetilde{\mathbf{D}}^{(\ell+1)}, \widetilde{\mathbf{H}}_1^{(\ell-1)}, \mathbf{W}^{(\ell)}) - \nabla_H \widetilde{f}^{(\ell)}(\widetilde{\mathbf{D}}^{(\ell+1)}, \widetilde{\mathbf{H}}_2^{(\ell-1)}, \mathbf{W}^{(\ell)})\|_{\mathrm{F}}$$

$$\leq \|[\widetilde{\mathbf{L}}^{(\ell)}]^\top \left( \widetilde{\mathbf{D}}^{(\ell+1)} \circ \sigma'(\widetilde{\mathbf{L}}^{(\ell)} \widetilde{\mathbf{H}}_1^{(\ell-1)} \mathbf{W}^{(\ell)}) \right) [\mathbf{W}^{(\ell)}]^\top$$

$$- [\widetilde{\mathbf{L}}^{(\ell)}]^\top \left( \widetilde{\mathbf{D}}^{(\ell+1)} \circ \sigma'(\widetilde{\mathbf{L}}^{(\ell)} \widetilde{\mathbf{H}}_2^{(\ell-1)} \mathbf{W}^{(\ell)}) \right) [\mathbf{W}^{(\ell)}]^\top \|_{\mathrm{F}} \tag{58}$$

$$\leq B_{LA}^2 B_D B_W^2 L_\sigma \|\widetilde{\mathbf{H}}_1^{(\ell-1)} - \widetilde{\mathbf{H}}_2^{(\ell-1)}\|_{\mathrm{F}}$$

$$\|\nabla_H \widetilde{f}^{(\ell)}(\widetilde{\mathbf{D}}^{(\ell+1)}, \widetilde{\mathbf{H}}^{(\ell-1)}, \mathbf{W}_1^{(\ell)}) - \nabla_H \widetilde{f}^{(\ell)}(\widetilde{\mathbf{D}}^{(\ell+1)}, \widetilde{\mathbf{H}}^{(\ell-1)}, \mathbf{W}_2^{(\ell)})\|_{\mathrm{F}}$$

$$\leq \|[\widetilde{\mathbf{L}}^{(\ell)}]^\top \left( \widetilde{\mathbf{D}}^{(\ell+1)} \circ \sigma'(\widetilde{\mathbf{L}}^{(\ell)} \widetilde{\mathbf{H}}^{(\ell-1)} \mathbf{W}_1^{(\ell)}) \right) [\mathbf{W}_1^{(\ell)}]^\top$$

$$- [\widetilde{\mathbf{L}}^{(\ell)}]^\top \left( \widetilde{\mathbf{D}}^{(\ell+1)} \circ \sigma'(\widetilde{\mathbf{L}}^{(\ell)} \widetilde{\mathbf{H}}^{(\ell-1)} \mathbf{W}_2^{(\ell)}) \right) [\mathbf{W}_2^{(\ell)}]^\top \|_{\mathrm{F}} \tag{59}$$

$$\leq (B_{LA} B_D C_\sigma + B_{LA}^2 B_D B_W L_\sigma B_H) \|\mathbf{W}_1^{(\ell)} - \mathbf{W}_2^{(\ell)}\|_{\mathrm{F}}$$

$\square$

**Proposition 6.** $\nabla_W \widetilde{f}^{(\ell)}(\cdot, \cdot, \cdot)$ *is* $L_W$-*Lipschitz continuous where*

$$L_W = \max \left\{ B_{LA} B_H C_\sigma, B_{LA}^2 B_H^2 H_D L_\sigma, B_{LA} B_D C_\sigma + B_{LA}^2 B_H^2 B_D L_\sigma \right\} \tag{60}$$

*Proof.*

$$\|\nabla_W \widetilde{f}^{(\ell)}(\widetilde{\mathbf{D}}_1^{(\ell+1)}, \widetilde{\mathbf{H}}^{(\ell-1)}, \mathbf{W}^{(\ell)}) - \nabla_W \widetilde{f}^{(\ell)}(\widetilde{\mathbf{D}}_2^{(\ell+1)}, \widetilde{\mathbf{H}}^{(\ell-1)}, \mathbf{W}^{(\ell)})\|_{\mathrm{F}}$$

$$\leq \|[\widetilde{\mathbf{L}}^{(\ell)} \widetilde{\mathbf{H}}^{(\ell-1)}]^\top \left( \widetilde{\mathbf{D}}_1^{(\ell+1)} \circ \sigma'(\widetilde{\mathbf{L}}^{(\ell)} \widetilde{\mathbf{H}}^{(\ell-1)} \mathbf{W}^{(\ell)}) \right)$$

$$- [\widetilde{\mathbf{L}}^{(\ell)} \widetilde{\mathbf{H}}^{(\ell-1)}]^\top \left( \widetilde{\mathbf{D}}_2^{(\ell+1)} \circ \sigma'(\widetilde{\mathbf{L}}^{(\ell)} \widetilde{\mathbf{H}}^{(\ell-1)} \mathbf{W}^{(\ell)}) \right) \|_{\mathrm{F}} \tag{61}$$

$$\leq B_{LA} B_H C_\sigma \|\widetilde{\mathbf{D}}_1^{(\ell+1)} - \widetilde{\mathbf{D}}_2^{(\ell+1)}\|_{\mathrm{F}}$$

$$\|\nabla_W \widetilde{f}^{(\ell)}(\widetilde{\mathbf{D}}^{(\ell+1)}, \widetilde{\mathbf{H}}_1^{(\ell-1)}, \mathbf{W}^{(\ell)}) - \nabla_W \widetilde{f}^{(\ell)}(\widetilde{\mathbf{D}}^{(\ell+1)}, \widetilde{\mathbf{H}}_2^{(\ell-1)}, \mathbf{W}^{(\ell)})\|_{\mathrm{F}}$$

$$\leq \|[\widetilde{\mathbf{L}}^{(\ell)} \widetilde{\mathbf{H}}_1^{(\ell-1)}]^\top \left( \widetilde{\mathbf{D}}^{(\ell+1)} \circ \sigma'(\widetilde{\mathbf{L}}^{(\ell)} \widetilde{\mathbf{H}}_1^{(\ell-1)} \mathbf{W}^{(\ell)}) \right)$$

$$- [\widetilde{\mathbf{L}}^{(\ell)} \widetilde{\mathbf{H}}_2^{(\ell-1)}]^\top \left( \widetilde{\mathbf{D}}^{(\ell+1)} \circ \sigma'(\widetilde{\mathbf{L}}^{(\ell)} \widetilde{\mathbf{H}}_2^{(\ell-1)} \mathbf{W}^{(\ell)}) \right) \|_{\mathrm{F}} \tag{62}$$

$$\leq (B_{LA}^2 B_D^2 C_\sigma + B_{LA}^2 B_H B_D L_\sigma B_W) \|\widetilde{\mathbf{H}}_1^{(\ell-1)} - \widetilde{\mathbf{H}}_2^{(\ell-1)}\|_{\mathrm{F}}$$

$$\|\nabla_W \widetilde{f}^{(\ell)}(\widetilde{\mathbf{D}}^{(\ell+1)}, \widetilde{\mathbf{H}}^{(\ell-1)}, \mathbf{W}_1^{(\ell)}) - \nabla_W \widetilde{f}^{(\ell)}(\widetilde{\mathbf{D}}^{(\ell+1)}, \widetilde{\mathbf{H}}^{(\ell-1)}, \mathbf{W}_2^{(\ell)})\|_{\mathrm{F}}$$

$$\leq \|[\widetilde{\mathbf{L}}^{(\ell)} \widetilde{\mathbf{H}}^{(\ell-1)}]^\top \left( \widetilde{\mathbf{D}}^{(\ell+1)} \circ \sigma'(\widetilde{\mathbf{L}}^{(\ell)} \widetilde{\mathbf{H}}^{(\ell-1)} \mathbf{W}_1^{(\ell)}) \right)$$

$$- [\widetilde{\mathbf{L}}^{(\ell)} \widetilde{\mathbf{H}}^{(\ell-1)}]^\top \left( \widetilde{\mathbf{D}}^{(\ell+1)} \circ \sigma'(\widetilde{\mathbf{L}}^{(\ell)} \widetilde{\mathbf{H}}^{(\ell-1)} \mathbf{W}_2^{(\ell)}) \right) \|_{\mathrm{F}} \tag{63}$$

$$\leq B_{LA}^2 B_H^2 H_D L_\sigma \|\mathbf{W}_1^{(\ell)} - \mathbf{W}_2^{(\ell)}\|_{\mathrm{F}}$$

$\square$

### F.4 LIPSCHITZ CONTINOUITY OF THE GRADIENT OF GRAPH CONVOLUTIONAL NETWORK

Let first recall the parameters and gradients of a $L$-layer GCN is defined as

$$\boldsymbol{\theta} = \{\mathbf{W}^{(1)}, \ldots, \mathbf{W}^{(L)}\}, \ \nabla \mathcal{L}(\boldsymbol{\theta}) = \{\mathbf{G}^{(1)}, \ldots, \mathbf{G}^{(L)}\} \tag{64}$$

where $\mathbf{G}^{(\ell)}$ is defined as the gradient w.r.t. the $\ell$th layer weight matrix. Let us slight abuse of notation and define the distance between two set of parameters $\boldsymbol{\theta}_1, \boldsymbol{\theta}_2$ and its gradient as

$$\|\boldsymbol{\theta}_1 - \boldsymbol{\theta}_2\|_{\mathrm{F}} = \sum_{\ell=1}^L \|\mathbf{W}_1^{(\ell)} - \mathbf{W}_2^{(\ell)}\|_{\mathrm{F}}, \ \|\nabla \mathcal{L}(\boldsymbol{\theta}_1) - \nabla \mathcal{L}(\boldsymbol{\theta}_2)\|_{\mathrm{F}} = \sum_{\ell=1}^L \|\mathbf{G}_1^{(\ell)} - \mathbf{G}_2^{(\ell)}\|_{\mathrm{F}} \tag{65}$$

Then, we derive the Lipschitz continuous constant of the gradient of a $L$-layer graph convolutonal network. Notice that the above result also hold for sampling-based GCN training.

**Lemma 1.** *The gradient of an L-layer GCN is $L_{\mathrm{F}}$-Lipschitz continuous with $L_{\mathrm{F}} = L(LU_{\max L}^2 U_{\max C}^2 + U_{\max L}^2)$, i.e.,*

$$\|\nabla\mathcal{L}(\boldsymbol{\theta}_1) - \nabla\mathcal{L}(\boldsymbol{\theta}_2)\|_{\mathrm{F}}^2 \leq L_{\mathrm{F}}\|\boldsymbol{\theta}_1 - \boldsymbol{\theta}_2\|_{\mathrm{F}}^2 \tag{66}$$

*where*

$$
\begin{aligned}
U_{\max C} &= \max_{j\in\{0,\dots,L-1\}} C_H^j C_W, \\
U_{\max L} &= \Big[\min_{j\in\{0,\dots,L-\ell-1\}} L_W L_H^j\Big] \times \max\Big\{1, L_{loss}\Big\}
\end{aligned}
\tag{67}
$$

*Proof.* We first consider the gradient w.r.t. the $\ell$th graph convolutional layer weight matrix

$$
\begin{aligned}
&\|\mathbf{G}_1^{(\ell)} - \mathbf{G}_2^{(\ell)}\|_{\mathrm{F}} \\
&= \|\nabla_W f^{(\ell)}(\nabla_H f^{(\ell+1)}(\dots\nabla_H f^{(L)}(\mathbf{D}_1^{(L+1)}, \mathbf{H}_1^{(L-1)}, \mathbf{W}_1^{(L)})\dots, \mathbf{H}_1^{(\ell)}, \mathbf{W}_1^{(\ell+1)}), \mathbf{H}_1^{(\ell-1)}, \mathbf{W}_1^{(\ell)}) \\
&\quad - \nabla_W f^{(\ell)}(\nabla_H f^{(\ell+1)}(\dots\nabla_H f^{(L)}(\mathbf{D}_2^{(L+1)}, \mathbf{H}_2^{(L-1)}, \mathbf{W}_2^{(L)})\dots, \mathbf{H}_2^{(\ell)}\mathbf{W}_2^{(\ell+1)}), \mathbf{H}_2^{(\ell-1)}, \mathbf{W}_2^{(\ell)})\|_{\mathrm{F}} \\
&\leq \|\nabla_W f^{(\ell)}(\nabla_H f^{(\ell+1)}(\dots\nabla_H f^{(L)}(\mathbf{D}_1^{(L+1)}, \mathbf{H}_1^{(L-1)}, \mathbf{W}_1^{(L)})\dots, \mathbf{H}_1^{(\ell)}, \mathbf{W}_1^{(\ell+1)}), \mathbf{H}_1^{(\ell-1)}, \mathbf{W}_1^{(\ell)}) \\
&\quad - \nabla_W f^{(\ell)}(\nabla_H f^{(\ell+1)}(\dots\nabla_H f^{(L)}(\mathbf{D}_2^{(L+1)}, \mathbf{H}_1^{(L-1)}, \mathbf{W}_1^{(L)})\dots, \mathbf{H}_1^{(\ell)}, \mathbf{W}_1^{(\ell+1)}), \mathbf{H}_1^{(\ell-1)}, \mathbf{W}_1^{(\ell)})\|_{\mathrm{F}} \\
&\quad + \|\nabla_W f^{(\ell)}(\nabla_H f^{(\ell+1)}(\dots\nabla_H f^{(L)}(\mathbf{D}_2^{(L+1)}, \mathbf{H}_1^{(L-1)}, \mathbf{W}_1^{(L)})\dots, \mathbf{H}_1^{(\ell)}, \mathbf{W}_1^{(\ell+1)}), \mathbf{H}_1^{(\ell-1)}, \mathbf{W}_1^{(\ell)}) \\
&\quad - \nabla_W f^{(\ell)}(\nabla_H f^{(\ell+1)}(\dots\nabla_H f^{(L)}(\mathbf{D}_2^{(L+1)}, \mathbf{H}_2^{(L-1)}, \mathbf{W}_1^{(L)})\dots, \mathbf{H}_1^{(\ell)}, \mathbf{W}_1^{(\ell+1)}), \mathbf{H}_1^{(\ell-1)}, \mathbf{W}_1^{(\ell)})\|_{\mathrm{F}} \\
&\quad + \|\nabla_W f^{(\ell)}(\nabla_H f^{(\ell+1)}(\dots\nabla_H f^{(L)}(\mathbf{D}_2^{(L+1)}, \mathbf{H}_2^{(L-1)}, \mathbf{W}_1^{(L)})\dots, \mathbf{H}_1^{(\ell)}, \mathbf{W}_1^{(\ell+1)}), \mathbf{H}_1^{(\ell-1)}, \mathbf{W}_1^{(\ell)}) \\
&\quad - \nabla_W f^{(\ell)}(\nabla_H f^{(\ell+1)}(\dots\nabla_H f^{(L)}(\mathbf{D}_2^{(L+1)}, \mathbf{H}_2^{(L-1)}, \mathbf{W}_2^{(L)})\dots, \mathbf{H}_1^{(\ell)}, \mathbf{W}_1^{(\ell+1)}), \mathbf{H}_1^{(\ell-1)}, \mathbf{W}_1^{(\ell)})\|_{\mathrm{F}} + \dots \\
&\quad + \|\nabla_W f^{(\ell)}(\mathbf{D}_2^{(\ell+1)}, \mathbf{H}_1^{(\ell-1)}, \mathbf{W}_1^{(\ell)}) - \nabla_W f^{(\ell)}(\mathbf{D}_2^{(\ell+1)}, \mathbf{H}_2^{(\ell-1)}, \mathbf{W}_1^{(\ell)})\|_{\mathrm{F}} \\
&\quad + \|\nabla_W f^{(\ell)}(\mathbf{D}_2^{(\ell+1)}, \mathbf{H}_2^{(\ell-1)}, \mathbf{W}_1^{(\ell)}) - \nabla_W f^{(\ell)}(\mathbf{D}_2^{(\ell+1)}, \mathbf{H}_2^{(\ell-1)}, \mathbf{W}_2^{(\ell)})\|_{\mathrm{F}}
\end{aligned}
\tag{68}
$$

By the Lipschitz continuity of $\nabla_W f^{(\ell)}(\cdot)$ and $\nabla_H f^{(\ell)}(\cdot)$

$$
\begin{aligned}
\|\mathbf{G}_1^{(\ell)} - \mathbf{G}_2^{(\ell)}\|_{\mathrm{F}} &\leq L_W L_H^{L-\ell-1} L_{\mathrm{loss}}\|\mathbf{H}_1^{(L)} - \mathbf{H}_2^{(L)}\|_{\mathrm{F}} \\
&\quad + L_W L_H^{L-\ell-1}(\|\mathbf{H}_1^{(L-1)} - \mathbf{H}_2^{(L-1)}\|_{\mathrm{F}} + \|\mathbf{W}_1^{(L)} - \mathbf{W}_2^{(L)}\|_{\mathrm{F}}) + \dots \\
&\quad + L_W L_H(\|\mathbf{H}_1^{(\ell)} - \mathbf{H}_2^{(\ell)}\|_{\mathrm{F}} + \|\mathbf{W}_1^{(\ell+1)} - \mathbf{W}_2^{(\ell+1)}\|_{\mathrm{F}}) \\
&\quad + L_W(\|\mathbf{H}_1^{(\ell-1)} - \mathbf{H}_2^{(\ell-1)}\|_{\mathrm{F}} + \|\mathbf{W}_1^{(\ell)} - \mathbf{W}_2^{(\ell)}\|_{\mathrm{F}})
\end{aligned}
\tag{69}
$$

Let define $U_{\max L}$ as

$$U_{\max L} = \Big[\min_{j\in\{0,\dots,L-\ell-1\}} L_W L_H^j\Big] \times \max\Big\{1, L_{\mathrm{loss}}\Big\} \tag{70}$$

then we can rewrite the above equation as

$$\|\mathbf{G}_1^{(\ell)} - \mathbf{G}_2^{(\ell)}\|_{\mathrm{F}} \leq U_{\max L}\Big(\sum_{j=1}^{L}\|\mathbf{H}_1^{(j)} - \mathbf{H}_2^{(j)}\|_{\mathrm{F}}\Big) + U_{\max L}\Big(\sum_{j=1}^{L}\|\mathbf{W}_1^{(j)} - \mathbf{W}_2^{(j)}\|_{\mathrm{F}}\Big) \tag{71}$$

Then, let consider the upper bound of $\|\mathbf{H}_1^{(\ell)} - \mathbf{H}_2^{(\ell)}\|_{\mathrm{F}}$

$$
\begin{aligned}
\|\mathbf{H}_1^{(\ell)} - \mathbf{H}_2^{(\ell)}\|_{\mathrm{F}} &= \|f^{(\ell)}(f^{(\ell-1)}(\dots f^{(1)}(\mathbf{X}, \mathbf{W}_1^{(1)})\dots, \mathbf{W}_1^{(\ell-1)}), \mathbf{W}_1^{(\ell)}) \\
&\quad - f^{(\ell)}(f^{(\ell-1)}(\dots f^{(1)}(\mathbf{X}, \mathbf{W}_2^{(1)})\dots, \mathbf{W}_2^{(\ell-1)}), \mathbf{W}_2^{(\ell)})\|_{\mathrm{F}} \\
&\leq \|f^{(\ell)}(f^{(\ell-1)}(\dots f^{(1)}(\mathbf{X}, \mathbf{W}_1^{(1)})\dots, \mathbf{W}_1^{(\ell-1)}), \mathbf{W}_1^{(\ell)}) \\
&\quad - f^{(\ell)}(f^{(\ell-1)}(\dots f^{(1)}(\mathbf{X}, \mathbf{W}_2^{(1)})\dots, \mathbf{W}_1^{(\ell-1)}), \mathbf{W}_1^{(\ell)})\|_{\mathrm{F}} + \dots \\
&\quad + \|f^{(\ell)}(f^{(\ell-1)}(\dots f^{(1)}(\mathbf{X}, \mathbf{W}_2^{(1)})\dots, \mathbf{W}_2^{(\ell-1)}), \mathbf{W}_1^{(\ell)}) \\
&\quad - f^{(\ell)}(f^{(\ell-1)}(\dots f^{(1)}(\mathbf{X}, \mathbf{W}_2^{(1)})\dots, \mathbf{W}_2^{(\ell-1)}), \mathbf{W}_1^{(\ell)})\|_{\mathrm{F}}
\end{aligned}
\tag{72}
$$

By the Lipschitz continuity of $f^{(\ell)}(\cdot, \cdot)$ we have

$$
\begin{aligned}
\|\mathbf{H}_1^{(\ell)} - \mathbf{H}_2^{(\ell)}\|_{\mathrm{F}} &\leq C_H^{L-1} C_W \|\mathbf{W}_1^{(1)} - \mathbf{W}_2^{(1)}\|_{\mathrm{F}} + \ldots + C_W \|\mathbf{W}_1^{(\ell)} - \mathbf{W}_2^{(\ell)}\|_{\mathrm{F}} \\
&\leq \Big[ \max_{j \in \{0,\ldots,L-1\}} C_H^j C_W \Big] \Big( \sum_{j=1}^L \|\mathbf{W}_1^{(j)} - \mathbf{W}_2^{(j)}\|_{\mathrm{F}} \Big)
\end{aligned}
\tag{73}
$$

Let define $U_{\max C}$ is defined as

$$
U_{\max C} = \max_{j \in \{0,\ldots,L-1\}} C_H^j C_W \tag{74}
$$

Plugging it back we have

$$
\|\mathbf{G}_1^{(\ell)} - \mathbf{G}_2^{(\ell)}\|_{\mathrm{F}} \leq (L U_{\max L} U_{\max C} + U_{\max L}) \Big( \sum_{j=1}^L \|\mathbf{W}_1^{(j)} - \mathbf{W}_2^{(j)}\|_{\mathrm{F}} \Big) \tag{75}
$$

Summing both size from $\ell = 1$ to $\ell = L$ we have

$$
\begin{aligned}
\|\nabla \mathcal{L}(\boldsymbol{\theta}_1) - \nabla \mathcal{L}(\boldsymbol{\theta}_2)\|_{\mathrm{F}} &= \sum_{\ell=1}^L \|\mathbf{G}_1^{(\ell)} - \mathbf{G}_2^{(\ell)}\|_{\mathrm{F}} \\
&\leq L(L U_{\max L} U_{\max C} + U_{\max L}) \Big( \sum_{j=1}^L \|\mathbf{W}_1^{(j)} - \mathbf{W}_2^{(j)}\|_{\mathrm{F}} \Big) \\
&\leq L(L U_{\max L} U_{\max C} + U_{\max L}) \|\boldsymbol{\theta}_1 - \boldsymbol{\theta}_2\|_{\mathrm{F}}
\end{aligned}
\tag{76}
$$

$\square$

## G  PROOF OF THEOREM 1

By bias-variance decomposition, we can decompose the mean-square error of stochastic gradient as

$$\sum_{\ell=1}^{L}\mathbb{E}[\|\widetilde{\mathbf{G}}^{(\ell)}-\mathbf{G}^{(\ell)}\|_{\mathrm{F}}^{2}]=\sum_{\ell=1}^{L}\Big[\underbrace{\mathbb{E}[\|\mathbb{E}[\widetilde{\mathbf{G}}^{(\ell)}]-\mathbf{G}^{(\ell)}\|_{\mathrm{F}}^{2}]}_{\text{bias }\mathbb{E}[\|\mathbf{b}\|_{\mathrm{F}}^{2}]}+\underbrace{\mathbb{E}[\|\widetilde{\mathbf{G}}^{(\ell)}-\mathbb{E}[\widetilde{\mathbf{G}}^{(\ell)}]\|_{\mathrm{F}}^{2}]}_{\text{variance }\mathbb{E}[\|\mathbf{n}\|_{\mathrm{F}}^{2}]}\Big] \quad (77)$$

Therefore, we have to explicitly define the computation of $\mathbb{E}[\widetilde{\mathbf{G}}^{(\ell)}]$, which requires computing $\bar{\mathbf{D}}^{(L+1)}=\mathbb{E}[\widetilde{\mathbf{D}}^{(L+1)}]$, $\bar{\mathbf{D}}^{(\ell)}=\mathbb{E}[\widetilde{\mathbf{D}}^{(\ell)}]$, and $\bar{\mathbf{G}}^{(\ell)}=\mathbb{E}[\widetilde{\mathbf{G}}^{(\ell)}]$.

Let defined a general form of the sampled Laplacian matrix $\widetilde{\mathbf{L}}^{(\ell)}\in\mathbb{R}^{N\times N}$ as

$$\widetilde{L}_{i,j}^{(\ell)}=\begin{cases}\frac{L_{i,j}}{\alpha_{i,j}} & \text{if } i\in\mathcal{B}^{(\ell)} \text{ and } j\in\mathcal{B}^{(\ell-1)} \\ 0 & \text{otherwise}\end{cases} \quad (78)$$

where $\alpha_{i,j}$ is the weighted constant depends on the sampling algorithms.

The expectation of $\widetilde{L}_{i,j}^{(\ell)}$ is computed as

$$\mathbb{E}[\widetilde{L}_{i,j}^{(\ell)}]=\mathbb{E}_{i\in\mathcal{B}^{(\ell)}}\Big[\mathbb{E}_{j\in\mathcal{B}^{(\ell-1)}}[\widetilde{L}_{i,j}^{(\ell)} \mid i\in\mathcal{B}^{(\ell)}]\Big] \quad (79)$$

In order to compute the expectation of SGCN's node embedding matrices, let define the propagation matrix $\mathbf{P}^{(\ell)}\in\mathbb{R}^{N\times N}$ as

$$P_{i,j}^{(\ell)}=\mathbb{E}_{i\in\mathcal{B}^{(\ell)}}\Big[\widetilde{L}_{i,j}^{(\ell)} \mid i\in\mathcal{B}^{(\ell)}\Big] \quad (80)$$

where the expectation is taken over row indices $i$. The above equation implies that under the condition that knowing the $i$th node is in $\mathcal{B}^{(\ell)}$, we have $P_{i,j}^{(\ell)}=\widetilde{L}_{i,j}$, $\forall j=\{1,\dots,N\}$. Let consider the mean-aggregation for the $i$th node as

$$\boldsymbol{x}_{i}^{(\ell)}=\sigma\Big(\sum_{j=1}^{N}\widetilde{L}_{i,j}^{(\ell)}\boldsymbol{x}_{j}^{(\ell-1)}\Big) \quad (81)$$

Then, under the condition $i$th node is in $\mathcal{B}^{(\ell)}$, we can replace $\widetilde{L}_{i,j}^{(\ell)}$ by $P_{i,j}^{(\ell)}$, which gives us

$$\boldsymbol{x}_{i}^{(\ell)}=\sigma\Big(\sum_{j=1}^{N}P_{i,j}^{(\ell)}\boldsymbol{x}_{j}^{(\ell-1)}\Big) \quad (82)$$

As a result, we can write the expectation of $\boldsymbol{x}_{i}^{(\ell)}$ with respect to the indices $i$ as

$$\mathbb{E}_{i\in\mathcal{B}^{(\ell)}}[\boldsymbol{x}_{i}^{(\ell)} \mid i\in\mathcal{B}^{(\ell)}]=\mathbb{E}_{i\in\mathcal{B}^{(\ell)}}\Big[\sigma\Big(\sum_{j=1}^{N}\widetilde{L}_{i,j}^{(\ell)}\boldsymbol{x}_{j}^{(\ell-1)}\Big) \mid i\in\mathcal{B}^{(\ell)}\Big]$$

$$=\mathbb{E}_{i\in\mathcal{B}^{(\ell)}}\Big[\sigma\Big(\sum_{j=1}^{N}P_{i,j}^{(\ell)}\boldsymbol{x}_{j}^{(\ell-1)}\Big) \mid i\in\mathcal{B}^{(\ell)}\Big] \quad (83)$$

$$=\sigma\Big(\sum_{j=1}^{N}P_{i,j}^{(\ell)}\boldsymbol{x}_{j}^{(\ell-1)}\Big)$$

Then define $\bar{\mathbf{H}}^{(\ell)}\in\mathbb{R}^{N\times d_{\ell}}$ as the node embedding of using full-batch but a subset of neighbors for neighbor aggregation, i.e.,

$$\bar{\mathbf{H}}^{(\ell)}=\sigma(\mathbf{P}^{(\ell)}\bar{\mathbf{H}}^{(\ell-1)}\mathbf{W}^{(\ell)}) \quad (84)$$

where all rows in $\bar{\mathbf{H}}^{(\ell)}$ are non-zero vectors.

Using the notations defined above, we can compute $\bar{\mathbf{D}}^{(L+1)} \in \mathbb{R}^{N \times d_L}$, $\bar{\mathbf{G}}^{(\ell)} \in \mathbb{R}^{d_{\ell-1} \times d_\ell}$, and $\bar{\mathbf{D}}^{(\ell)} \in \mathbb{R}^{N \times d_{\ell-1}}$ as

$$\bar{\mathbf{D}}^{(L+1)} = \mathbb{E}\Big[\frac{\partial \mathrm{Loss}(\bar{\mathbf{H}}^{(L)})}{\partial \bar{\mathbf{H}}^{(L)}}\Big] \in \mathbb{R}^{N \times d_L}, \bar{\boldsymbol{d}}_i = \frac{1}{N}\frac{\partial \mathrm{Loss}(\bar{\boldsymbol{h}}_i^{(L)}, y_i)}{\partial \bar{\boldsymbol{h}}_i^{(L)}} \in \mathbb{R}^{d_L} \tag{85}$$

and

$$\bar{\mathbf{D}}^{(\ell)} = \Big[\nabla_H \bar{f}^{(\ell)}(\bar{\mathbf{D}}^{(\ell+1)}, \bar{\mathbf{H}}^{(\ell-1)}, \mathbf{W}^{(\ell)}) := [\mathbf{L}]^\top \Big(\bar{\mathbf{D}}^{(\ell+1)} \circ \sigma'(\mathbf{P}^{(\ell)}\bar{\mathbf{H}}^{(\ell-1)}\mathbf{W}^{(\ell)})\Big)[\mathbf{W}^{(\ell)}]^\top\Big] \tag{86}$$

and

$$\bar{\mathbf{G}}^{(\ell)} = \Big[\nabla_W \bar{f}^{(\ell)}(\bar{\mathbf{D}}^{(\ell+1)}, \bar{\mathbf{H}}^{(\ell-1)}, \mathbf{W}^{(\ell)}) := [\mathbf{L}\bar{\mathbf{H}}^{(\ell-1)}]^\top \Big(\bar{\mathbf{D}}^{(\ell+1)} \circ \sigma'(\mathbf{P}^{(\ell)}\bar{\mathbf{H}}^{(\ell-1)}\mathbf{W}^{(\ell)})\Big)\Big] \tag{87}$$

As a result, we can represent $\bar{\mathbf{G}}^{(\ell)} = \mathbb{E}[\widetilde{\mathbf{G}}^{(\ell)}]$ as

$$\bar{\mathbf{G}}^{(\ell)} = \nabla_W \bar{f}^{(\ell)}(\nabla_H \bar{f}^{(\ell+1)}(\dots \nabla_H \bar{f}^{(L)}(\bar{\mathbf{D}}^{(L+1)}, \bar{\mathbf{H}}^{(L-1)}, \mathbf{W}^{(L)})\dots, \bar{\mathbf{H}}^{(\ell)}, \mathbf{W}^{(\ell+1)}), \bar{\mathbf{H}}^{(\ell-1)}, \mathbf{W}^{(\ell)})) \tag{88}$$

## G.1 SUPPORTING LEMMAS

We derive the upper-bound of the bias and variance of the stochastic gradient in the following lemmas.

**Lemma 2** (Upper-bound on variance). *We can upper-bound the variance of stochastic gradient in* SGCN *as*

$$\sum_{\ell=1}^{L} \mathbb{E}[\|\widetilde{\mathbf{G}}^{(\ell)} - \mathbb{E}[\widetilde{\mathbf{G}}^{(\ell)}]\|_F^2] \leq \sum_{\ell=1}^{L} \mathcal{O}(\mathbb{E}[\|\widetilde{\mathbf{L}}^{(\ell)} - \mathbf{P}^{(\ell)}\|_F^2]) + \mathcal{O}(\|\mathbf{P}^{(\ell)} - \mathbf{L}\|_F^2) \tag{89}$$

*Proof.* By definition, we can write the variance in SGCN as

$$\mathbb{E}[\|\widetilde{\mathbf{G}}^{(\ell)} - \mathbb{E}[\widetilde{\mathbf{G}}^{(\ell)}]\|_F^2]$$
$$= \mathbb{E}[\|\nabla_W \widetilde{f}^{(\ell)}(\nabla_H \widetilde{f}^{(\ell+1)}(\dots \nabla_H \widetilde{f}^{(L)}(\widetilde{\mathbf{D}}^{(L+1)}, \widetilde{\mathbf{H}}^{(L-1)}\mathbf{W}^{(L)})\dots, \widetilde{\mathbf{H}}^{(\ell)}, \mathbf{W}^{(\ell+1)}), \widetilde{\mathbf{H}}^{(\ell-1)}, \mathbf{W}^{(\ell)})$$
$$\quad - \nabla_W \bar{f}^{(\ell)}(\nabla_H \bar{f}^{(\ell+1)}(\dots \nabla_H \bar{f}^{(L)}(\bar{\mathbf{D}}^{(L+1)}, \bar{\mathbf{H}}^{(L-1)}, \mathbf{W}^{(L)})\dots, \bar{\mathbf{H}}^{(\ell)}, \mathbf{W}^{(\ell+1)}), \bar{\mathbf{H}}^{(\ell-1)}), \mathbf{W}^{(\ell)}\|_F^2]$$
$$\leq (L+1)\mathbb{E}[\|\nabla_W \widetilde{f}^{(\ell)}(\nabla_H \widetilde{f}^{(\ell+1)}(\dots \nabla_H \widetilde{f}^{(L)}(\widetilde{\mathbf{D}}^{(L+1)}, \widetilde{\mathbf{H}}^{(L-1)}\mathbf{W}^{(L)})\dots, \widetilde{\mathbf{H}}^{(\ell)}, \mathbf{W}^{(\ell+1)}), \widetilde{\mathbf{H}}^{(\ell-1)}, \mathbf{W}^{(\ell)})$$
$$\quad - \nabla_W \widetilde{f}^{(\ell)}(\nabla_H \widetilde{f}^{(\ell+1)}(\dots \nabla_H \widetilde{f}^{(L)}(\bar{\mathbf{D}}^{(L+1)}, \widetilde{\mathbf{H}}^{(L-1)}\mathbf{W}^{(L)})\dots, \widetilde{\mathbf{H}}^{(\ell)}, \mathbf{W}^{(\ell+1)}), \widetilde{\mathbf{H}}^{(\ell-1)}, \mathbf{W}^{(\ell)})\|_F^2]$$
$$\quad + (L+1)\mathbb{E}[\|\nabla_W \widetilde{f}^{(\ell)}(\nabla_H \widetilde{f}^{(\ell+1)}(\dots \nabla_H \widetilde{f}^{(L)}(\bar{\mathbf{D}}^{(L+1)}, \widetilde{\mathbf{H}}^{(L-1)}\mathbf{W}^{(L)})\dots, \widetilde{\mathbf{H}}^{(\ell)}, \mathbf{W}^{(\ell+1)}), \widetilde{\mathbf{H}}^{(\ell-1)}, \mathbf{W}^{(\ell)})$$
$$\quad - \nabla_W \widetilde{f}^{(\ell)}(\nabla_H \widetilde{f}^{(\ell+1)}(\dots \nabla_H \bar{f}^{(L)}(\bar{\mathbf{D}}^{(L+1)}, \bar{\mathbf{H}}^{(L-1)}\mathbf{W}^{(L)})\dots, \widetilde{\mathbf{H}}^{(\ell)}, \mathbf{W}^{(\ell+1)}), \widetilde{\mathbf{H}}^{(\ell-1)}, \mathbf{W}^{(\ell)})\|_F^2] + \dots$$
$$\quad + (L+1)\mathbb{E}[\|\nabla_W \widetilde{f}^{(\ell)}(\nabla_H \bar{f}^{(\ell+1)}(\bar{\mathbf{D}}^{(\ell+2)}, \widetilde{\mathbf{H}}^{(\ell)}, \mathbf{W}^{(\ell+1)}), \widetilde{\mathbf{H}}^{(\ell-1)}, \mathbf{W}^{(\ell)})$$
$$\quad - \nabla_W \bar{f}^{(\ell)}(\nabla_H \bar{f}^{(\ell+1)}(\bar{\mathbf{D}}^{(\ell+2)}, \bar{\mathbf{H}}^{(\ell)}, \mathbf{W}^{(\ell+1)}), \widetilde{\mathbf{H}}^{(\ell-1)}, \mathbf{W}^{(\ell)})\|_F^2]$$
$$\quad + (L+1)\mathbb{E}[\|\nabla_W \widetilde{f}^{(\ell)}(\bar{\mathbf{D}}^{(\ell+1)}, \widetilde{\mathbf{H}}^{(\ell-1)}, \mathbf{W}^{(\ell)}) - \nabla_W \bar{f}^{(\ell)}(\bar{\mathbf{D}}^{(\ell+1)}, \bar{\mathbf{H}}^{(\ell-1)}, \mathbf{W}^{(\ell)})\|_F^2]$$
$$\leq (L+1)L_W^2 L_H^{2(L-\ell-1)}\mathbb{E}[\|\widetilde{\mathbf{D}}^{(L+1)} - \bar{\mathbf{D}}^{(L+1)}\|_F^2]$$
$$\quad + (L+1)L_W^2 L_H^{2(L-\ell-2)}\mathbb{E}[\|\nabla_H \widetilde{f}^{(L)}(\bar{\mathbf{D}}^{(L+1)}, \widetilde{\mathbf{H}}^{(L-1)}\mathbf{W}^{(L)}) - \nabla_H \bar{f}^{(L)}(\bar{\mathbf{D}}^{(L+1)}, \bar{\mathbf{H}}^{(L-1)}\mathbf{W}^{(L)})\|_F^2] + \dots$$
$$\quad + (L+1)L_W^2 \mathbb{E}[\|\nabla_H \widetilde{f}^{(\ell+1)}(\bar{\mathbf{D}}^{(\ell+2)}, \widetilde{\mathbf{H}}^{(\ell)}, \mathbf{W}^{(\ell+1)}) - \nabla_H \bar{f}^{(\ell+1)}(\bar{\mathbf{D}}^{(\ell+2)}, \bar{\mathbf{H}}^{(\ell)}, \mathbf{W}^{(\ell+1)})\|_F^2]$$
$$\quad + (L+1)\mathbb{E}[\|\nabla_W \widetilde{f}^{(\ell)}(\bar{\mathbf{D}}^{(\ell+1)}, \widetilde{\mathbf{H}}^{(\ell-1)}, \mathbf{W}^{(\ell)}) - \nabla_W \bar{f}^{(\ell)}(\bar{\mathbf{D}}^{(\ell+1)}, \bar{\mathbf{H}}^{(\ell-1)}, \mathbf{W}^{(\ell)})\|_F^2] \tag{90}$$

From the previous equation, we know that there are three key factors that will affect the variance:

- The difference of gradient with respect to the last layer node representations

$$\mathbb{E}[\|\widetilde{\mathbf{D}}^{(L+1)} - \bar{\mathbf{D}}^{(L+1)}\|_F^2] \tag{91}$$

- The difference of gradient with respect to the input node embedding matrix at each graph convolutional layer

$$\mathbb{E}[\|\nabla_H \widetilde{f}^{(\ell+1)}(\bar{\mathbf{D}}^{(\ell+2)}, \widetilde{\mathbf{H}}^{(\ell)}, \mathbf{W}^{(\ell+1)}) - \nabla_H \bar{f}^{(\ell+1)}(\bar{\mathbf{D}}^{(\ell+2)}, \bar{\mathbf{H}}^{(\ell)}, \mathbf{W}^{(\ell+1)})\|_{\mathrm{F}}^2] \quad (92)$$

- The difference of gradient with respect to the weight matrix at each graph convolutional layer

$$\mathbb{E}[\|\nabla_W \widetilde{f}^{(\ell)}(\bar{\mathbf{D}}^{(\ell+1)}, \widetilde{\mathbf{H}}^{(\ell-1)}, \mathbf{W}^{(\ell)}) - \nabla_W \bar{f}^{(\ell)}(\bar{\mathbf{D}}^{(\ell+1)}, \bar{\mathbf{H}}^{(\ell-1)}, \mathbf{W}^{(\ell)})\|_{\mathrm{F}}^2] \quad (93)$$

First, Let consider the upper-bound of Eq. 91.

$$
\begin{aligned}
\mathbb{E}[\|\widetilde{\mathbf{D}}^{(L+1)} - \bar{\mathbf{D}}^{(L+1)}\|_{\mathrm{F}}^2] &= \mathbb{E}[\|\frac{\partial \mathrm{Loss}(\widetilde{\mathbf{H}}^{(L)}, \boldsymbol{y})}{\partial \widetilde{\mathbf{H}}^{(L)}} - \frac{\partial \mathrm{Loss}(\bar{\mathbf{H}}^{(L)}, \boldsymbol{y})}{\partial \bar{\mathbf{H}}^{(L)}}\|_{\mathrm{F}}^2] \\
&\leq L_{loss}^2 \mathbb{E}[\|\widetilde{\mathbf{H}}^{(L)} - \bar{\mathbf{H}}^{(L)}\|_{\mathrm{F}}^2] \\
&\leq L_{loss}^2 \mathbb{E}[\|\sigma(\widetilde{\mathbf{L}}^{(L)}\widetilde{\mathbf{H}}^{(L-1)}\mathbf{W}^{(L)}) - \sigma(\mathbf{P}^{(\ell)}\bar{\mathbf{H}}^{(L-1)}\mathbf{W}^{(L)})\|_{\mathrm{F}}^2] \\
&\leq L_{loss}^2 C_\sigma^2 B_W^2 \mathbb{E}[\|\widetilde{\mathbf{L}}^{(L)}\widetilde{\mathbf{H}}^{(L-1)} - \mathbf{P}^{(L)}\bar{\mathbf{H}}^{(L-1)}\|_{\mathrm{F}}^2] \\
&\leq L_{loss}^2 C_\sigma^2 B_W^2 B_H^2 \mathbb{E}[\|\widetilde{\mathbf{L}}^{(L)} - \mathbf{P}^{(L)}\|_{\mathrm{F}}^2]
\end{aligned}
\quad (94)
$$

Then, let consider the upper-bound of Eq. 92.

$$
\begin{aligned}
&\mathbb{E}[\|\nabla_H \widetilde{f}^{(\ell)}(\bar{\mathbf{D}}^{(\ell+1)}, \widetilde{\mathbf{H}}^{(\ell-1)}, \mathbf{W}^{(\ell)}) - \nabla_H \bar{f}^{(\ell)}(\bar{\mathbf{D}}^{(\ell+1)}, \bar{\mathbf{H}}^{(\ell-1)}, \mathbf{W}^{(\ell)})\|_{\mathrm{F}}^2] \\
&= \mathbb{E}[\|[\widetilde{\mathbf{L}}^{(\ell)}]^\top \Big(\bar{\mathbf{D}}^{(\ell+1)} \circ \sigma'(\widetilde{\mathbf{L}}^{(\ell)}\widetilde{\mathbf{H}}^{(\ell-1)}\mathbf{W}^{(\ell)})\Big)[\mathbf{W}^{(\ell)}]^\top - [\mathbf{L}]^\top \Big(\bar{\mathbf{D}}^{(\ell+1)} \circ \sigma'(\mathbf{P}^{(\ell)}\bar{\mathbf{H}}^{(\ell-1)}\mathbf{W}^{(\ell)})\Big)[\mathbf{W}^{(\ell)}]^\top\|_{\mathrm{F}}^2] \\
&\leq 2\mathbb{E}[\|[\widetilde{\mathbf{L}}^{(\ell)}]^\top \Big(\bar{\mathbf{D}}^{(\ell+1)} \circ \sigma'(\widetilde{\mathbf{L}}^{(\ell)}\widetilde{\mathbf{H}}^{(\ell-1)}\mathbf{W}^{(\ell)})\Big)[\mathbf{W}^{(\ell)}]^\top - [\widetilde{\mathbf{L}}^{(\ell)}]^\top \Big(\bar{\mathbf{D}}^{(\ell+1)} \circ \sigma'(\mathbf{P}^{(\ell)}\bar{\mathbf{H}}^{(\ell-1)}\mathbf{W}^{(\ell)})\Big)[\mathbf{W}^{(\ell)}]^\top\|_{\mathrm{F}}^2] \\
&\quad + 2\mathbb{E}[\|[\widetilde{\mathbf{L}}^{(\ell)}]^\top \Big(\bar{\mathbf{D}}^{(\ell+1)} \circ \sigma'(\mathbf{P}^{(\ell)}\bar{\mathbf{H}}^{(\ell-1)}\mathbf{W}^{(\ell)})\Big)[\mathbf{W}^{(\ell)}]^\top - [\mathbf{L}]^\top \Big(\bar{\mathbf{D}}^{(\ell+1)} \circ \sigma'(\mathbf{P}^{(\ell)}\bar{\mathbf{H}}^{(\ell-1)}\mathbf{W}^{(\ell)})\Big)[\mathbf{W}^{(\ell)}]^\top\|_{\mathrm{F}}^2] \\
&\leq 2B_{LA}^2 B_D^2 B_W^4 L_\sigma^2 \mathbb{E}[\|\widetilde{\mathbf{L}}^{(\ell)}\widetilde{\mathbf{H}}^{(\ell-1)} - \mathbf{P}^{(\ell)}\bar{\mathbf{H}}^{(\ell-1)}\|_{\mathrm{F}}^2] + 2B_D^2 C_\sigma^2 B_W^2 \mathbb{E}[\|\widetilde{\mathbf{L}}^{(\ell)} - \mathbf{L}\|_{\mathrm{F}}^2] \\
&\leq 2B_{LA}^2 B_D^2 B_H^2 B_W^4 L_\sigma^2 \mathbb{E}[\|\widetilde{\mathbf{L}}^{(\ell)} - \mathbf{P}^{(\ell)}\|_{\mathrm{F}}^2] + 2B_D^2 C_\sigma^2 B_W^2 \mathbb{E}[\|\widetilde{\mathbf{L}}^{(\ell)} - \mathbf{P}^{(\ell)} + \mathbf{P}^{(\ell)} - \mathbf{L}\|_{\mathrm{F}}^2] \\
&\leq 2\Big(B_{LA}^2 B_D^2 B_H^2 B_W^4 L_\sigma^2 + 2B_D^2 C_\sigma^2 B_W^2\Big)\mathbb{E}[\|\widetilde{\mathbf{L}}^{(\ell)} - \mathbf{P}^{(\ell)}\|_{\mathrm{F}}^2] + 4B_D^2 C_\sigma^2 B_W^2 \mathbb{E}[\|\mathbf{P}^{(\ell)} - \mathbf{L}\|_{\mathrm{F}}^2] \\
&\leq \mathcal{O}(\mathbb{E}[\|\widetilde{\mathbf{L}}^{(\ell)} - \mathbf{P}^{(\ell)}\|_{\mathrm{F}}^2]) + \mathcal{O}(\mathbb{E}[\|\mathbf{P}^{(\ell)} - \mathbf{L}\|_{\mathrm{F}}^2])
\end{aligned}
$$

$$(95)$$

Finally, let consider the upper-bound of Eq. 93.

$$
\begin{aligned}
&\mathbb{E}[\|\nabla_W \widetilde{f}^{(\ell)}(\bar{\mathbf{D}}^{(\ell+1)}, \widetilde{\mathbf{H}}^{(\ell-1)}, \mathbf{W}^{(\ell)}) - \nabla_W \bar{f}^{(\ell)}(\bar{\mathbf{D}}^{(\ell+1)}, \bar{\mathbf{H}}^{(\ell-1)}, \mathbf{W}^{(\ell)})\|_{\mathrm{F}}^2] \\
&\leq \mathbb{E}[\|[\widetilde{\mathbf{L}}^{(\ell)}\widetilde{\mathbf{H}}^{(\ell-1)}]^\top \Big(\bar{\mathbf{D}}^{(\ell+1)} \circ \sigma'(\widetilde{\mathbf{L}}^{(\ell)}\widetilde{\mathbf{H}}^{(\ell-1)}\mathbf{W}^{(\ell)})\Big) - [\mathbf{L}\bar{\mathbf{H}}^{(\ell-1)}]^\top \Big(\bar{\mathbf{D}}^{(\ell+1)} \circ \sigma'(\mathbf{P}^{(\ell)}\bar{\mathbf{H}}^{(\ell-1)}\mathbf{W}^{(\ell)})\Big)\|_{\mathrm{F}}^2] \\
&\leq 2\mathbb{E}[\|[\widetilde{\mathbf{L}}^{(\ell)}\widetilde{\mathbf{H}}^{(\ell-1)}]^\top \Big(\bar{\mathbf{D}}^{(\ell+1)} \circ \sigma'(\widetilde{\mathbf{L}}^{(\ell)}\widetilde{\mathbf{H}}^{(\ell-1)}\mathbf{W}^{(\ell)})\Big) - [\widetilde{\mathbf{L}}^{(\ell)}\widetilde{\mathbf{H}}^{(\ell-1)}]^\top \Big(\bar{\mathbf{D}}^{(\ell+1)} \circ \sigma'(\mathbf{P}^{(\ell)}\bar{\mathbf{H}}^{(\ell-1)}\mathbf{W}^{(\ell)})\Big)\|_{\mathrm{F}}^2] \\
&\quad + 2\mathbb{E}[\|[\widetilde{\mathbf{L}}^{(\ell)}\widetilde{\mathbf{H}}^{(\ell-1)}]^\top \Big(\bar{\mathbf{D}}^{(\ell+1)} \circ \sigma'(\mathbf{P}^{(\ell)}\bar{\mathbf{H}}^{(\ell-1)}\mathbf{W}^{(\ell)})\Big) - [\mathbf{L}\bar{\mathbf{H}}^{(\ell-1)}]^\top \Big(\bar{\mathbf{D}}^{(\ell+1)} \circ \sigma'(\mathbf{P}^{(\ell)}\bar{\mathbf{H}}^{(\ell-1)}\mathbf{W}^{(\ell)})\Big)\|_{\mathrm{F}}^2] \\
&\leq 2B_{LA}^2 B_H^2 B_D^2 B_W^2 L_\sigma^2 \mathbb{E}[\|\widetilde{\mathbf{L}}^{(\ell)}\widetilde{\mathbf{H}}^{(\ell-1)} - \mathbf{P}^{(\ell)}\bar{\mathbf{H}}^{(\ell-1)}\|_{\mathrm{F}}^2] + 2B_D^2 C_\sigma^2 \mathbb{E}[\|\widetilde{\mathbf{L}}^{(\ell)}\widetilde{\mathbf{H}}^{(\ell-1)} - \mathbf{L}\bar{\mathbf{H}}^{(\ell-1)}\|_{\mathrm{F}}^2] \\
&\leq 2\Big(B_{LA}^2 B_H^2 B_D^2 B_W^2 L_\sigma^2 + B_D^2 C_\sigma^2\Big)\mathbb{E}[\|\widetilde{\mathbf{L}}^{(\ell)}\widetilde{\mathbf{H}}^{(\ell-1)} - \mathbf{P}^{(\ell)}\bar{\mathbf{H}}^{(\ell-1)}\|_{\mathrm{F}}^2] \\
&\quad + 2B_D^2 B_H^2 C_\sigma^2 \mathbb{E}[\|\mathbf{P}^{(\ell)} - \mathbf{L}\|_{\mathrm{F}}^2] \\
&\leq 2\Big(B_{LA}^2 B_H^4 B_D^2 B_W^2 L_\sigma^2 + B_H^2 B_D^2 C_\sigma^2\Big)\mathbb{E}[\|\widetilde{\mathbf{L}}^{(\ell)} - \mathbf{P}^{(\ell)}\|_{\mathrm{F}}^2] \\
&\quad + 2B_D^2 B_H^2 C_\sigma^2 \mathbb{E}[\|\mathbf{P}^{(\ell)} - \mathbf{L}\|_{\mathrm{F}}^2] \\
&\leq \mathcal{O}(\mathbb{E}[\|\widetilde{\mathbf{L}}^{(\ell)} - \mathbf{P}^{(\ell)}\|_{\mathrm{F}}^2]) + \mathcal{O}(\mathbb{E}[\|\mathbf{P}^{(\ell)} - \mathbf{L}\|_{\mathrm{F}}^2])
\end{aligned}
$$

$$(96)$$

Combining the result from Eq. 91, 92, 93 we have

$$
\begin{aligned}
\mathbb{E}[\|\widetilde{\mathbf{G}}^{(\ell)} - \mathbb{E}[\widetilde{\mathbf{G}}^{(\ell)}]\|_{\mathrm{F}}^2] \leq{}& \mathcal{O}(\mathbb{E}[\|\widetilde{\mathbf{L}}^{(\ell)} - \mathbf{P}^{(\ell)}\|_{\mathrm{F}}^2]) + \ldots + \mathcal{O}(\mathbb{E}[\|\widetilde{\mathbf{L}}^{(L)} - \mathbf{P}^{(L)}\|_{\mathrm{F}}^2]) \\
& + \mathcal{O}(\mathbb{E}[\|\mathbf{P}^{(\ell)} - \mathbf{L}\|_{\mathrm{F}}^2]) + \ldots + \mathcal{O}(\mathbb{E}[\|\mathbf{P}^{(L)} - \mathbf{L}\|_{\mathrm{F}}^2])
\end{aligned}
\tag{97}
$$

$\square$

**Lemma 3** (Upper-bound on bias). *We can upper-bound the bias of stochastic gradient in* SGCN *as*

$$
\sum_{\ell=1}^{L} \mathbb{E}[\|\mathbb{E}[\widetilde{\mathbf{G}}^{(\ell)}] - \mathbf{G}^{(\ell)}\|_{\mathrm{F}}^2] \leq \sum_{\ell=1}^{L} \mathcal{O}(\|\mathbf{P}^{(\ell)} - \mathbf{L}\|_{\mathrm{F}}^2)
\tag{98}
$$

*Proof.* By definition, we can write the bias of stochastic gradient in SGCN as

$$
\begin{aligned}
&\mathbb{E}[\|\mathbb{E}[\widetilde{\mathbf{G}}^{(\ell)}] - \mathbf{G}^{(\ell)}\|_{\mathrm{F}}^2] \\
&\leq \mathbb{E}[\|\nabla_W \bar{f}^{(\ell)}(\nabla_H \bar{f}^{(\ell+1)}(\ldots \nabla_H \bar{f}^{(L)}(\bar{\mathbf{D}}^{(L+1)}, \bar{\mathbf{H}}^{(L-1)}, \mathbf{W}^{(L)}) \ldots, \bar{\mathbf{H}}^{(\ell)}, \mathbf{W}^{(\ell+1)}), \bar{\mathbf{H}}^{(\ell-1)}, \mathbf{W}^{(\ell)}) \\
&\quad - \nabla_W f^{(\ell)}(\nabla_H f^{(\ell+1)}(\ldots \nabla_H f^{(L)}(\mathbf{D}^{(L+1)}, \mathbf{H}^{(L-1)}, \mathbf{W}^{(L)}) \ldots, \mathbf{H}^{(\ell)}, \mathbf{W}^{(\ell+1)}), \mathbf{H}^{(\ell-1)}, \mathbf{W}^{(\ell)})\|_{\mathrm{F}}^2] \\
&\leq (L+1)\mathbb{E}[\|\nabla_W \bar{f}^{(\ell)}(\nabla_H \bar{f}^{(\ell+1)}(\ldots \nabla_H \bar{f}^{(L)}(\bar{\mathbf{D}}^{(L+1)}, \bar{\mathbf{H}}^{(L-1)}, \mathbf{W}^{(L)}) \ldots, \bar{\mathbf{H}}^{(\ell)}, \mathbf{W}^{(\ell+1)}), \bar{\mathbf{H}}^{(\ell-1)}, \mathbf{W}^{(\ell)}) \\
&\quad - \nabla_W \bar{f}^{(\ell)}(\nabla_H \bar{f}^{(\ell+1)}(\ldots \nabla_H \bar{f}^{(L)}(\mathbf{D}^{(L+1)}, \bar{\mathbf{H}}^{(L-1)}, \mathbf{W}^{(L)}) \ldots, \bar{\mathbf{H}}^{(\ell)}, \mathbf{W}^{(\ell+1)}), \bar{\mathbf{H}}^{(\ell-1)}, \mathbf{W}^{(\ell)})\|_{\mathrm{F}}^2] \\
&\quad + (L+1)\mathbb{E}[\|\nabla_W \bar{f}^{(\ell)}(\nabla_H \bar{f}^{(\ell+1)}(\ldots \nabla_H \bar{f}^{(L)}(\mathbf{D}^{(L+1)}, \bar{\mathbf{H}}^{(L-1)}, \mathbf{W}^{(L)}) \ldots, \bar{\mathbf{H}}^{(\ell)}, \mathbf{W}^{(\ell+1)}), \bar{\mathbf{H}}^{(\ell-1)}, \mathbf{W}^{(\ell)}) \\
&\quad - \nabla_W \bar{f}^{(\ell)}(\nabla_H \bar{f}^{(\ell+1)}(\ldots \nabla_H f^{(L)}(\mathbf{D}^{(L+1)}, \mathbf{H}^{(L-1)}, \mathbf{W}^{(L)}) \ldots, \bar{\mathbf{H}}^{(\ell)}, \mathbf{W}^{(\ell+1)}), \bar{\mathbf{H}}^{(\ell-1)}, \mathbf{W}^{(\ell)})\|_{\mathrm{F}}^2] + \ldots \\
&\quad + (L+1)\mathbb{E}[\|\nabla_W \bar{f}^{(\ell)}(\nabla_H \bar{f}^{(\ell+1)}(\mathbf{D}^{(\ell+2)}, \bar{\mathbf{H}}^{(\ell)}, \mathbf{W}^{(\ell+1)}), \bar{\mathbf{H}}^{(\ell-1)}, \mathbf{W}^{(\ell)}) \\
&\quad - \nabla_W \bar{f}^{(\ell)}(\nabla_H f^{(\ell+1)}(\mathbf{D}^{(\ell+2)}, \mathbf{H}^{(\ell)}, \mathbf{W}^{(\ell+1)}), \bar{\mathbf{H}}^{(\ell-1)}, \mathbf{W}^{(\ell)})\|_{\mathrm{F}}^2] \\
&\quad + (L+1)\mathbb{E}[\|\nabla_W \bar{f}^{(\ell)}(\mathbf{D}^{(\ell+1)}, \bar{\mathbf{H}}^{(\ell-1)}, \mathbf{W}^{(\ell)}) - \nabla_W f^{(\ell)}(\mathbf{D}^{(\ell+1)}, \mathbf{H}^{(\ell-1)}, \mathbf{W}^{(\ell)})\|_{\mathrm{F}}^2] \\
&\leq (L+1)L_W^2 L_H^{2(L-\ell-1)}\mathbb{E}[\|\bar{\mathbf{D}}^{(L+1)} - \mathbf{D}^{(L+1)}\|_{\mathrm{F}}^2] \\
&\quad + (L+1)L_W^2 L_H^{2(L-\ell-2)}\mathbb{E}[\|\nabla_H \bar{f}^{(L)}(\mathbf{D}^{(L+1)}, \bar{\mathbf{H}}^{(L-1)}, \mathbf{W}^{(L)}) - \nabla_H f^{(L)}(\mathbf{D}^{(L+1)}, \mathbf{H}^{(L-1)}, \mathbf{W}^{(L)})\|_{\mathrm{F}}^2] + \ldots \\
&\quad + (L+1)L_W^2\mathbb{E}[\|\nabla_H \bar{f}^{(\ell+1)}(\mathbf{D}^{(\ell+2)}, \bar{\mathbf{H}}^{(\ell)}, \mathbf{W}^{(\ell+1)}) - \nabla_H f^{(\ell+1)}(\mathbf{D}^{(\ell+2)}, \mathbf{H}^{(\ell)}, \mathbf{W}^{(\ell+1)})\|_{\mathrm{F}}^2] \\
&\quad + (L+1)\mathbb{E}[\|\nabla_W \bar{f}^{(\ell)}(\mathbf{D}^{(\ell+1)}, \bar{\mathbf{H}}^{(\ell-1)}, \mathbf{W}^{(\ell)}) - \nabla_W f^{(\ell)}(\mathbf{D}^{(\ell+1)}, \mathbf{H}^{(\ell-1)}, \mathbf{W}^{(\ell)})\|_{\mathrm{F}}^2]
\end{aligned}
\tag{99}
$$

From the previous equation, we know that there are three key factors that will affect the bias:

- The difference of gradient with respect to the last layer node representations

$$
\mathbb{E}[\|\bar{\mathbf{D}}^{(L+1)} - \mathbf{D}^{(L+1)}\|_{\mathrm{F}}^2]
\tag{100}
$$

- The difference of gradient with respect to the input node embedding matrix at each graph convolutional layer

$$
\mathbb{E}[\|\nabla_H \bar{f}^{(\ell+1)}(\mathbf{D}^{(\ell+2)}, \bar{\mathbf{H}}^{(\ell)}, \mathbf{W}^{(\ell+1)}) - \nabla_H f^{(\ell+1)}(\mathbf{D}^{(\ell+2)}, \mathbf{H}^{(\ell)}, \mathbf{W}^{(\ell+1)})\|_{\mathrm{F}}^2]
\tag{101}
$$

- The difference of gradient with respect to the weight matrix at each graph convolutional layer

$$
\mathbb{E}[\|\nabla_W \bar{f}^{(\ell)}(\mathbf{D}^{(\ell+1)}, \bar{\mathbf{H}}^{(\ell-1)}, \mathbf{W}^{(\ell)}) - \nabla_W f^{(\ell)}(\mathbf{D}^{(\ell+1)}, \mathbf{H}^{(\ell-1)}, \mathbf{W}^{(\ell)})\|_{\mathrm{F}}^2]
\tag{102}
$$

Firstly, let consider the upper-bound of Eq. 100.

$$
\begin{aligned}
\mathbb{E}[\|\bar{\mathbf{D}}^{(L+1)} - \mathbf{D}^{(L+1)}\|_{\mathrm{F}}^2] &= \mathbb{E}[\|\frac{\partial \mathrm{Loss}(\bar{\mathbf{H}}^{(L)}, \boldsymbol{y})}{\partial \bar{\mathbf{H}}^{(L)}} - \frac{\partial \mathrm{Loss}(\mathbf{H}^{(L)}, \boldsymbol{y})}{\partial \mathbf{H}^{(L)}}\|_{\mathrm{F}}^2] \\
&\leq L_{loss}^2 \mathbb{E}[\|\bar{\mathbf{H}}^{(L)} - \mathbf{H}^{(L)}\|_{\mathrm{F}}^2]
\end{aligned}
\tag{103}
$$

The upper-bound for $\mathbb{E}[\|\bar{\mathbf{H}}^{(\ell)} - \mathbf{H}^{(\ell)}\|_{\mathrm{F}}^2]$ as

$$
\begin{aligned}
\|\bar{\mathbf{H}}^{(\ell)} - \mathbf{H}^{(\ell)}\|_{\mathrm{F}}^2 &= \|\sigma(\mathbf{P}^{(\ell)}\bar{\mathbf{H}}^{(\ell-1)}\mathbf{W}^{(\ell)}) - \sigma(\mathbf{L}\mathbf{H}^{(\ell-1)}\mathbf{W}^{(\ell)})\|_{\mathrm{F}}^2 \\
&\leq C_\sigma^2 B_W^2 \|\mathbf{P}^{(\ell)}\bar{\mathbf{H}}^{(\ell-1)} - \mathbf{L}\bar{\mathbf{H}}^{(\ell-1)} + \mathbf{L}\bar{\mathbf{H}}^{(\ell-1)} - \mathbf{L}\mathbf{H}^{(\ell-1)}\|_{\mathrm{F}}^2 \\
&\leq 2C_\sigma^2 B_W^2 B_H^2 \|\mathbf{P}^{(\ell)} - \mathbf{L}\|_{\mathrm{F}}^2 + 2C_\sigma^2 B_W^2 B_{LA}^2 \|\bar{\mathbf{H}}^{(\ell-1)} - \mathbf{H}^{(\ell-1)}\|_{\mathrm{F}}^2 \\
&\leq \mathcal{O}(\|\mathbf{P}^{(1)} - \mathbf{L}\|_{\mathrm{F}}^2) + \ldots + \mathcal{O}(\|\mathbf{P}^{(\ell)} - \mathbf{L}\|_{\mathrm{F}}^2)
\end{aligned}
\tag{104}
$$

Therefore, we have

$$
\mathbb{E}[\|\bar{\mathbf{D}}^{(L+1)} - \mathbf{D}^{(L+1)}\|_{\mathrm{F}}^2] \leq \mathcal{O}(\|\mathbf{P}^{(1)} - \mathbf{L}\|_{\mathrm{F}}^2) + \ldots + \mathcal{O}(\|\mathbf{P}^{(L)} - \mathbf{L}\|_{\mathrm{F}}^2)
\tag{105}
$$

Then, let consider the upper-bound of Eq. 101.

$$
\begin{aligned}
&\mathbb{E}[\|\nabla_H \bar{f}^{(\ell)}(\mathbf{D}^{(\ell+1)}, \bar{\mathbf{H}}^{(\ell-1)}, \mathbf{W}^{(\ell)}) - \nabla_H f^{(\ell)}(\mathbf{D}^{(\ell+1)}, \mathbf{H}^{(\ell-1)}, \mathbf{W}^{(\ell)})\|_{\mathrm{F}}^2] \\
&= \mathbb{E}[\|[\mathbf{L}]^\top\Big(\mathbf{D}^{(\ell+1)} \circ \sigma'(\mathbf{P}^{(\ell)}\bar{\mathbf{H}}^{(\ell-1)}\mathbf{W}^{(\ell)})\Big)[\mathbf{W}^{(\ell)}]^\top - [\mathbf{L}]^\top\Big(\mathbf{D}^{(\ell+1)} \circ \sigma'(\mathbf{L}\mathbf{H}^{(\ell-1)}\mathbf{W}^{(\ell)})\Big)[\mathbf{W}^{(\ell)}]^\top\|_{\mathrm{F}}^2] \\
&\leq B_{LA}^2 B_D^2 B_W^4 L_\sigma^2 \mathbb{E}[\|\mathbf{P}^{(\ell)}\bar{\mathbf{H}}^{(\ell-1)} - \mathbf{L}\bar{\mathbf{H}}^{(\ell-1)} + \mathbf{L}\bar{\mathbf{H}}^{(\ell-1)} - \mathbf{L}\mathbf{H}^{(\ell-1)}\|_{\mathrm{F}}^2] \\
&\leq 2B_{LA}^2 B_D^2 B_W^4 L_\sigma^2 B_H^2 \mathbb{E}[\|\mathbf{P}^{(\ell)} - \mathbf{L}\|_{\mathrm{F}}^2] + 2B_{LA}^4 B_D^2 B_W^4 L_\sigma^2 \mathbb{E}[\|\bar{\mathbf{H}}^{(\ell-1)} - \mathbf{H}^{(\ell-1)}\|_{\mathrm{F}}^2] \\
&\leq \mathcal{O}(\|\mathbf{P}^{(1)} - \mathbf{L}\|_{\mathrm{F}}^2) + \ldots + \mathcal{O}(\|\mathbf{P}^{(\ell)} - \mathbf{L}\|_{\mathrm{F}}^2)
\end{aligned}
\tag{106}
$$

Finally, let consider the upper-bound of Eq. 102.

$$
\begin{aligned}
&\mathbb{E}[\|\nabla_W \bar{f}^{(\ell)}(\mathbf{D}^{(\ell+1)}, \bar{\mathbf{H}}^{(\ell-1)}, \mathbf{W}^{(\ell)}) - \nabla_W f^{(\ell)}(\mathbf{D}^{(\ell+1)}, \mathbf{H}^{(\ell-1)}, \mathbf{W}^{(\ell)})\|_{\mathrm{F}}^2] \\
&= \mathbb{E}[\|[\mathbf{L}\bar{\mathbf{H}}^{(\ell-1)}]^\top\Big(\mathbf{D}^{(\ell+1)} \circ \sigma'(\mathbf{P}^{(\ell)}\bar{\mathbf{H}}^{(\ell-1)}\mathbf{W}^{(\ell)})\Big) - [\mathbf{L}\mathbf{H}^{(\ell-1)}]^\top\Big(\mathbf{D}^{(\ell+1)} \circ \sigma'(\mathbf{L}^{(\ell)}\mathbf{H}^{(\ell-1)}\mathbf{W}^{(\ell)})\Big)\|_{\mathrm{F}}^2] \\
&\leq 2\mathbb{E}[\|[\mathbf{L}\bar{\mathbf{H}}^{(\ell-1)}]^\top\Big(\mathbf{D}^{(\ell+1)} \circ \sigma'(\mathbf{P}^{(\ell)}\bar{\mathbf{H}}^{(\ell-1)}\mathbf{W}^{(\ell)})\Big) - [\mathbf{L}\mathbf{H}^{(\ell-1)}]^\top\Big(\mathbf{D}^{(\ell+1)} \circ \sigma'(\mathbf{P}^{(\ell)}\bar{\mathbf{H}}^{(\ell-1)}\mathbf{W}^{(\ell)})\Big)\|_{\mathrm{F}}^2] \\
&\quad + 2\mathbb{E}[\|[\mathbf{L}\mathbf{H}^{(\ell-1)}]^\top\Big(\mathbf{D}^{(\ell+1)} \circ \sigma'(\mathbf{P}^{(\ell)}\bar{\mathbf{H}}^{(\ell-1)}\mathbf{W}^{(\ell)})\Big) - [\mathbf{L}\mathbf{H}^{(\ell-1)}]^\top\Big(\mathbf{D}^{(\ell+1)} \circ \sigma'(\mathbf{L}^{(\ell)}\mathbf{H}^{(\ell-1)}\mathbf{W}^{(\ell)})\Big)\|_{\mathrm{F}}^2] \\
&\leq 2B_D^2 C_\sigma^2 B_{LA}^2 \mathbb{E}[\|\bar{\mathbf{H}}^{(\ell-1)} - \mathbf{H}^{(\ell-1)}\|_{\mathrm{F}}^2] \\
&\quad + 2B_{LA}^2 B_H^2 B_D^2 L_\sigma^2 B_W^2 \mathbb{E}[\|\mathbf{P}^{(\ell)}\bar{\mathbf{H}}^{(\ell-1)} - \mathbf{L}\bar{\mathbf{H}}^{(\ell-1)} + \mathbf{L}\bar{\mathbf{H}}^{(\ell-1)} - \mathbf{L}\mathbf{H}^{(\ell-1)}\|_{\mathrm{F}}^2] \\
&\leq 2\Big(B_D^2 C_\sigma^2 B_{LA}^2 + B_{LA}^4 B_H^2 B_D^2 L_\sigma^2 B_W^2\Big)\mathbb{E}[\|\bar{\mathbf{H}}^{(\ell-1)} - \mathbf{H}^{(\ell-1)}\|_{\mathrm{F}}^2] \\
&\quad + 2B_{LA}^2 B_H^4 B_D^2 L_\sigma^2 B_W^2 \mathbb{E}[\|\mathbf{P}^{(\ell)} - \mathbf{L}\|_{\mathrm{F}}^2] \\
&\leq \mathcal{O}(\|\mathbf{P}^{(1)} - \mathbf{L}\|_{\mathrm{F}}^2) + \ldots + \mathcal{O}(\|\mathbf{P}^{(\ell)} - \mathbf{L}\|_{\mathrm{F}}^2)
\end{aligned}
\tag{107}
$$

Combining the result from Eq. 100, 101, 102 we have

$$
\mathbb{E}[\|\mathbb{E}[\widetilde{\mathbf{G}}^{(\ell)}] - \mathbf{G}^{(\ell)}\|_{\mathrm{F}}^2] \leq \mathcal{O}(\mathbb{E}[\|\mathbf{P}^{(1)} - \mathbf{L}\|_{\mathrm{F}}^2]) + \ldots + \mathcal{O}(\mathbb{E}[\|\mathbf{P}^{(L)} - \mathbf{L}\|_{\mathrm{F}}^2])
\tag{108}
$$

$\square$

## G.2 REMAINING STEPS TOWARD THEOREM 1

By the smoothness of $\mathcal{L}(\boldsymbol{\theta}_t)$, we have

$$
\begin{aligned}
\mathcal{L}(\boldsymbol{\theta}_{t+1}) &\leq \mathcal{L}(\boldsymbol{\theta}_t) + \langle\nabla\mathcal{L}(\boldsymbol{\theta}_t), \boldsymbol{\theta}_{t+1} - \boldsymbol{\theta}_t\rangle + \frac{L_f}{2}\|\boldsymbol{\theta}_{t+1} - \boldsymbol{\theta}_t\|_{\mathrm{F}}^2 \\
&= \mathcal{L}(\boldsymbol{\theta}_t) - \eta\langle\nabla\mathcal{L}(\boldsymbol{\theta}_t), \nabla\widetilde{\mathcal{L}}(\boldsymbol{\theta}_t)\rangle + \frac{\eta^2 L_f}{2}\|\nabla\widetilde{\mathcal{L}}(\boldsymbol{\theta}_t)\|_{\mathrm{F}}^2
\end{aligned}
\tag{109}
$$

Let $\mathcal{F}_t = \{\{\mathcal{B}_1^{(\ell)}\}_{\ell=1}^L, \ldots, \{\mathcal{B}_{t-1}^{(\ell)}\}_{\ell=1}^L\}$. Note that the weight parameters $\boldsymbol{\theta}_t$ is a function of history of the generated random process and hence is random. Taking expectation on both sides condition

on $\mathcal{F}_t$ and using $\eta < 1/L_f$ we have

$\mathbb{E}[\mathcal{L}(\boldsymbol{\theta}_{t+1})|\mathcal{F}_t]$

$$\leq \mathcal{L}(\boldsymbol{\theta}_t) - \eta\langle\nabla\mathcal{L}(\boldsymbol{\theta}_t), \mathbb{E}[\nabla\widetilde{\mathcal{L}}(\boldsymbol{\theta}_t)|\mathcal{F}_t]\rangle + \frac{\eta^2 L_f}{2}\Big(\mathbb{E}[\|\nabla\widetilde{\mathcal{L}}(\boldsymbol{\theta}_t) - \mathbb{E}[\nabla\widetilde{\mathcal{L}}(\boldsymbol{\theta}_t)|\mathcal{F}_t]\|_{\mathrm{F}}^2|\mathcal{F}_t] + \mathbb{E}[\|\mathbb{E}[\nabla\widetilde{\mathcal{L}}(\boldsymbol{\theta}_t)|\mathcal{F}_t]\|_{\mathrm{F}}^2|\mathcal{F}_t]\Big)$$

$$= \mathcal{L}(\boldsymbol{\theta}_t) - \eta\langle\nabla\mathcal{L}(\boldsymbol{\theta}_t), \nabla\mathcal{L}(\boldsymbol{\theta}_t) + \mathbb{E}[\mathbf{b}_t|\mathcal{F}_t]\rangle + \frac{\eta^2 L_f}{2}\Big(\mathbb{E}[\|\mathbf{n}_t\|_{\mathrm{F}}^2|\mathcal{F}_t] + \|\nabla\mathcal{L}(\boldsymbol{\theta}_t) + \mathbb{E}[\mathbf{b}_t|\mathcal{F}_t]\|_{\mathrm{F}}^2\Big)$$

$$\leq \mathcal{L}(\boldsymbol{\theta}_t) + \frac{\eta}{2}\Big(-2\langle\nabla\mathcal{L}(\boldsymbol{\theta}_t), \nabla\mathcal{L}(\boldsymbol{\theta}_t) + \mathbb{E}[\mathbf{b}_t|\mathcal{F}_t]\rangle + \|\nabla\mathcal{L}(\boldsymbol{\theta}_t) + \mathbb{E}[\mathbf{b}_t|\mathcal{F}_t]\|_{\mathrm{F}}^2\Big) + \frac{\eta^2 L_f}{2}\mathbb{E}[\|\mathbf{n}_t\|_{\mathrm{F}}^2|\mathcal{F}_t]$$

$$= \mathcal{L}(\boldsymbol{\theta}_t) + \frac{\eta}{2}\Big(-\|\nabla\mathcal{L}(\boldsymbol{\theta}_t)\|_{\mathrm{F}}^2 + \mathbb{E}[\|\mathbf{b}_t\|_{\mathrm{F}}^2|\mathcal{F}_t]\Big) + \frac{\eta^2 L_f}{2}\mathbb{E}[\|\mathbf{n}_t\|_{\mathrm{F}}^2|\mathcal{F}_t] \tag{110}$$

Denote $\Delta_{\mathbf{b}}$ as the upper bound of bias of stochasitc gradient as shown in Lemma 3 and Denote $\Delta_{\mathbf{n}}$ as the upper bound of bias of stochastic gradient as shown in Lemma 2. Plugging in the upper bound of bias and variance, taking expectation over $\mathcal{F}_t$, and rearranging the term we have

$$\mathbb{E}[\|\nabla\mathcal{L}(\boldsymbol{\theta}_t)\|_{\mathrm{F}}^2] \leq \frac{2}{\eta}\Big(\mathbb{E}[\mathcal{L}(\boldsymbol{\theta}_t)] - \mathbb{E}[\mathcal{L}(\boldsymbol{\theta}_{t+1})]\Big) + \eta L_f\Delta_{\mathbf{n}} + \Delta_{\mathbf{b}} \tag{111}$$

Summing up from $t = 1$ to $T$, rearranging we have

$$\frac{1}{T}\sum_{t=1}^{T}\mathbb{E}[\|\nabla\mathcal{L}(\boldsymbol{\theta}_t)\|_{\mathrm{F}}^2] \leq \frac{2}{\eta T}\sum_{t=1}^{T}(\mathbb{E}[\mathcal{L}(\boldsymbol{\theta}_t)] - \mathbb{E}[\mathcal{L}(\boldsymbol{\theta}_{t+1})]) + \eta L_f\Delta_{\mathbf{n}} + \Delta_{\mathbf{b}}$$

$$\underset{(a)}{\leq} \frac{2}{\eta T}(\mathcal{L}(\boldsymbol{\theta}_1) - \mathcal{L}(\boldsymbol{\theta}^\star)) + \eta L_f\Delta_{\mathbf{n}} + \Delta_{\mathbf{b}} \tag{112}$$

where the inequality $(a)$ is due to $\mathcal{L}(\boldsymbol{\theta}^\star) \leq \mathbb{E}[\mathcal{L}(\boldsymbol{\theta}_{T+1})]$.

By selecting learning rate as $\eta = 1/\sqrt{T}$, we have

$$\frac{1}{T}\sum_{t=1}^{T}\mathbb{E}[\|\nabla\mathcal{L}(\boldsymbol{\theta}_t)\|_{\mathrm{F}}^2] \leq \frac{2(\mathcal{L}(\boldsymbol{\theta}_1) - \mathcal{L}(\boldsymbol{\theta}^\star))}{\sqrt{T}} + \frac{L_f\Delta_{\mathbf{n}}}{\sqrt{T}} + \Delta_{\mathbf{b}} \tag{113}$$

# H   PROOF OF THEOREM 2

## H.1   SUPPORTING LEMMAS

In the following lemma, we derive the upper-bound on the node embedding approximation error of each GCN layer in SGCN+. This upper-bound plays an important role in the analysis of the upper-bound of the bias term for the stochastic gradient. Suppose the input node embedding matrix for the $\ell$ GCN layer as $\bar{\mathbf{H}}_t^{(\ell-1)}$, the forward propagation for the $\ell$th layer in SGCN+ is defined as

$$\widetilde{f}^{(\ell)}(\bar{\mathbf{H}}_t^{(\ell-1)}, \mathbf{W}^{(\ell)}) = \left[ \bar{\mathbf{Z}}_t^{(\ell)} := \mathbf{P}^{(\ell)} \bar{\mathbf{H}}_t^{(\ell)} \mathbf{W}^{(\ell)} \right] \tag{114}$$

and the forward propagation for the $\ell$th layer in FullGCN is defined as

$$f^{(\ell)}(\bar{\mathbf{H}}_t^{(\ell-1)}, \mathbf{W}^{(\ell)}) = \mathbf{L} \bar{\mathbf{H}}_t^{(\ell-1)} \mathbf{W}^{(\ell)} \tag{115}$$

In the following, we derive the upper-bound of

$$\mathbb{E}[\| \widetilde{f}^{(\ell)}(\bar{\mathbf{H}}_t^{(\ell-1)}, \mathbf{W}^{(\ell)}) - f^{(\ell)}(\bar{\mathbf{H}}_t^{(\ell-1)}, \mathbf{W}^{(\ell)}) \|_{\mathrm{F}}^2] = \mathbb{E}[\| \mathbf{L} \bar{\mathbf{H}}_t^{(\ell-1)} \mathbf{W}_t^{(\ell)} - \bar{\mathbf{Z}}_t^{(\ell)} \|_{\mathrm{F}}^2] \tag{116}$$

**Lemma 4.** *Let denote $E \in [0, T/K - 1]$ as the current epoch, let $t$ be the current step. Therefore, for any $t \in \{EK + 1, \ldots, EK + K\}$, we have*

$$\begin{aligned} &\mathbb{E}[\| \mathbf{L} \bar{\mathbf{H}}_t^{(\ell-1)} \mathbf{W}_t^{(\ell)} - \bar{\mathbf{Z}}_t^{(\ell)} \|_{\mathrm{F}}^2] \\ &\leq \eta^2 K \times \mathcal{O}\left( \left| \mathbb{E}[\| \mathbf{P}^{(\ell)} \|_{\mathrm{F}}^2] - \| \mathbf{L} \|_{\mathrm{F}}^2 \right| + \ldots + \left| \mathbb{E}[\| \mathbf{P}^{(1)} \|_{\mathrm{F}}^2] - \| \mathbf{L} \|_{\mathrm{F}}^2 \right| \right) \end{aligned} \tag{117}$$

*Proof.*

$$\begin{aligned} &\| \mathbf{L} \bar{\mathbf{H}}_t^{(\ell-1)} \mathbf{W}_t^{(\ell)} - \bar{\mathbf{Z}}_t^{(\ell)} \|_{\mathrm{F}}^2 \\ &= \| [\mathbf{L} \bar{\mathbf{H}}_t^{(\ell-1)} \mathbf{W}_t^{(\ell)} - \mathbf{L} \bar{\mathbf{H}}_{t-1}^{(\ell-1)} \mathbf{W}_{t-1}^{(\ell)}] + [\mathbf{L} \bar{\mathbf{H}}_{t-1}^{(\ell-1)} \mathbf{W}_{t-1}^{(\ell)} - \bar{\mathbf{Z}}_{t-1}^{(\ell)}] - [\bar{\mathbf{Z}}_t^{(\ell)} - \bar{\mathbf{Z}}_{t-1}^{(\ell)}] \|_{\mathrm{F}}^2 \\ &= \| \mathbf{L} \bar{\mathbf{H}}_t^{(\ell-1)} \mathbf{W}_t^{(\ell)} - \mathbf{L} \bar{\mathbf{H}}_{t-1}^{(\ell-1)} \mathbf{W}_{t-1}^{(\ell)} \|_{\mathrm{F}}^2 + \| \mathbf{L} \bar{\mathbf{H}}_{t-1}^{(\ell-1)} \mathbf{W}_{t-1}^{(\ell)} - \bar{\mathbf{Z}}_{t-1}^{(\ell)} \|_{\mathrm{F}}^2 + \| \bar{\mathbf{Z}}_t^{(\ell)} - \bar{\mathbf{Z}}_{t-1}^{(\ell)} \|_{\mathrm{F}}^2 \\ &\quad + 2 \langle \mathbf{L} \bar{\mathbf{H}}_t^{(\ell-1)} \mathbf{W}_t^{(\ell)} - \mathbf{L} \bar{\mathbf{H}}_{t-1}^{(\ell-1)} \mathbf{W}_{t-1}^{(\ell)}, \mathbf{L} \bar{\mathbf{H}}_{t-1}^{(\ell-1)} \mathbf{W}_{t-1}^{(\ell)} - \bar{\mathbf{Z}}_{t-1}^{(\ell)} \rangle \\ &\quad - 2 \langle \mathbf{L} \bar{\mathbf{H}}_t^{(\ell-1)} \mathbf{W}_t^{(\ell)} - \mathbf{L} \bar{\mathbf{H}}_{t-1}^{(\ell-1)} \mathbf{W}_{t-1}^{(\ell)}, \bar{\mathbf{Z}}_t^{(\ell)} - \bar{\mathbf{Z}}_{t-1}^{(\ell)} \rangle \\ &\quad - 2 \langle \mathbf{L} \bar{\mathbf{H}}_{t-1}^{(\ell-1)} \mathbf{W}_{t-1}^{(\ell)} - \bar{\mathbf{Z}}_{t-1}^{(\ell)}, \bar{\mathbf{Z}}_t^{(\ell)} - \bar{\mathbf{Z}}_{t-1}^{(\ell)} \rangle \end{aligned} \tag{118}$$

Recall that by the update rule, we have

$$\bar{\mathbf{Z}}_t^{(\ell)} - \bar{\mathbf{Z}}_{t-1}^{(\ell)} = \mathbf{P}^{(\ell)} \bar{\mathbf{H}}_t^{(\ell-1)} \mathbf{W}_t^{(\ell)} - \mathbf{P}^{(\ell)} \bar{\mathbf{H}}_{t-1}^{(\ell-1)} \mathbf{W}_{t-1}^{(\ell)} \tag{119}$$

and

$$\mathbb{E}[\bar{\mathbf{Z}}_t^{(\ell)} - \bar{\mathbf{Z}}_{t-1}^{(\ell)} | \mathcal{F}_t] = \mathbf{L} \bar{\mathbf{H}}_t^{(\ell-1)} \mathbf{W}_t^{(\ell)} - \mathbf{L} \bar{\mathbf{H}}_{t-1}^{(\ell-1)} \mathbf{W}_{t-1}^{(\ell)} \tag{120}$$

Taking expectation on both side condition on $\mathcal{F}_t$, we have

$$\begin{aligned} \mathbb{E}[\| \mathbf{L} \bar{\mathbf{H}}_t^{(\ell-1)} \mathbf{W}_t^{(\ell)} - \bar{\mathbf{Z}}_t^{(\ell)} \|_{\mathrm{F}}^2 | \mathcal{F}_t] &\leq \| \mathbf{L} \bar{\mathbf{H}}_{t-1}^{(\ell-1)} \mathbf{W}_{t-1}^{(\ell)} - \bar{\mathbf{Z}}_{t-1}^{(\ell)} \|_{\mathrm{F}}^2 + \mathbb{E}[\| \bar{\mathbf{Z}}_t^{(\ell)} - \bar{\mathbf{Z}}_{t-1}^{(\ell)} \|_{\mathrm{F}}^2 | \mathcal{F}_t] \\ &\quad - \| \mathbf{L} \bar{\mathbf{H}}_t^{(\ell-1)} \mathbf{W}_t^{(\ell)} - \mathbf{L} \bar{\mathbf{H}}_{t-1}^{(\ell-1)} \mathbf{W}_{t-1}^{(\ell)} \|_{\mathrm{F}}^2 \end{aligned} \tag{121}$$

Then take the expectation over $\mathcal{F}_t$, we have

$$\begin{aligned} \mathbb{E}[\| \mathbf{L} \bar{\mathbf{H}}_t^{(\ell-1)} \mathbf{W}_t^{(\ell)} - \bar{\mathbf{Z}}_t^{(\ell)} \|_{\mathrm{F}}^2] &\leq \| \mathbf{L} \bar{\mathbf{H}}_{t-1}^{(\ell-1)} \mathbf{W}_{t-1}^{(\ell)} - \bar{\mathbf{Z}}_{t-1}^{(\ell)} \|_{\mathrm{F}}^2 + \mathbb{E}[\| \bar{\mathbf{Z}}_t^{(\ell)} - \bar{\mathbf{Z}}_{t-1}^{(\ell)} \|_{\mathrm{F}}^2] \\ &\quad - \| \mathbf{L} \bar{\mathbf{H}}_t^{(\ell-1)} \mathbf{W}_t^{(\ell)} - \mathbf{L} \bar{\mathbf{H}}_{t-1}^{(\ell-1)} \mathbf{W}_{t-1}^{(\ell)} \|_{\mathrm{F}}^2 \end{aligned} \tag{122}$$

Since we know $t \in \{EK+1, \dots, EK+K\}$, we can denote $t = EK+k$, $k \le K$ such that

$$
\begin{aligned}
\mathbb{E}[\|\mathbf{L}\bar{\mathbf{H}}_t^{(\ell-1)}\mathbf{W}_t^{(\ell)} - \bar{\mathbf{Z}}_t^{(\ell)}\|_\mathrm{F}^2] &= \mathbb{E}[\|\mathbf{L}\bar{\mathbf{H}}_{EK+k}^{(\ell-1)}\mathbf{W}_{EK+k}^{(\ell)} - \bar{\mathbf{Z}}_{EK+k}^{(\ell)}\|_\mathrm{F}^2] \\
&= \underbrace{\mathbb{E}[\|\mathbf{L}\bar{\mathbf{H}}_{EK}^{(\ell-1)}\mathbf{W}_{EK}^{(\ell)} - \bar{\mathbf{Z}}_{EK}^{(\ell)}\|_\mathrm{F}^2]}_{(A)} \\
&\quad + \sum_{t=EK+1}^{EK+K}\left(\mathbb{E}[\|\bar{\mathbf{Z}}_t^{(\ell)} - \bar{\mathbf{Z}}_{t-1}^{(\ell)}\|_\mathrm{F}^2] - \|\mathbf{L}\bar{\mathbf{H}}_t^{(\ell-1)}\mathbf{W}_t^{(\ell)} - \mathbf{L}\bar{\mathbf{H}}_{t-1}^{(\ell-1)}\mathbf{W}_{t-1}^{(\ell)}\|_\mathrm{F}^2\right)
\end{aligned}
\tag{123}
$$

Knowing that we are using all neighbors at the snapshot step $(t \mod K) = 0$, we have $(A) = 0$. As a result, we have

$$
\begin{aligned}
&\mathbb{E}[\|\mathbf{L}\bar{\mathbf{H}}_{EK+k}^{(\ell-1)}\mathbf{W}_{EK+k}^{(\ell)} - \bar{\mathbf{Z}}_{EK+k}^{(\ell)}\|_\mathrm{F}^2] \\
&\le \sum_{t=EK+1}^{EK+K}\left(\underbrace{\mathbb{E}[\|\bar{\mathbf{Z}}_t^{(\ell)} - \bar{\mathbf{Z}}_{t-1}^{(\ell)}\|_\mathrm{F}^2] - \|\mathbf{L}\bar{\mathbf{H}}_t^{(\ell-1)}\mathbf{W}_t^{(\ell)} - \mathbf{L}\bar{\mathbf{H}}_{t-1}^{(\ell-1)}\mathbf{W}_{t-1}^{(\ell)}\|_\mathrm{F}^2}_{(B)}\right)
\end{aligned}
\tag{124}
$$

Let take a closer look at term $(B)$.

$$
\begin{aligned}
&\mathbb{E}[\|\bar{\mathbf{Z}}_t^{(\ell)} - \bar{\mathbf{Z}}_{t-1}^{(\ell)}\|_\mathrm{F}^2] - \|\mathbf{L}\bar{\mathbf{H}}_t^{(\ell-1)}\mathbf{W}_t^{(\ell)} - \mathbf{L}\bar{\mathbf{H}}_{t-1}^{(\ell-1)}\mathbf{W}_{t-1}^{(\ell)}\|_\mathrm{F}^2 \\
&= \mathbb{E}[\|\mathbf{P}^{(\ell)}\bar{\mathbf{H}}_t^{(\ell-1)}\mathbf{W}_t^{(\ell)} - \mathbf{P}^{(\ell)}\bar{\mathbf{H}}_{t-1}^{(\ell-1)}\mathbf{W}_{t-1}^{(\ell)}\|_\mathrm{F}^2] - \|\mathbf{L}\bar{\mathbf{H}}_t^{(\ell-1)}\mathbf{W}_t^{(\ell)} - \mathbf{L}\bar{\mathbf{H}}_{t-1}^{(\ell-1)}\mathbf{W}_{t-1}^{(\ell)}\|_\mathrm{F}^2 \\
&\le \mathbb{E}[\|\mathbf{P}^{(\ell)}(\bar{\mathbf{H}}_t^{(\ell-1)}\mathbf{W}_t^{(\ell)} - \bar{\mathbf{H}}_{t-1}^{(\ell-1)}\mathbf{W}_{t-1}^{(\ell)})\|_\mathrm{F}^2] - \|\mathbf{L}(\bar{\mathbf{H}}_t^{(\ell-1)}\mathbf{W}_t^{(\ell)} - \bar{\mathbf{H}}_{t-1}^{(\ell-1)}\mathbf{W}_{t-1}^{(\ell)})\|_\mathrm{F}^2 \\
&\le \left(\mathbb{E}[\|\mathbf{P}^{(\ell)}\|_\mathrm{F}^2] - \|\mathbf{L}\|_\mathrm{F}^2\right)\underbrace{\mathbb{E}[\|\bar{\mathbf{H}}_t^{(\ell-1)}\mathbf{W}_t^{(\ell)} - \bar{\mathbf{H}}_{t-1}^{(\ell-1)}\mathbf{W}_{t-1}^{(\ell)}\|_\mathrm{F}^2]}_{(C)}
\end{aligned}
\tag{125}
$$

Let take a closer look at term $(C)$.

$$
\begin{aligned}
&\mathbb{E}[\|\bar{\mathbf{H}}_t^{(\ell-1)}\mathbf{W}_t^{(\ell)} - \bar{\mathbf{H}}_{t-1}^{(\ell-1)}\mathbf{W}_{t-1}^{(\ell)}\|_\mathrm{F}^2] \\
&= \mathbb{E}[\|\bar{\mathbf{H}}_t^{(\ell-1)}\mathbf{W}_t^{(\ell)} - \bar{\mathbf{H}}_t^{(\ell-1)}\mathbf{W}_{t-1}^{(\ell)} + \bar{\mathbf{H}}_t^{(\ell-1)}\mathbf{W}_{t-1}^{(\ell)} - \bar{\mathbf{H}}_{t-1}^{(\ell-1)}\mathbf{W}_{t-1}^{(\ell)}\|_\mathrm{F}^2] \\
&\le 2B_H^2\mathbb{E}[\|\mathbf{W}_t^{(\ell)} - \mathbf{W}_{t-1}^{(\ell)}\|_\mathrm{F}^2] + 2\mathbb{E}[\|\bar{\mathbf{H}}_t^{(\ell-1)}\mathbf{W}_{t-1}^{(\ell)} - \bar{\mathbf{H}}_{t-1}^{(\ell-1)}\mathbf{W}_{t-1}^{(\ell)}\|_\mathrm{F}^2]
\end{aligned}
\tag{126}
$$

By induction, we have

$$
\begin{aligned}
&\mathbb{E}[\|\bar{\mathbf{H}}_t^{(\ell-1)}\mathbf{W}_t^{(\ell)} - \bar{\mathbf{H}}_{t-1}^{(\ell-1)}\mathbf{W}_{t-1}^{(\ell)}\|_\mathrm{F}^2] \\
&\le 2B_H^2\mathbb{E}[\|\mathbf{W}_t^{(\ell)} - \mathbf{W}_{t-1}^{(\ell)}\|_\mathrm{F}^2] + 2^2 B_H^4\mathbb{E}[\|\mathbf{W}_t^{(\ell-1)} - \mathbf{W}_{t-1}^{(\ell-1)}\|_\mathrm{F}^2] + \dots \\
&\quad + 2^\ell B_H^{2\ell}\mathbb{E}[\|\mathbf{W}_t^{(1)} - \mathbf{W}_{t-1}^{(1)}\|_\mathrm{F}^2]
\end{aligned}
\tag{127}
$$

By the update rule of weight matrices, we know

$$
\mathbb{E}[\|\mathbf{W}_t^{(\ell)} - \mathbf{W}_{t-1}^{(\ell)}\|_\mathrm{F}^2] = \eta^2\mathbb{E}[\|\bar{\mathbf{G}}_{t-1}^{(\ell)}\|_\mathrm{F}^2]
\tag{128}
$$

Therefore, we have

$$
\begin{aligned}
&\mathbb{E}[\|\mathbf{L}\bar{\mathbf{H}}_{EK+k}^{(\ell-1)}\mathbf{W}_{EK+k}^{(\ell)} - \bar{\mathbf{Z}}_{EK+k}^{(\ell)}\|_\mathrm{F}^2] \\
&\le \sum_{t=EK+1}^{EK+K}\eta^2\mathcal{O}\left(|\mathbb{E}[\|\mathbf{P}^{(\ell)}\|_\mathrm{F}^2] - \|\mathbf{L}\|_\mathrm{F}^2| \times \mathbb{E}[\|\bar{\mathbf{G}}_{t-1}^{(\ell)}\|_\mathrm{F}^2] + \dots + |\mathbb{E}[\|\mathbf{P}^{(1)}\|_\mathrm{F}^2] - \|\mathbf{L}\|_\mathrm{F}^2| \times \mathbb{E}[\|\bar{\mathbf{G}}_{t-1}^{(1)}\|_\mathrm{F}^2]\right)
\end{aligned}
\tag{129}
$$

By the definition of $\bar{\mathbf{G}}_{t-1}^{(\ell)}$, we have that

$$
\mathbb{E}[\|\bar{\mathbf{G}}_{t-1}^{(\ell)}\|_\mathrm{F}^2] \le B_{LA}^2 B_H^2 B_D^2 C_\sigma^2
\tag{130}
$$

Plugging it back, we have

$$
\begin{aligned}
&\mathbb{E}[\|\mathbf{L}\bar{\mathbf{H}}_t^{(\ell-1)}\mathbf{W}_t^{(\ell)} - \bar{\mathbf{Z}}_t^{(\ell)}\|_{\mathrm{F}}^2] \\
&:= \mathbb{E}[\|\mathbf{L}\bar{\mathbf{H}}_{EK+k}^{(\ell-1)}\mathbf{W}_{EK+k}^{(\ell)} - \bar{\mathbf{Z}}_{EK+k}^{(\ell)}\|_{\mathrm{F}}^2] \\
&\leq \eta^2 K \times \mathcal{O}\Big(\big|\mathbb{E}[\|\mathbf{P}^{(\ell)}\|_{\mathrm{F}}^2] - \|\mathbf{L}\|_{\mathrm{F}}^2\big| + \ldots + \big|\mathbb{E}[\|\mathbf{P}^{(1)}\|_{\mathrm{F}}^2] - \|\mathbf{L}\|_{\mathrm{F}}^2\big|\big)\Big)
\end{aligned}
\tag{131}
$$

$\square$

Based on the upper-bound of node embedding approximation error of each graph convolutional layer, we derived the upper-bound on the bias of stochastic gradient in SGCN.

**Lemma 5** (Upper-bound on bias). *We can upper-bound the bias of stochastic gradient in* SGCN+ *as*

$$
\sum_{\ell=1}^{L} \mathbb{E}[\|\mathbb{E}[\widetilde{\mathbf{G}}^{(\ell)}] - \mathbf{G}^{(\ell)}\|_{\mathrm{F}}^2] \leq \eta^2 K \sum_{\ell=1}^{L} \mathcal{O}\Big(\big|\mathbb{E}[\|\mathbf{P}^{(\ell)}\|_{\mathrm{F}}^2] - \|\mathbf{L}\|_{\mathrm{F}}^2\big|\Big)
\tag{132}
$$

*Proof.* From the decomposition of bias as shown in previously in Eq. 108, we have

$$
\begin{aligned}
&\mathbb{E}[\|\mathbb{E}[\widetilde{\mathbf{G}}^{(\ell)}] - \mathbf{G}^{(\ell)}\|_{\mathrm{F}}^2] \\
&\leq (L+1)L_W^2 L_H^{2(L-\ell-1)}\mathbb{E}[\|\mathbf{D}^{(\bar{L}+1)} - \mathbf{D}^{(L+1)}\|_{\mathrm{F}}^2] \\
&\quad + (L+1)L_W^2 L_H^{2(L-\ell-2)}\mathbb{E}[\|\nabla_H \bar{f}^{(L)}(\mathbf{D}^{(L+1)},\bar{\mathbf{H}}^{(L-1)},\mathbf{W}^{(L)}) - \nabla_H f^{(L)}(\mathbf{D}^{(L+1)},\mathbf{H}^{(L-1)},\mathbf{W}^{(L)})\|_{\mathrm{F}}^2] + \ldots \\
&\quad + (L+1)L_W^2 \mathbb{E}[\|\nabla_H \bar{f}^{(\ell+1)}(\mathbf{D}^{(\ell+2)},\bar{\mathbf{H}}^{(\ell)},\mathbf{W}^{(\ell+1)}) - \nabla_H f^{(\ell+1)}(\mathbf{D}^{(\ell+2)},\mathbf{H}^{(\ell)},\mathbf{W}^{(\ell+1)})\|_{\mathrm{F}}^2] \\
&\quad + (L+1)\mathbb{E}[\|\nabla_W \bar{f}^{(\ell)}(\mathbf{D}^{(\ell+1)},\bar{\mathbf{H}}^{(\ell-1)},\mathbf{W}^{(\ell)}) - \nabla_W f^{(\ell)}(\mathbf{D}^{(\ell+1)},\mathbf{H}^{(\ell-1)},\mathbf{W}^{(\ell)})\|_{\mathrm{F}}^2]
\end{aligned}
\tag{133}
$$

From the previous equation, we know that there are three key factors that will affect the bias:

- The difference of gradient with respect to the last layer node representations

$$
\mathbb{E}[\|\bar{\mathbf{D}}^{(L+1)} - \mathbf{D}^{(L+1)}\|_{\mathrm{F}}^2]
\tag{134}
$$

- The difference of gradient with respect to the input node embedding matrix at each graph convolutional layer

$$
\mathbb{E}[\|\nabla_H \bar{f}^{(\ell+1)}(\mathbf{D}^{(\ell+2)},\bar{\mathbf{H}}^{(\ell)},\mathbf{W}^{(\ell+1)}) - \nabla_H f^{(\ell+1)}(\mathbf{D}^{(\ell+2)},\mathbf{H}^{(\ell)},\mathbf{W}^{(\ell+1)})\|_{\mathrm{F}}^2]
\tag{135}
$$

- The difference of gradient with respect to the weight matrix at each graph convolutional layer

$$
\mathbb{E}[\|\nabla_W \bar{f}^{(\ell)}(\mathbf{D}^{(\ell+1)},\bar{\mathbf{H}}^{(\ell-1)},\mathbf{W}^{(\ell)}) - \nabla_W f^{(\ell)}(\mathbf{D}^{(\ell+1)},\mathbf{H}^{(\ell-1)},\mathbf{W}^{(\ell)})\|_{\mathrm{F}}^2]
\tag{136}
$$

Firstly, let consider the upper-bound of Eq. 134.

$$
\begin{aligned}
\mathbb{E}[\|\bar{\mathbf{D}}^{(L+1)} - \mathbf{D}^{(L+1)}\|_{\mathrm{F}}^2] &= \mathbb{E}[\|\frac{\partial \mathrm{Loss}(\bar{\mathbf{H}}^{(L)},\boldsymbol{y})}{\partial \bar{\mathbf{H}}^{(L)}} - \frac{\partial \mathrm{Loss}(\mathbf{H}^{(L)},\boldsymbol{y})}{\partial \mathbf{H}^{(L)}}\|_{\mathrm{F}}^2] \\
&\leq L_{loss}^2 \mathbb{E}[\|\bar{\mathbf{H}}^{(L)} - \mathbf{H}^{(L)}\|_{\mathrm{F}}^2] \\
&\leq L_{loss}^2 C_\sigma^2 \mathbb{E}[\|\bar{\mathbf{Z}}^{(L)} - \mathbf{Z}^{(L)}\|_{\mathrm{F}}^2]
\end{aligned}
\tag{137}
$$

We can decompose $\mathbb{E}[\|\bar{\mathbf{Z}}^{(L)} - \mathbf{Z}^{(L)}\|_{\mathrm{F}}^2]$ as

$$
\begin{aligned}
&\mathbb{E}[\|\bar{\mathbf{Z}}^{(L)} - \mathbf{Z}^{(L)}\|_{\mathrm{F}}^2] \\
&= \mathbb{E}[\|\bar{\mathbf{Z}}^{(L)} - \mathbf{L}\mathbf{H}^{(L-1)}\mathbf{W}^{(L)}\|_{\mathrm{F}}^2] \\
&\leq 2\mathbb{E}[\|\bar{\mathbf{Z}}^{(L)} - \mathbf{L}\bar{\mathbf{H}}^{(L-1)}\mathbf{W}^{(L)}\|_{\mathrm{F}}^2] + 2\mathbb{E}[\|\mathbf{L}\bar{\mathbf{H}}^{(L-1)}\mathbf{W}^{(L)} - \mathbf{L}\mathbf{H}^{(L-1)}\mathbf{W}^{(L)}\|_{\mathrm{F}}^2] \\
&\leq 2\mathbb{E}[\|\bar{\mathbf{Z}}^{(L)} - \mathbf{L}\bar{\mathbf{H}}^{(L-1)}\mathbf{W}^{(L)}\|_{\mathrm{F}}^2] + 2B_{LA}^2 B_W^2 C_\sigma^2 \mathbb{E}[\|\bar{\mathbf{Z}}^{(L-1)} - \mathbf{Z}^{(L-1)}\|_{\mathrm{F}}^2] \\
&\leq \sum_{\ell=1}^{L} \mathcal{O}(\mathbb{E}[\|\bar{\mathbf{Z}}^{(\ell)} - \mathbf{L}\bar{\mathbf{H}}^{(\ell-1)}\mathbf{W}^{(\ell)}\|_{\mathrm{F}}^2])
\end{aligned}
\tag{138}
$$

Using result from Lemma 4, we have

$$\mathbb{E}[\|\bar{\mathbf{D}}^{(L+1)} - \mathbf{D}^{(L+1)}\|_F^2] \leq \sum_{\ell=1}^{L} \eta^2 K \times \mathcal{O}\Big(|\mathbb{E}[\|\mathbf{P}^{(\ell)}\|_F^2] - \|\mathbf{L}\|_F^2|\Big) \tag{139}$$

Then, let consider the upper-bound of Eq. 135.

$$\mathbb{E}[\|\nabla_H \bar{f}^{(\ell)}(\mathbf{D}^{(\ell+1)}, \bar{\mathbf{H}}^{(\ell-1)}, \mathbf{W}^{(\ell)}) - \nabla_H f^{(\ell)}(\mathbf{D}^{(\ell+1)}, \mathbf{H}^{(\ell-1)}, \mathbf{W}^{(\ell)})\|_F^2]$$

$$= \mathbb{E}[\|[\mathbf{L}]^\top\Big(\mathbf{D}^{(\ell+1)} \circ \sigma'(\mathbf{P}^{(\ell)}\bar{\mathbf{H}}^{(\ell-1)}\mathbf{W}^{(\ell)})\Big)[\mathbf{W}^{(\ell)}]^\top - [\mathbf{L}]^\top\Big(\mathbf{D}^{(\ell+1)} \circ \sigma'(\mathbf{L}\mathbf{H}^{(\ell-1)}\mathbf{W}^{(\ell)})\Big)[\mathbf{W}^{(\ell)}]^\top\|_F^2]$$

$$\leq B_{LA}^2 B_D^2 B_W^2 L_\sigma^2 \mathbb{E}[\|\bar{\mathbf{Z}}_t^{(\ell)} - \mathbf{L}\mathbf{H}^{(\ell-1)}\mathbf{W}^{(\ell)}\|_F^2]$$

$$\leq 2B_{LA}^2 B_D^2 B_W^2 L_\sigma^2 \mathbb{E}[\|\bar{\mathbf{Z}}_t^{(\ell)} - \mathbf{L}\bar{\mathbf{H}}^{(\ell-1)}\mathbf{W}^{(\ell)}\|_F^2]$$

$$+ 2B_{LA}^2 B_D^2 B_W^2 L_\sigma^2 \underbrace{\mathbb{E}[\|\mathbf{L}\bar{\mathbf{H}}^{(\ell-1)}\mathbf{W}^{(\ell)} - \mathbf{L}\mathbf{H}^{(\ell-1)}\mathbf{W}^{(\ell)}\|_F^2]}_{(A)} \tag{140}$$

where $\bar{\mathbf{Z}}_t^{(\ell)} = \bar{\mathbf{Z}}_{t-1}^{(\ell)} + \mathbf{P}^{(\ell)}\bar{\mathbf{H}}_t^{(\ell-1)}\mathbf{W}_t^{(\ell)} - \mathbf{P}^{(\ell)}\bar{\mathbf{H}}_{t-1}^{(\ell-1)}\mathbf{W}_{t-1}^{(\ell)}$.

Let take a closer look at term $(A)$, we have

$$\mathbb{E}[\|\mathbf{L}\bar{\mathbf{H}}^{(\ell-1)}\mathbf{W}^{(\ell)} - \mathbf{L}\mathbf{H}^{(\ell-1)}\mathbf{W}^{(\ell)}\|_F^2] \leq B_{LA}^2 B_W^2 C_\sigma^2 \mathbb{E}[\|\bar{\mathbf{Z}}^{(\ell-2)} - \mathbf{L}\mathbf{H}^{(\ell-2)}\mathbf{W}^{(\ell-1)}\|_F^2]$$

$$\leq 2B_{LA}^2 B_W^2 C_\sigma^2 \mathbb{E}[\|\bar{\mathbf{Z}}^{(\ell-2)} - \mathbf{L}\bar{\mathbf{H}}^{(\ell-2)}\mathbf{W}^{(\ell-1)}\|_F^2]$$

$$+ 2B_{LA}^2 B_W^2 C_\sigma^2 \mathbb{E}[\|\mathbf{L}\bar{\mathbf{H}}^{(\ell-2)}\mathbf{W}^{(\ell-1)} - \mathbf{L}\mathbf{H}^{(\ell-2)}\mathbf{W}^{(\ell-1)}\|_F^2] \tag{141}$$

Therefore, by induction we have

$$\mathbb{E}[\|\nabla_H \bar{f}^{(\ell)}(\mathbf{D}^{(\ell+1)}, \bar{\mathbf{H}}^{(\ell-1)}, \mathbf{W}^{(\ell)}) - \nabla_H f^{(\ell)}(\mathbf{D}^{(\ell+1)}, \mathbf{H}^{(\ell-1)}, \mathbf{W}^{(\ell)})\|_F^2]$$

$$\leq \mathcal{O}(\mathbb{E}[\|\bar{\mathbf{Z}}^{(\ell)} - \mathbf{L}\bar{\mathbf{H}}^{(\ell-1)}\mathbf{W}^{(\ell)}\|_F^2]) + \mathcal{O}(\mathbb{E}[\|\bar{\mathbf{Z}}^{(\ell-1)} - \mathbf{L}\bar{\mathbf{H}}^{(\ell-2)}\mathbf{W}^{(\ell-1)}\|_F^2]) + \dots \tag{142}$$

$$+ \mathcal{O}(\mathbb{E}[\|\bar{\mathbf{Z}}^{(2)} - \mathbf{L}\bar{\mathbf{H}}^{(1)}\mathbf{W}^{(2)}\|_F^2]) + \mathcal{O}(\mathbb{E}[\|\bar{\mathbf{Z}}^{(1)} - \mathbf{L}\mathbf{X}\mathbf{W}^{(1)}\|_F^2])$$

Using result from Lemma 4, we have

$$\mathbb{E}[\|\nabla_H \bar{f}^{(\ell)}(\mathbf{D}^{(\ell+1)}, \bar{\mathbf{H}}^{(\ell-1)}, \mathbf{W}^{(\ell)}) - \nabla_H f^{(\ell)}(\mathbf{D}^{(\ell+1)}, \mathbf{H}^{(\ell-1)}, \mathbf{W}^{(\ell)})\|_F^2]$$

$$\leq \eta^2 K \times \mathcal{O}\Big(|\mathbb{E}[\|\mathbf{P}^{(\ell)}\|_F^2] - \|\mathbf{L}\|_F^2|\Big) + \dots + \eta^2 K \times \mathcal{O}\Big(|\mathbb{E}[\|\mathbf{P}^{(1)}\|_F^2] - \|\mathbf{L}\|_F^2|\Big) \tag{143}$$

Finally, let consider the upper-bound of Eq. 136.

$$\mathbb{E}[\|\nabla_W \bar{f}^{(\ell)}(\mathbf{D}^{(\ell+1)}, \bar{\mathbf{H}}^{(\ell-1)}, \mathbf{W}^{(\ell)}) - \nabla_W f^{(\ell)}(\mathbf{D}^{(\ell+1)}, \mathbf{H}^{(\ell-1)}, \mathbf{W}^{(\ell)})\|_F^2]$$

$$= \mathbb{E}[\|[\mathbf{L}\bar{\mathbf{H}}^{(\ell-1)}]^\top\Big(\mathbf{D}^{(\ell+1)} \circ \sigma'(\bar{\mathbf{Z}}^{(\ell)})\Big) - [\mathbf{L}\mathbf{H}^{(\ell-1)}]^\top\Big(\mathbf{D}^{(\ell+1)} \circ \sigma'(\mathbf{L}^{(\ell)}\mathbf{H}^{(\ell-1)}\mathbf{W}^{(\ell)})\Big)\|_F^2]$$

$$\leq 2\mathbb{E}[\|[\mathbf{L}\bar{\mathbf{H}}^{(\ell-1)}]^\top\Big(\mathbf{D}^{(\ell+1)} \circ \sigma'(\bar{\mathbf{Z}}^{(\ell)})\Big) - [\mathbf{L}\mathbf{H}^{(\ell-1)}]^\top\Big(\mathbf{D}^{(\ell+1)} \circ \sigma'(\bar{\mathbf{Z}}^{(\ell)})\Big)\|_F^2]$$

$$+ 2\mathbb{E}[\|[\mathbf{L}\mathbf{H}^{(\ell-1)}]^\top\Big(\mathbf{D}^{(\ell+1)} \circ \sigma'(\bar{\mathbf{Z}}^{(\ell)})\Big) - [\mathbf{L}\mathbf{H}^{(\ell-1)}]^\top\Big(\mathbf{D}^{(\ell+1)} \circ \sigma'(\mathbf{L}^{(\ell)}\mathbf{H}^{(\ell-1)}\mathbf{W}^{(\ell)})\Big)\|_F^2]$$

$$\leq 2B_D^2 C_\sigma^2 B_{LA}^2 \underbrace{\mathbb{E}[\|\bar{\mathbf{H}}^{(\ell-1)} - \mathbf{H}^{(\ell-1)}\|_F^2]}_{(B)} + 4B_{LA}^2 B_H^2 B_D^2 L_\sigma^2 \mathbb{E}[\|\bar{\mathbf{Z}}^{(\ell)} - \mathbf{L}\bar{\mathbf{H}}^{(\ell-1)}\mathbf{W}^{(\ell)}\|_F^2]$$

$$+ 4B_{LA}^2 B_H^2 B_D^2 L_\sigma^2 \mathbb{E}[\|\mathbf{L}\bar{\mathbf{H}}^{(\ell-1)}\mathbf{W}^{(\ell)} - \mathbf{L}\mathbf{H}^{(\ell-1)}\mathbf{W}^{(\ell)}\|_F^2] \tag{144}$$

By definition, we can write the term $(B)$ as

$$\mathbb{E}[\|\bar{\mathbf{H}}^{(\ell-1)} - \mathbf{H}^{(\ell-1)}\|_F^2] \leq C_\sigma^2 \mathbb{E}[\|\bar{\mathbf{Z}}^{(\ell)} - \mathbf{L}\mathbf{H}^{(\ell-2)}\mathbf{W}^{(\ell-1)}\|_F^2]$$

$$\leq 2C_\sigma^2 \mathbb{E}[\|\bar{\mathbf{Z}}^{(\ell)} - \mathbf{L}\bar{\mathbf{H}}^{(\ell-2)}\mathbf{W}^{(\ell-1)}\|_F^2] \tag{145}$$

$$+ 2C_\sigma^2 \mathbb{E}[\|\mathbf{L}\bar{\mathbf{H}}^{(\ell-2)}\mathbf{W}^{(\ell)} - \mathbf{L}\mathbf{H}^{(\ell-2)}\mathbf{W}^{(\ell-1)}\|_F^2]$$

Plugging term $(B)$ back and using Eq. 141 and Lemma 4, we have

$$
\begin{aligned}
&\mathbb{E}[\|\nabla_W \bar{f}^{(\ell)}(\mathbf{D}^{(\ell+1)}, \bar{\mathbf{H}}^{(\ell-1)}, \mathbf{W}^{(\ell)}) - \nabla_W f^{(\ell)}(\mathbf{D}^{(\ell+1)}, \mathbf{H}^{(\ell-1)}, \mathbf{W}^{(\ell)})\|_{\mathrm{F}}^2] \\
&\leq \eta^2 K \times \mathcal{O}\Big(|\mathbb{E}[\|\mathbf{P}^{(\ell)}\|_{\mathrm{F}}^2] - \|\mathbf{L}\|_{\mathrm{F}}^2|\Big) + \ldots + \eta^2 K \times \mathcal{O}\Big(|\mathbb{E}[\|\mathbf{P}^{(1)}\|_{\mathrm{F}}^2] - \|\mathbf{L}\|_{\mathrm{F}}^2|\Big)
\end{aligned}
\tag{146}
$$

Combining the result from Eq. 134, 135, 136 we have

$$
\mathbb{E}[\|\mathbb{E}[\widetilde{\mathbf{G}}^{(\ell)}] - \mathbf{G}^{(\ell)}\|_{\mathrm{F}}^2] \leq \eta^2 K \sum_{\ell=1}^{L} \mathcal{O}\Big(|\mathbb{E}[\|\mathbf{P}^{(\ell)}\|_{\mathrm{F}}^2] - \|\mathbf{L}\|_{\mathrm{F}}^2|\Big)
\tag{147}
$$

$\square$

## H.2 REMAINING STEPS TOWARD THEOREM 2

Now we are ready to prove Theorem 2. By the smoothness of $\mathcal{L}(\boldsymbol{\theta}_t)$, we have

$$
\begin{aligned}
\mathcal{L}(\boldsymbol{\theta}_{t+1}) &\leq \mathcal{L}(\boldsymbol{\theta}_t) + \langle \nabla \mathcal{L}(\boldsymbol{\theta}_t), \boldsymbol{\theta}_{t+1} - \boldsymbol{\theta}_t \rangle + \frac{L_{\mathrm{F}}}{2}\|\boldsymbol{\theta}_{t+1} - \boldsymbol{\theta}_t\|^2 \\
&= \mathcal{L}(\boldsymbol{\theta}_t) - \eta \langle \nabla \mathcal{L}(\boldsymbol{\theta}_t), \nabla \widetilde{\mathcal{L}}(\boldsymbol{\theta}_t) \rangle + \frac{\eta^2 L_{\mathrm{F}}}{2}\|\nabla \widetilde{\mathcal{L}}(\boldsymbol{\theta}_t)\|^2
\end{aligned}
\tag{148}
$$

Let $\mathcal{F}_t = \{\{\mathcal{B}_1^{(\ell)}\}_{\ell=1}^{L}, \ldots, \{\mathcal{B}_{t-1}^{(\ell)}\}_{\ell=1}^{L}\}$. Note that the weight parameters $\boldsymbol{\theta}_t$ is a function of history of the generated random process and hence is random. Taking expectation on both sides condition on $\mathcal{F}_t$ and using $\eta < 1/L_{\mathrm{F}}$ we have

$$
\mathbb{E}[\nabla \mathcal{L}(\boldsymbol{\theta}_{t+1})|\mathcal{F}_t]
$$

$$
\begin{aligned}
&\leq \mathcal{L}(\boldsymbol{\theta}_t) - \eta \langle \nabla \mathcal{L}(\boldsymbol{\theta}_t), \mathbb{E}[\nabla \widetilde{\mathcal{L}}(\boldsymbol{\theta}_t)|\mathcal{F}_t] \rangle + \frac{\eta^2 L_{\mathrm{F}}}{2}\Big(\mathbb{E}[\|\nabla \widetilde{\mathcal{L}}(\boldsymbol{\theta}_t) - \mathbb{E}[\nabla \widetilde{\mathcal{L}}(\boldsymbol{\theta}_t)|\mathcal{F}_t]\|^2|\mathcal{F}_t] + \mathbb{E}[\|\mathbb{E}[\boldsymbol{g}|\mathcal{F}_t]\|^2|\mathcal{F}_t]\Big) \\
&= \mathcal{L}(\boldsymbol{\theta}_t) - \eta \langle \nabla \mathcal{L}(\boldsymbol{\theta}_t), \nabla \mathcal{L}(\boldsymbol{\theta}_t) + \mathbb{E}[\mathbf{b}_t|\mathcal{F}_t] \rangle + \frac{\eta^2 L_{\mathrm{F}}}{2}\Big(\mathbb{E}[\|\mathbf{n}_t\|^2|\mathcal{F}_t] + \|\nabla \mathcal{L}(\boldsymbol{\theta}_t) + \mathbb{E}[\mathbf{b}_t|\mathcal{F}_t]\|^2\Big) \\
&\leq \mathcal{L}(\boldsymbol{\theta}_t) + \frac{\eta}{2}\Big(-2\langle \nabla \mathcal{L}(\boldsymbol{\theta}_t), \nabla \mathcal{L}(\boldsymbol{\theta}_t) + \mathbb{E}[\mathbf{b}_t|\mathcal{F}_t] \rangle + \|\nabla \mathcal{L}(\boldsymbol{\theta}_t) + \mathbb{E}[\mathbf{b}_t|\mathcal{F}_t]\|^2\Big) + \frac{\eta^2 L_{\mathrm{F}}}{2}\mathbb{E}[\|\mathbf{n}_t\|^2|\mathcal{F}_t] \\
&\leq \mathcal{L}(\boldsymbol{\theta}_t) + \frac{\eta}{2}\Big(-\|\nabla \mathcal{L}(\boldsymbol{\theta}_t)\|^2 + \mathbb{E}[\|\mathbf{b}_t\|^2|\mathcal{F}_t]\Big) + \frac{\eta^2 L_{\mathrm{F}}}{2}\mathbb{E}[\|\mathbf{n}_t\|^2|\mathcal{F}_t]
\end{aligned}
\tag{149}
$$

Plugging in the upper bound of bias and variance, taking expectation over $\mathcal{F}_t$, and rearranging the term we have

$$
\mathbb{E}[\|\nabla \mathcal{L}(\boldsymbol{\theta}_t)\|^2] \leq \frac{2}{\eta}\Big(\mathbb{E}[\mathcal{L}(\boldsymbol{\theta}_t)] - \mathbb{E}[\nabla \mathcal{L}(\boldsymbol{\theta}_{t+1})]\Big) + \eta L_{\mathrm{F}} \Delta_{\mathbf{n}} + \eta^2 \Delta_{\mathbf{b}}^{+\prime}
\tag{150}
$$

Summing up from $t = 1$ to $T$, rearranging we have

$$
\begin{aligned}
\frac{1}{T}\sum_{t=1}^{T}\mathbb{E}[\|\nabla \mathcal{L}(\boldsymbol{\theta}_t)\|^2] &\leq \frac{2}{\eta T}\sum_{t=1}^{T}(\mathbb{E}[\mathcal{L}(\boldsymbol{\theta}_t)] - \mathbb{E}[\mathcal{L}(\boldsymbol{\theta}_{t+1})]) + \eta L_{\mathrm{F}} \Delta_{\mathbf{n}} + \eta^2 \Delta_{\mathbf{b}}^{+\prime} \\
&\underset{(a)}{\leq} \frac{2}{\eta T}(\mathcal{L}(\boldsymbol{\theta}_1) - \mathcal{L}(\boldsymbol{\theta}^\star)) + \eta L_{\mathrm{F}} \Delta_{\mathbf{n}} + \eta^2 \Delta_{\mathbf{b}}^{+\prime}
\end{aligned}
\tag{151}
$$

where the inequality $(a)$ is due to $\mathcal{L}(\boldsymbol{\theta}^\star) \leq \mathbb{E}[\mathcal{L}(\boldsymbol{\theta}_{T+1})]$.

By selecting learning rate as $\eta = 1/\sqrt{T}$, we have

$$
\frac{1}{T}\sum_{t=1}^{T}\mathbb{E}[\|\nabla \mathcal{L}(\boldsymbol{\theta}_t)\|^2] \leq \frac{2(\mathcal{L}(\boldsymbol{\theta}_1) - \mathcal{L}(\boldsymbol{\theta}^\star))}{\sqrt{T}} + \frac{L_{\mathrm{F}} \Delta_{\mathbf{n}}}{\sqrt{T}} + \frac{\Delta_{\mathbf{b}}^{+\prime}}{T}
\tag{152}
$$

## I   PROOF OF THEOREM 3

### I.1   SUPPORTING LEMMAS

In the following lemma, we decompose the mean-square error of stochastic gradient at the $\ell$th layer $\mathbb{E}[\|\widetilde{\mathbf{G}}^{(\ell)} - \mathbf{G}^{(\ell)}\|F^2]$ as the summation of

- The difference between the gradient with respect to the last layer node embedding matrix

$$\mathbb{E}[\|\widetilde{\mathbf{D}}^{(L+1)} - \mathbf{D}^{(L+1)}\|_{\mathrm{F}}^2] \tag{153}$$

- The difference of gradient passing from the $(\ell+1)$th layer node embedding to the $\ell$th layer node embedding

$$\mathbb{E}[\|\nabla_H \widetilde{f}^{(\ell+1)}(\mathbf{D}^{(\ell+2)}, \widetilde{\mathbf{H}}^{(\ell)}, \mathbf{W}^{(\ell+1)}) - \nabla_H f^{(\ell+1)}(\mathbf{D}^{(\ell+2)}, \mathbf{H}^{(\ell)}, \mathbf{W}^{(\ell+1)})\|_{\mathrm{F}}^2] \tag{154}$$

- The difference of gradient passing from the $\ell$th layer node embedding to the $\ell$th layer weight matrix

$$\mathbb{E}[\|\nabla_W \widetilde{f}^{(\ell)}(\mathbf{D}^{(\ell+1)}, \widetilde{\mathbf{H}}^{(\ell-1)}, \mathbf{W}^{(\ell)}) - \nabla_W f^{(\ell)}(\mathbf{D}^{(\ell+1)}, \mathbf{H}^{(\ell-1)}, \mathbf{W}^{(\ell)})\|_{\mathrm{F}}^2] \tag{155}$$

**Lemma 6.** *The mean-square error of stochastic gradient at the $\ell$th layer can be decomposed as*

$$\begin{aligned}
&\mathbb{E}[\|\widetilde{\mathbf{G}}^{(\ell)} - \mathbf{G}^{(\ell)}\|F^2] \\
&\leq \mathcal{O}(\mathbb{E}[\|\widetilde{\mathbf{D}}^{(L+1)} - \mathbf{D}^{(L+1)}\|_{\mathrm{F}}^2]) \\
&\quad + \mathcal{O}(\mathbb{E}[\|\nabla_H \widetilde{f}^{(L)}(\mathbf{D}^{(L+1)}, \widetilde{\mathbf{H}}^{(L-1)}\mathbf{W}^{(L)}) - \nabla_H f^{(L)}(\mathbf{D}^{(L+1)}, \mathbf{H}^{(L-1)}\mathbf{W}^{(L)})\|_{\mathrm{F}}^2]) + \cdots \\
&\quad + \mathcal{O}(\mathbb{E}[\|\nabla_H \widetilde{f}^{(\ell+1)}(\mathbf{D}^{(\ell+2)}, \widetilde{\mathbf{H}}^{(\ell)}, \mathbf{W}^{(\ell+1)}) - \nabla_H f^{(\ell+1)}(\mathbf{D}^{(\ell+2)}, \mathbf{H}^{(\ell)}, \mathbf{W}^{(\ell+1)})\|_{\mathrm{F}}^2]) \\
&\quad + \mathcal{O}(\mathbb{E}[\|\nabla_W \widetilde{f}^{(\ell)}(\mathbf{D}^{(\ell+1)}, \widetilde{\mathbf{H}}^{(\ell-1)}, \mathbf{W}^{(\ell)}) - \nabla_W f^{(\ell)}(\mathbf{D}^{(\ell+1)}, \mathbf{H}^{(\ell-1)}, \mathbf{W}^{(\ell)})\|_{\mathrm{F}}^2])
\end{aligned} \tag{156}$$

*Proof.* By definition, we can write down the mean-square error of stochastic gradient as

$$\begin{aligned}
&\mathbb{E}[\|\widetilde{\mathbf{G}}^{(\ell)} - \mathbf{G}^{(\ell)}\|_{\mathrm{F}}^2] \\
&= \mathbb{E}[\|\nabla_W \widetilde{f}^{(\ell)}(\nabla_H \widetilde{f}^{(\ell+1)}(\ldots \nabla_H \widetilde{f}^{(L)}(\widetilde{\mathbf{D}}^{(L+1)}, \widetilde{\mathbf{H}}^{(L-1)}\mathbf{W}^{(L)})\ldots, \widetilde{\mathbf{H}}^{(\ell)}, \mathbf{W}^{(\ell+1)}), \widetilde{\mathbf{H}}^{(\ell-1)}, \mathbf{W}^{(\ell)}) \\
&\quad - \nabla_W f^{(\ell)}(\nabla_H f^{(\ell+1)}(\ldots \nabla_H f^{(L)}(\mathbf{D}^{(L+1)}, \mathbf{H}^{(L-1)}, \mathbf{W}^{(L)})\ldots, \mathbf{H}^{(\ell)}, \mathbf{W}^{(\ell+1)}), \mathbf{H}^{(\ell-1)}), \mathbf{W}^{(\ell)}\|_{\mathrm{F}}^2] \\
&\leq (L+1)\mathbb{E}[\|\nabla_W \widetilde{f}^{(\ell)}(\nabla_H \widetilde{f}^{(\ell+1)}(\ldots \nabla_H \widetilde{f}^{(L)}(\widetilde{\mathbf{D}}^{(L+1)}, \widetilde{\mathbf{H}}^{(L-1)}\mathbf{W}^{(L)})\ldots, \widetilde{\mathbf{H}}^{(\ell)}, \mathbf{W}^{(\ell+1)}), \widetilde{\mathbf{H}}^{(\ell-1)}, \mathbf{W}^{(\ell)}) \\
&\quad\quad - \nabla_W \widetilde{f}^{(\ell)}(\nabla_H \widetilde{f}^{(\ell+1)}(\ldots \nabla_H \widetilde{f}^{(L)}(\mathbf{D}^{(L+1)}, \widetilde{\mathbf{H}}^{(L-1)}\mathbf{W}^{(L)})\ldots, \mathbf{H}^{(\ell)}, \mathbf{W}^{(\ell+1)}), \mathbf{H}^{(\ell-1)}, \mathbf{W}^{(\ell)})\|_{\mathrm{F}}^2] \\
&\quad + (L+1)\mathbb{E}[\|\nabla_W \widetilde{f}^{(\ell)}(\nabla_H \widetilde{f}^{(\ell+1)}(\ldots \nabla_H \widetilde{f}^{(L)}(\mathbf{D}^{(L+1)}, \widetilde{\mathbf{H}}^{(L-1)}\mathbf{W}^{(L)})\ldots, \widetilde{\mathbf{H}}^{(\ell)}, \mathbf{W}^{(\ell+1)}), \widetilde{\mathbf{H}}^{(\ell-1)}, \mathbf{W}^{(\ell)}) \\
&\quad\quad - \nabla_W \widetilde{f}^{(\ell)}(\nabla_H \widetilde{f}^{(\ell+1)}(\ldots \nabla_H f^{(L)}(\mathbf{D}^{(L+1)}, \mathbf{H}^{(L-1)}\mathbf{W}^{(L)})\ldots, \widetilde{\mathbf{H}}^{(\ell)}, \mathbf{W}^{(\ell+1)}), \widetilde{\mathbf{H}}^{(\ell-1)}, \mathbf{W}^{(\ell)})\|_{\mathrm{F}}^2] + \cdots \\
&\quad + (L+1)\mathbb{E}[\|\nabla_W \widetilde{f}^{(\ell)}(\nabla_H \widetilde{f}^{(\ell+1)}(\mathbf{D}^{(\ell+2)}, \widetilde{\mathbf{H}}^{(\ell)}, \mathbf{W}^{(\ell+1)}), \widetilde{\mathbf{H}}^{(\ell-1)}, \mathbf{W}^{(\ell)}) \\
&\quad\quad - \nabla_W \widetilde{f}^{(\ell)}(\nabla_H f^{(\ell+1)}(\mathbf{D}^{(\ell+2)}, \mathbf{H}^{(\ell)}, \mathbf{W}^{(\ell+1)}), \widetilde{\mathbf{H}}^{(\ell-1)}, \mathbf{W}^{(\ell)})\|_{\mathrm{F}}^2] \\
&\quad + (L+1)\mathbb{E}[\|\nabla_W \widetilde{f}^{(\ell)}(\mathbf{D}^{(\ell+1)}, \widetilde{\mathbf{H}}^{(\ell-1)}, \mathbf{W}^{(\ell)}) - \nabla_W f^{(\ell)}(\mathbf{D}^{(\ell+1)}, \mathbf{H}^{(\ell-1)}, \mathbf{W}^{(\ell)})\|_{\mathrm{F}}^2] \\
&\leq \mathcal{O}(\mathbb{E}[\|\widetilde{\mathbf{D}}^{(L+1)} - \mathbf{D}^{(L+1)}\|_{\mathrm{F}}^2]) \\
&\quad + \mathcal{O}(\mathbb{E}[\|\nabla_H \widetilde{f}^{(L)}(\mathbf{D}^{(L+1)}, \widetilde{\mathbf{H}}^{(L-1)}\mathbf{W}^{(L)}) - \nabla_H f^{(L)}(\mathbf{D}^{(L+1)}, \mathbf{H}^{(L-1)}\mathbf{W}^{(L)})\|_{\mathrm{F}}^2]) + \cdots \\
&\quad + \mathcal{O}(\mathbb{E}[\|\nabla_H \widetilde{f}^{(\ell+1)}(\mathbf{D}^{(\ell+2)}, \widetilde{\mathbf{H}}^{(\ell)}, \mathbf{W}^{(\ell+1)}) - \nabla_H f^{(\ell+1)}(\mathbf{D}^{(\ell+2)}, \mathbf{H}^{(\ell)}, \mathbf{W}^{(\ell+1)})\|_{\mathrm{F}}^2]) \\
&\quad + \mathcal{O}(\mathbb{E}[\|\nabla_W \widetilde{f}^{(\ell)}(\mathbf{D}^{(\ell+1)}, \widetilde{\mathbf{H}}^{(\ell-1)}, \mathbf{W}^{(\ell)}) - \nabla_W f^{(\ell)}(\mathbf{D}^{(\ell+1)}, \mathbf{H}^{(\ell-1)}, \mathbf{W}^{(\ell)})\|_{\mathrm{F}}^2])
\end{aligned} \tag{157}$$

$\square$

Recall the definition of stochastic gradient for all model parameters $\nabla \widetilde{\mathcal{L}}(\boldsymbol{\theta}_t) = \{\widetilde{\mathbf{G}}_t^{(\ell)}\}_{\ell=1}^L$ where $\widetilde{\mathbf{G}}_t^{(\ell)}$ is the gradient for the $\ell$th weight matrix, i.e.,

$$\mathbf{W}_t^{(\ell)} = \mathbf{W}_{t-1}^{(\ell)} - \eta \widetilde{\mathbf{G}}_{t-1}^{(\ell)} \tag{158}$$

In the following lemma, we derive the upper-bound on the difference of the gradient passing from the $\ell$th to $(\ell-1)$th layer given the same inputs $\mathbf{D}_t^{(\ell+1)}$, $\widetilde{\mathbf{H}}_t^{(\ell-1)}$, where the backward propagation for the $\ell$th layer in SGCN++ is defined as

$$
\begin{aligned}
&\nabla_H \widetilde{f}^{(\ell)}(\mathbf{D}_t^{(\ell+1)}, \widetilde{\mathbf{H}}_t^{(\ell-1)}, \mathbf{W}_t^{(\ell)}) \\
&= \widetilde{\mathbf{D}}_{t-1}^{(\ell)} + [\widetilde{\mathbf{L}}^{(\ell)}]^\top (\mathbf{D}_t^{(\ell+1)} \circ \sigma'(\widetilde{\mathbf{Z}}_t^{(\ell)})) \mathbf{W}_t^{(\ell)} - [\widetilde{\mathbf{L}}^{(\ell)}]^\top (\mathbf{D}_{t-1}^{(\ell+1)} \circ \sigma'(\widetilde{\mathbf{Z}}_{t-1}^{(\ell)})) \mathbf{W}_{t-1}^{(\ell)}]
\end{aligned}
\tag{159}
$$

and the backward propagation for the $\ell$th layer in FullGCN is defined as

$$
\nabla_H f^{(\ell)}(\mathbf{D}_t^{(\ell+1)}, \widetilde{\mathbf{H}}_t^{(\ell-1)}, \mathbf{W}_t^{(\ell)}) = \mathbf{L}^\top (\mathbf{D}_t^{(\ell+1)} \circ \sigma'(\widetilde{\mathbf{Z}}_t^{(\ell)})) \mathbf{W}_t^{(\ell)}
\tag{160}
$$

**Lemma 7.** *Let suppose $t \in \{EK+1, \ldots, EK+K\}$. The upper-bound on the difference of the gradient with respect to the input node embedding matrix at the $\ell$th graph convolutional layer given the same input $\mathbf{D}_t^{(\ell+1)}$ and $\widetilde{\mathbf{H}}_t^{(\ell-1)}$ is defined as*

$$
\begin{aligned}
&\mathbb{E}[\|\nabla_H \widetilde{f}^{(\ell)}(\mathbf{D}_t^{(\ell+1)}, \widetilde{\mathbf{H}}_t^{(\ell-1)}, \mathbf{W}_t^{(\ell)}) - \nabla_H f^{(\ell)}(\mathbf{D}_t^{(\ell+1)}, \widetilde{\mathbf{H}}_t^{(\ell-1)}, \mathbf{W}_t^{(\ell)})\|_F^2] \\
&\leq \sum_{t=EK+1}^{EK+K} \eta^2 \mathcal{O}\Big(|\mathbb{E}[\|\widetilde{\mathbf{L}}^{(\ell)}\|_F^2] - \|\mathbf{L}\|_F^2| \times \mathbb{E}[\|\nabla\widetilde{\mathcal{L}}(\boldsymbol{\theta}_{t-1})\|_F^2]\Big)
\end{aligned}
\tag{161}
$$

*Proof.* To simplify the presentation, let us denote $\widetilde{\mathbf{D}}_t^{(\ell)} = \nabla_H \widetilde{f}^{(\ell)}(\mathbf{D}_t^{(\ell+1)}, \widetilde{\mathbf{H}}_t^{(\ell-1)}, \mathbf{W}_t^{(\ell)})$. Then, by definition we have

$$
\widetilde{\mathbf{D}}_t^{(\ell)} := \widetilde{\mathbf{D}}_{t-1}^{(\ell)} + [\widetilde{\mathbf{L}}^{(\ell)}]^\top (\mathbf{D}_t^{(\ell+1)} \circ \sigma'(\widetilde{\mathbf{Z}}_t^{(\ell)})) \mathbf{W}_t^{(\ell)} - [\widetilde{\mathbf{L}}^{(\ell)}]^\top (\mathbf{D}_{t-1}^{(\ell+1)} \circ \sigma'(\widetilde{\mathbf{Z}}_{t-1}^{(\ell)})) \mathbf{W}_{t-1}^{(\ell)}
\tag{162}
$$

Therefore, we know that

$$
\begin{aligned}
&\|\mathbf{L}^\top (\mathbf{D}_t^{(\ell+1)} \circ \sigma'(\widetilde{\mathbf{Z}}_t^{(\ell)})) \mathbf{W}_t^{(\ell)} - \widetilde{\mathbf{D}}_t^{(\ell)}\|_F^2 \\
&= \|[\mathbf{L}^\top (\mathbf{D}_t^{(\ell+1)} \circ \sigma'(\widetilde{\mathbf{Z}}_t^{(\ell)})) \mathbf{W}_t^{(\ell)} - \mathbf{L}^\top (\mathbf{D}_{t-1}^{(\ell+1)} \circ \sigma'(\widetilde{\mathbf{Z}}_{t-1}^{(\ell)})) \mathbf{W}_{t-1}^{(\ell)}] \\
&\quad + [\mathbf{L}^\top (\mathbf{D}_{t-1}^{(\ell+1)} \circ \sigma'(\widetilde{\mathbf{Z}}_{t-1}^{(\ell)})) \mathbf{W}_{t-1}^{(\ell)} - \widetilde{\mathbf{D}}_{t-1}^{(\ell)}] - [\widetilde{\mathbf{D}}_t^{(\ell)} - \widetilde{\mathbf{D}}_{t-1}^{(\ell)}]\|_F^2 \\
&\leq \|\underbrace{\mathbf{L}^\top (\mathbf{D}_t^{(\ell+1)} \circ \sigma'(\widetilde{\mathbf{Z}}_t^{(\ell)})) \mathbf{W}_t^{(\ell)} - \mathbf{L}^\top (\mathbf{D}_{t-1}^{(\ell+1)} \circ \sigma'(\widetilde{\mathbf{Z}}_{t-1}^{(\ell)})) \mathbf{W}_{t-1}^{(\ell)}}_{(A_1)}\|_F^2 \\
&\quad + \|\underbrace{\mathbf{L}^\top (\mathbf{D}_{t-1}^{(\ell+1)} \circ \sigma'(\widetilde{\mathbf{Z}}_{t-1}^{(\ell)})) \mathbf{W}_{t-1}^{(\ell)} - \widetilde{\mathbf{D}}_{t-1}^{(\ell)}}_{A_2}\|_F^2 + \|\underbrace{\widetilde{\mathbf{D}}_t^{(\ell)} - \widetilde{\mathbf{D}}_{t-1}^{(\ell)}}_{A_3}\|_F^2 \\
&\quad + 2\langle A_1, A_2 \rangle - 2\langle A_1, A_3 \rangle - 2\langle A_2, A_3 \rangle
\end{aligned}
\tag{163}
$$

Taking expectation condition on $\mathcal{F}_t$ on both side, and using the fact that

$$
\begin{aligned}
&\mathbb{E}[\widetilde{\mathbf{D}}_t^{(\ell)} - \widetilde{\mathbf{D}}_{t-1}^{(\ell)} | \mathcal{F}_t] \\
&= \mathbb{E}[[\widetilde{\mathbf{L}}^{(\ell)}]^\top (\mathbf{D}_t^{(\ell+1)} \circ \sigma'(\widetilde{\mathbf{Z}}_t^{(\ell)})) \mathbf{W}_t^{(\ell)} - [\widetilde{\mathbf{L}}^{(\ell)}]^\top (\mathbf{D}_{t-1}^{(\ell+1)} \circ \sigma'(\widetilde{\mathbf{Z}}_{t-1}^{(\ell)})) \mathbf{W}_{t-1}^{(\ell)} | \mathcal{F}_t] \\
&= \mathbf{L}^\top (\mathbf{D}_t^{(\ell+1)} \circ \sigma'(\widetilde{\mathbf{Z}}_t^{(\ell)})) \mathbf{W}_t^{(\ell)} - \mathbf{L}^\top (\mathbf{D}_{t-1}^{(\ell+1)} \circ \sigma'(\widetilde{\mathbf{Z}}_{t-1}^{(\ell)})) \mathbf{W}_{t-1}^{(\ell)}
\end{aligned}
\tag{164}
$$

the following inequality holds

$$
\begin{aligned}
&\mathbb{E}[\|\mathbf{L}^\top (\mathbf{D}_t^{(\ell+1)} \circ \sigma'(\widetilde{\mathbf{Z}}_t^{(\ell)})) \mathbf{W}_t^{(\ell)} - \widetilde{\mathbf{D}}_t^{(\ell)}\|_F^2 | \mathcal{F}_t] \\
&\leq \|\mathbf{L}^\top (\mathbf{D}_{t-1}^{(\ell+1)} \circ \sigma'(\widetilde{\mathbf{Z}}_{t-1}^{(\ell)})) \mathbf{W}_t^{(\ell)} - \widetilde{\mathbf{D}}_{t-1}^{(\ell)}\|_F^2 + \mathbb{E}[\|\widetilde{\mathbf{D}}_t^{(\ell)} - \widetilde{\mathbf{D}}_{t-1}^{(\ell)}\|_F^2 | \mathcal{F}_t] \\
&\quad - \|\mathbf{L}^\top (\mathbf{D}_t^{(\ell+1)} \circ \sigma'(\widetilde{\mathbf{Z}}_t^{(\ell)})) \mathbf{W}_t^{(\ell)} - \mathbf{L}^\top (\mathbf{D}_{t-1}^{(\ell+1)} \circ \sigma'(\widetilde{\mathbf{Z}}_{t-1}^{(\ell)})) \mathbf{W}_{t-1}^{(\ell)}\|_F^2
\end{aligned}
\tag{165}
$$

Then, taking expectation over $\mathcal{F}_t$, we have

$$
\begin{aligned}
&\mathbb{E}[\|\mathbf{L}^\top (\mathbf{D}_t^{(\ell+1)} \circ \sigma'(\widetilde{\mathbf{Z}}_t^{(\ell)})) \mathbf{W}_t^{(\ell)} - \widetilde{\mathbf{D}}_t^{(\ell)}\|_F^2] \\
&\leq \mathbb{E}[\|\mathbf{L}^\top (\mathbf{D}_{t-1}^{(\ell+1)} \circ \sigma'(\widetilde{\mathbf{Z}}_{t-1}^{(\ell)})) \mathbf{W}_{t-1}^{(\ell)} - \widetilde{\mathbf{D}}_{t-1}^{(\ell)}\|_F^2] + \mathbb{E}[\|\widetilde{\mathbf{D}}_t^{(\ell)} - \widetilde{\mathbf{D}}_{t-1}^{(\ell)}\|_F^2] \\
&\quad - [\|\mathbf{L}^\top (\mathbf{D}_t^{(\ell+1)} \circ \sigma'(\widetilde{\mathbf{Z}}_t^{(\ell)})) \mathbf{W}_t^{(\ell)} - \mathbf{L}^\top (\mathbf{D}_{t-1}^{(\ell+1)} \circ \sigma'(\widetilde{\mathbf{Z}}_{t-1}^{(\ell)})) \mathbf{W}_{t-1}^{(\ell)}\|_F^2]
\end{aligned}
\tag{166}
$$

Let suppose $t \in \{EK+1, \ldots, EK+K\}$. Then we can denote $t = EK+k$ for some $k \leq K$ such that

$$\mathbb{E}[\|\mathbf{L}^\top(\mathbf{D}_t^{(\ell+1)} \circ \sigma'(\widetilde{\mathbf{Z}}_t^{(\ell)}))\mathbf{W}_t^{(\ell)} - \widetilde{\mathbf{D}}_t^{(\ell)}\|_{\mathrm{F}}^2]$$

$$\mathbb{E}[\|\mathbf{L}^\top(\mathbf{D}_{EK+k}^{(\ell+1)} \circ \sigma'(\widetilde{\mathbf{Z}}_{EK+k}^{(\ell)}))\mathbf{W}_{EK+k}^{(\ell)} - \widetilde{\mathbf{D}}_{EK+k}^{(\ell)}\|_{\mathrm{F}}^2]$$

$$\leq \mathbb{E}[\|\mathbf{L}^\top(\mathbf{D}_{EK}^{(\ell+1)} \circ \sigma'(\widetilde{\mathbf{Z}}_{EK}^{(\ell)}))\mathbf{W}_{EK}^{(\ell)} - \widetilde{\mathbf{D}}_{EK}^{(\ell)}\|_{\mathrm{F}}^2] + \sum_{t=EK+1}^{EK+K} \Big( \mathbb{E}[\|\widetilde{\mathbf{D}}_t^{(\ell)} - \widetilde{\mathbf{D}}_{t-1}^{(\ell)}\|_{\mathrm{F}}^2]$$

$$- [\|\mathbf{L}^\top(\mathbf{D}_t^{(\ell+1)} \circ \sigma'(\widetilde{\mathbf{Z}}_t^{(\ell)}))\mathbf{W}_t^{(\ell)} - \mathbf{L}^\top(\mathbf{D}_{t-1}^{(\ell+1)} \circ \sigma'(\widetilde{\mathbf{Z}}_{t-1}^{(\ell)}))\mathbf{W}_{t-1}^{(\ell)}\|_{\mathrm{F}}^2] \Big)$$

$$\leq \mathbb{E}[\|\mathbf{L}^\top(\mathbf{D}_{EK}^{(\ell+1)} \circ \sigma'(\widetilde{\mathbf{Z}}_{EK}^{(\ell)}))\mathbf{W}_{EK}^{(\ell)} - \widetilde{\mathbf{D}}_{EK}^{(\ell)}\|_{\mathrm{F}}^2]$$

$$+ \sum_{t=EK+1}^{EK+K} \Big( \mathbb{E}[\|[\widetilde{\mathbf{L}}^{(\ell)}]^\top(\mathbf{D}_t^{(\ell+1)} \circ \sigma'(\widetilde{\mathbf{Z}}_t^{(\ell)}))\mathbf{W}_t^{(\ell)} - [\widetilde{\mathbf{L}}^{(\ell)}]^\top(\mathbf{D}_{t-1}^{(\ell+1)} \circ \sigma'(\widetilde{\mathbf{Z}}_{t-1}^{(\ell)}))\mathbf{W}_{t-1}^{(\ell)}\|_{\mathrm{F}}^2]$$

$$- \mathbb{E}[\|\mathbf{L}^\top(\mathbf{D}_t^{(\ell+1)} \circ \sigma'(\widetilde{\mathbf{Z}}_t^{(\ell)}))\mathbf{W}_t^{(\ell)} - \mathbf{L}^\top(\mathbf{D}_{t-1}^{(\ell+1)} \circ \sigma'(\widetilde{\mathbf{Z}}_{t-1}^{(\ell)}))\mathbf{W}_{t-1}^{(\ell)}\|_{\mathrm{F}}^2] \Big) \tag{167}$$

Knowing that we are taking full-batch gradient descent when $(t \mod K) = 0$, we have

$$\mathbb{E}[\|\mathbf{L}^\top(\mathbf{D}_t^{(\ell+1)} \circ \sigma'(\widetilde{\mathbf{Z}}_t^{(\ell)}))\mathbf{W}_t^{(\ell)} - \widetilde{\mathbf{D}}_t^{(\ell)}\|_{\mathrm{F}}^2]$$

$$\leq \sum_{t=EK+1}^{EK+K} \Big( |\mathbb{E}[\|\widetilde{\mathbf{L}}^{(\ell)}\|_{\mathrm{F}}^2] - \|\mathbf{L}\|_{\mathrm{F}}^2| \Big) \times \underbrace{\mathbb{E}[\|(\mathbf{D}_t^{(\ell+1)} \circ \sigma'(\widetilde{\mathbf{Z}}_t^{(\ell)}))\mathbf{W}_t^{(\ell)} - (\mathbf{D}_{t-1}^{(\ell+1)} \circ \sigma'(\widetilde{\mathbf{Z}}_{t-1}^{(\ell)}))\mathbf{W}_{t-1}^{(\ell)}\|_{\mathrm{F}}^2]}_{(B)} \tag{168}$$

Let take closer look at term $(B)$.

$$\mathbb{E}[\|(\mathbf{D}_t^{(\ell+1)} \circ \sigma'(\widetilde{\mathbf{Z}}_t^{(\ell)}))\mathbf{W}_t^{(\ell)} - (\mathbf{D}_{t-1}^{(\ell+1)} \circ \sigma'(\widetilde{\mathbf{Z}}_{t-1}^{(\ell)}))\mathbf{W}_{t-1}^{(\ell)}\|_{\mathrm{F}}^2]$$

$$\leq 3C_\sigma^2 B_W^2 \underbrace{\|\mathbf{D}_t^{(\ell+1)} - \mathbf{D}_{t-1}^{(\ell+1)}\|_{\mathrm{F}}^2}_{(C_1)} + 3B_D^2 B_W^2 L_\sigma^2 \underbrace{\mathbb{E}[\|\widetilde{\mathbf{Z}}_t^{(\ell)} - \widetilde{\mathbf{Z}}_{t-1}^{(\ell)}\|_{\mathrm{F}}^2]}_{(C_2)} + 3C_\sigma^2 B_D^2 \mathbb{E}[\|\mathbf{W}_t^{(\ell)} - \mathbf{W}_{t-1}^{(\ell)}\|_{\mathrm{F}}^2] \tag{169}$$

For term $(C_1)$ by definition we know

$$\|\mathbf{D}_t^{(\ell+1)} - \mathbf{D}_{t-1}^{(\ell+1)}\|_{\mathrm{F}}^2$$

$$= \|\Big(\mathbf{L}^\top(\mathbf{D}_t^{(\ell+2)} \circ \sigma'(\mathbf{Z}_t^{(\ell+1)}))\mathbf{W}_t^{(\ell+1)}\Big) - \Big(\mathbf{L}^\top(\mathbf{D}_{t-1}^{(\ell+2)} \circ \sigma'(\mathbf{Z}_{t-1}^{(\ell+1)}))\mathbf{W}_{t-1}^{(\ell+1)}\Big)\|_{\mathrm{F}}^2$$

$$\leq 3\|\Big(\mathbf{L}^\top(\mathbf{D}_t^{(\ell+2)} \circ \sigma'(\mathbf{Z}_t^{(\ell+1)}))\mathbf{W}_t^{(\ell+1)}\Big) - \Big(\mathbf{L}^\top(\mathbf{D}_{t-1}^{(\ell+2)} \circ \sigma'(\mathbf{Z}_t^{(\ell+1)}))\mathbf{W}_t^{(\ell+1)}\Big)\|_{\mathrm{F}}^2$$

$$+ 3\|\Big(\mathbf{L}^\top(\mathbf{D}_{t-1}^{(\ell+2)} \circ \sigma'(\mathbf{Z}_t^{(\ell+1)}))\mathbf{W}_t^{(\ell+1)}\Big) - \Big(\mathbf{L}^\top(\mathbf{D}_{t-1}^{(\ell+2)} \circ \sigma'(\mathbf{Z}_{t-1}^{(\ell+1)}))\mathbf{W}_t^{(\ell+1)}\Big)\|_{\mathrm{F}}^2 \tag{170}$$

$$+ 3\|\Big(\mathbf{L}^\top(\mathbf{D}_{t-1}^{(\ell+2)} \circ \sigma'(\mathbf{Z}_{t-1}^{(\ell+1)}))\mathbf{W}_t^{(\ell+1)}\Big) - \Big(\mathbf{L}^\top(\mathbf{D}_{t-1}^{(\ell+2)} \circ \sigma'(\mathbf{Z}_{t-1}^{(\ell+1)}))\mathbf{W}_{t-1}^{(\ell+1)}\Big)\|_{\mathrm{F}}^2$$

$$\leq \mathcal{O}(\|\mathbf{D}_t^{(\ell+2)} - \mathbf{D}_{t-1}^{(\ell+2)}\|_{\mathrm{F}}^2) + \mathcal{O}(\|\mathbf{Z}_t^{(\ell+1)} - \mathbf{Z}_{t-1}^{(\ell+1)}\|_{\mathrm{F}}^2) + \mathcal{O}(\|\mathbf{W}_t^{(\ell+1)} - \mathbf{W}_{t-1}^{(\ell+1)}\|_{\mathrm{F}}^2)$$

By induction, we have

$$\|\mathbf{D}_t^{(\ell+1)} - \mathbf{D}_{t-1}^{(\ell+1)}\|_{\mathrm{F}}^2 \leq \underbrace{\mathcal{O}(\|\mathbf{D}_t^{(L+1)} - \mathbf{D}_{t-1}^{(L+1)}\|_{\mathrm{F}}^2)}_{(D_1)} + \underbrace{\mathcal{O}(\|\mathbf{Z}_t^{(\ell+1)} - \mathbf{Z}_{t-1}^{(\ell+1)}\|_{\mathrm{F}}^2)}_{(D_2)} + \ldots + \mathcal{O}(\|\mathbf{Z}_t^{(L)} - \mathbf{Z}_{t-1}^{(L)}\|_{\mathrm{F}}^2)$$

$$+ \mathcal{O}(\|\mathbf{W}_t^{(\ell+1)} - \mathbf{W}_{t-1}^{(\ell+1)}\|_{\mathrm{F}}^2) + \ldots + \mathcal{O}(\|\mathbf{W}_t^{(L)} - \mathbf{W}_{t-1}^{(L)}\|_{\mathrm{F}}^2) \tag{171}$$

For term $(D_1)$ we have

$$
\begin{aligned}
\|\mathbf{D}_t^{(L+1)} - \mathbf{D}_{t-1}^{(L+1)}\|_{\mathrm{F}}^2 &= \|\frac{\partial \mathcal{L}(\boldsymbol{\theta}_t)}{\partial \mathbf{W}_t^{(L)}} - \frac{\partial \mathcal{L}(\boldsymbol{\theta}_{t-1})}{\partial \mathbf{W}_{t-1}^{(L)}}\|_{\mathrm{F}}^2 \\
&\leq L_{\mathrm{loss}}^2 C_\sigma^2 \|\mathbf{Z}_t^{(L)} - \mathbf{Z}_{t-1}^{(L)}\|_{\mathrm{F}}^2
\end{aligned}
\tag{172}
$$

For term $(D_2)$ we have

$$
\begin{aligned}
\|\mathbf{Z}_t^{(\ell+1)} - \mathbf{Z}_{t-1}^{(\ell+1)}\|_{\mathrm{F}}^2 &\leq C_\sigma^2 \|\mathbf{L}\mathbf{H}_t^{(\ell)}\mathbf{W}_t^{(\ell+1)} - \mathbf{L}\mathbf{H}_{t-1}^{(\ell)}\mathbf{W}_{t-1}^{(\ell+1)}\|_{\mathrm{F}}^2 \\
&\leq C_\sigma^2 B_{LA}^2 \|\mathbf{H}_t^{(\ell)}\mathbf{W}_t^{(\ell+1)} - \mathbf{H}_{t-1}^{(\ell)}\mathbf{W}_t^{(\ell+1)} + \mathbf{H}_{t-1}^{(\ell)}\mathbf{W}_t^{(\ell+1)} - \mathbf{H}_{t-1}^{(\ell)}\mathbf{W}_{t-1}^{(\ell+1)}\|_{\mathrm{F}}^2 \\
&\leq 2C_\sigma^2 B_{LA}^2 \|\mathbf{H}_t^{(\ell)}\mathbf{W}_t^{(\ell+1)} - \mathbf{H}_{t-1}^{(\ell)}\mathbf{W}_t^{(\ell+1)}\|_{\mathrm{F}}^2 + 2C_\sigma^2 B_{LA}^2 \|\mathbf{H}_{t-1}^{(\ell)}\mathbf{W}_t^{(\ell+1)} - \mathbf{H}_{t-1}^{(\ell)}\mathbf{W}_{t-1}^{(\ell+1)}\|_{\mathrm{F}}^2 \\
&\leq 2C_\sigma^4 B_{LA}^2 B_W^2 \|\mathbf{Z}_t^{(\ell)} - \mathbf{Z}_{t-1}^{(\ell)}\|_{\mathrm{F}}^2 + 2C_\sigma^2 B_{LA}^2 B_H^2 \|\mathbf{W}_t^{(\ell+1)} - \mathbf{W}_{t-1}^{(\ell+1)}\|_{\mathrm{F}}^2
\end{aligned}
\tag{173}
$$

By induction we have

$$
\|\mathbf{Z}_t^{(\ell+1)} - \mathbf{Z}_{t-1}^{(\ell+1)}\|_{\mathrm{F}}^2 \leq \mathcal{O}(\|\mathbf{W}_t^{(\ell+1)} - \mathbf{W}_{t-1}^{(\ell+1)}\|_{\mathrm{F}}^2) + \ldots + \mathcal{O}(\|\mathbf{W}_t^{(1)} - \mathbf{W}_{t-1}^{(1)}\|_{\mathrm{F}}^2)
\tag{174}
$$

For term $(C_2)$ by definition we have

$$
\begin{aligned}
\mathbb{E}[\|\widetilde{\mathbf{Z}}_t^{(\ell)} - \widetilde{\mathbf{Z}}_{t-1}^{(\ell)}\|_{\mathrm{F}}^2] &= \mathbb{E}[\|\widetilde{\mathbf{L}}^{(\ell)}\widetilde{\mathbf{H}}_t^{(\ell-1)}\mathbf{W}_t^{(\ell)} - \widetilde{\mathbf{L}}^{(\ell)}\widetilde{\mathbf{H}}_{t-1}^{(\ell-1)}\mathbf{W}_{t-1}^{(\ell)}\|_{\mathrm{F}}^2] \\
&\leq 2B_{LA}^2 B_W^2 C_\sigma^2 \mathbb{E}[\|\widetilde{\mathbf{Z}}_t^{(\ell-1)} - \widetilde{\mathbf{Z}}_{t-1}^{(\ell-1)}\|_{\mathrm{F}}^2] + 2B_{LA}^2 B_H^2 \mathbb{E}[\|\mathbf{W}_t^{(\ell)} - \mathbf{W}_{t-1}^{(\ell)}\|_{\mathrm{F}}^2]
\end{aligned}
\tag{175}
$$

By induction we have

$$
\mathbb{E}[\|\widetilde{\mathbf{Z}}_t^{(\ell)} - \widetilde{\mathbf{Z}}_{t-1}^{(\ell)}\|_{\mathrm{F}}^2] \leq \mathcal{O}(\mathbb{E}[\|\mathbf{W}_t^{(\ell)} - \mathbf{W}_{t-1}^{(\ell)}\|_{\mathrm{F}}^2]) + \ldots + \mathcal{O}(\mathbb{E}[\|\mathbf{W}_t^{(1)} - \mathbf{W}_{t-1}^{(1)}\|_{\mathrm{F}}^2])
\tag{176}
$$

Plugging $(D_1), (D_2)$ back to $(C_1)$ and $(C_1), (C_2), (C_3)$ back to $(B)$, we have

$$
\begin{aligned}
&\mathbb{E}[\|(\mathbf{D}_t^{(\ell+1)} \circ \sigma'(\widetilde{\mathbf{Z}}_t^{(\ell)}))\mathbf{W}_t^{(\ell)} - (\mathbf{D}_{t-1}^{(\ell+1)} \circ \sigma'(\widetilde{\mathbf{Z}}_{t-1}^{(\ell)}))\mathbf{W}_{t-1}^{(\ell)}\|_{\mathrm{F}}^2] \\
&\leq \sum_{t=EK+1}^{EK+K} \left( \mathcal{O}(\mathbb{E}[\|\mathbf{W}_t^{(1)} - \mathbf{W}_{t-1}^{(1)}\|_{\mathrm{F}}^2]) + \ldots + \mathcal{O}(\mathbb{E}[\|\mathbf{W}_t^{(L)} - \mathbf{W}_{t-1}^{(L)}\|_{\mathrm{F}}^2]) \right) \\
&= \sum_{t=EK+1}^{EK+K} \eta^2 \mathcal{O}\left( \mathbb{E}[\|\nabla\widetilde{\mathcal{L}}(\boldsymbol{\theta}_{t-1})\|_{\mathrm{F}}^2] \right)
\end{aligned}
\tag{177}
$$

Then plugging term $(B)$ back to Eq. 169 we conclude the proof.

$\square$

Using the previous lemma, we provide the upper-bound of Eq. 154, which is one of the three key factors that affect the mean-square error of stochastic gradient at the $\ell$th layer.

**Lemma 8.** *Let suppose $t \in \{EK+1, \ldots, EK+K\}$. The upper-bound on the difference of the gradient with respect to the input node embedding matrix at the $\ell$th graph convolutional layer given the same input $\mathbf{D}_t^{(\ell+1)}$ but different input $\widetilde{\mathbf{H}}_t^{(\ell-1)}, \mathbf{H}_t^{(\ell-1)}$ is defined as*

$$
\begin{aligned}
&\mathbb{E}[\|\nabla_H \widetilde{f}^{(\ell)}(\mathbf{D}_t^{(\ell+1)}, \widetilde{\mathbf{H}}_t^{(\ell-1)}, \mathbf{W}_t^{(\ell)}) - \nabla_H f^{(\ell)}(\mathbf{D}_t^{(\ell+1)}, \mathbf{H}_t^{(\ell-1)}, \mathbf{W}_t^{(\ell)})\|_{\mathrm{F}}^2] \\
&\leq \sum_{t=EK+1}^{EK+K} \eta^2 \mathcal{O}\left( |\mathbb{E}[\|\widetilde{\mathbf{L}}^{(\ell)}\|_{\mathrm{F}}^2] - \|\mathbf{L}\|_{\mathrm{F}}^2| \times \mathbb{E}[\|\nabla\widetilde{\mathcal{L}}(\boldsymbol{\theta}_{t-1})\|_{\mathrm{F}}^2] \right)
\end{aligned}
\tag{178}
$$

*Proof.* For the gradient w.r.t. the node embedding matrices, we have

$$
\begin{aligned}
&\mathbb{E}[\|\nabla_H \widetilde{f}^{(\ell)}(\mathbf{D}_t^{(\ell+1)}, \widetilde{\mathbf{H}}_t^{(\ell-1)}, \mathbf{W}_t^{(\ell)}) - \nabla_H f^{(\ell)}(\mathbf{D}_t^{(\ell+1)}, \mathbf{H}_t^{(\ell-1)}, \mathbf{W}_t^{(\ell)})\|_F^2] \\
&= \mathbb{E}[\|\Big(\widetilde{\mathbf{D}}_{t-1}^{(\ell)} + [\widetilde{\mathbf{L}}^{(\ell)}]^\top (\mathbf{D}_t^{(\ell+1)} \circ \sigma'(\widetilde{\mathbf{Z}}_t^{(\ell)}))\mathbf{W}_t^{(\ell)} - [\widetilde{\mathbf{L}}^{(\ell)}]^\top (\mathbf{D}_{t-1}^{(\ell+1)} \circ \sigma'(\widetilde{\mathbf{Z}}_{t-1}^{(\ell)}))\mathbf{W}_{t-1}^{(\ell)}]\Big) \\
&\quad - \Big(\mathbf{L}^\top (\mathbf{D}_t^{(\ell+1)} \circ \sigma'(\mathbf{Z}_t^{(\ell)}))\mathbf{W}_t^{(\ell)}\Big)\|_F^2] \\
&\leq 2\,\mathbb{E}[\|\underbrace{\Big(\widetilde{\mathbf{D}}_{t-1}^{(\ell)} + [\widetilde{\mathbf{L}}^{(\ell)}]^\top (\mathbf{D}_t^{(\ell+1)} \circ \sigma'(\widetilde{\mathbf{Z}}_t^{(\ell)}))\mathbf{W}_t^{(\ell)} - [\widetilde{\mathbf{L}}^{(\ell)}]^\top (\mathbf{D}_{t-1}^{(\ell+1)} \circ \sigma'(\widetilde{\mathbf{Z}}_{t-1}^{(\ell)}))\mathbf{W}_{t-1}^{(\ell)}]\Big)}_{(A)} \\
&\quad \underbrace{- \Big(\mathbf{L}^\top (\mathbf{D}_t^{(\ell+1)} \circ \sigma'(\widetilde{\mathbf{Z}}_t^{(\ell)}))\mathbf{W}_t^{(\ell)}\Big)}_{(A)}\|_F^2] \\
&\quad + 2\,\mathbb{E}[\|\underbrace{\Big(\mathbf{L}^\top (\mathbf{D}_t^{(\ell+1)} \circ \sigma'(\widetilde{\mathbf{Z}}_t^{(\ell)}))\mathbf{W}_t^{(\ell)}\Big) - \Big(\mathbf{L}^\top (\mathbf{D}_t^{(\ell+1)} \circ \sigma'(\mathbf{Z}_t^{(\ell)}))\mathbf{W}_t^{(\ell)}\Big)}_{(B)}\|_F^2]
\end{aligned}
\tag{179}
$$

Let first take a closer look at term $(A)$. Let suppose $t \in \{EK+1, \ldots, EK+K\}$, where $E = t \mod K$ is the current epoch number and $K$ is the inner-loop size. By the previous lemma, term $(A)$ can be bounded by

$$
(A) \leq \sum_{t=EK+1}^{EK+K} \eta^2 \mathcal{O}\Big(|\mathbb{E}[\|\widetilde{\mathbf{L}}^{(\ell)}\|_F^2] - \|\mathbf{L}\|_F^2| \times \mathbb{E}[\|\nabla\widetilde{\mathcal{L}}(\boldsymbol{\theta}_{t-1})\|_F^2]\Big)
\tag{180}
$$

Then we take a closer look at term $(B)$.

$$
\begin{aligned}
&\mathbb{E}[\|\Big(\mathbf{L}^\top (\mathbf{D}_t^{(\ell+1)} \circ \sigma'(\widetilde{\mathbf{Z}}_t^{(\ell)}))\mathbf{W}_t^{(\ell)}\Big) - \Big(\mathbf{L}^\top (\mathbf{D}_t^{(\ell+1)} \circ \sigma'(\mathbf{Z}_t^{(\ell)}))\mathbf{W}_t^{(\ell)}\Big)\|_F^2] \\
&\leq B_{LA}^2 B_D^2 B_W^2 L_\sigma^2 \underbrace{\mathbb{E}[\|\widetilde{\mathbf{Z}}_t^{(\ell)} - \mathbf{Z}_t^{(\ell)}\|_F^2]}_{(C)}
\end{aligned}
\tag{181}
$$

The term $(C)$ can be decomposed as

$$
\begin{aligned}
&\mathbb{E}[\|\widetilde{\mathbf{Z}}_t^{(\ell)} - \mathbf{Z}_t^{(\ell)}\|_F^2] \\
&= \mathbb{E}[\|\widetilde{\mathbf{Z}}_t^{(\ell)} - \mathbf{L}\mathbf{H}_t^{(\ell-1)}\mathbf{W}_t^{(\ell)}\|_F^2] \\
&\leq 2\mathbb{E}[\|\widetilde{\mathbf{Z}}_t^{(\ell)} - \mathbf{L}\widetilde{\mathbf{H}}_t^{(\ell-1)}\mathbf{W}_t^{(\ell)}\|_F^2] + 2\mathbb{E}[\|\mathbf{L}\widetilde{\mathbf{H}}_t^{(\ell-1)}\mathbf{W}_t^{(\ell)} - \mathbf{L}\mathbf{H}_t^{(\ell-1)}\mathbf{W}_t^{(\ell)}\|_F^2] \\
&\leq 2\mathbb{E}[\|\widetilde{\mathbf{Z}}_t^{(\ell)} - \mathbf{L}\widetilde{\mathbf{H}}_t^{(\ell-1)}\mathbf{W}_t^{(\ell)}\|_F^2] + 2B_{LA}^2 B_W^2 C_\sigma^2 \mathbb{E}[\|\widetilde{\mathbf{Z}}_t^{(\ell-1)} - \mathbf{Z}_t^{(\ell-1)}\|_F^2]
\end{aligned}
\tag{182}
$$

By induction, we have

$$
\mathbb{E}[\|\widetilde{\mathbf{Z}}_t^{(\ell)} - \mathbf{Z}_t^{(\ell)}\|_F^2] \leq \mathcal{O}\Big(\underbrace{\mathbb{E}[\|\widetilde{\mathbf{Z}}_t^{(\ell)} - \mathbf{L}\widetilde{\mathbf{H}}_t^{(\ell-1)}\mathbf{W}_t^{(\ell)}\|_F^2]}_{(D)}\Big) + \ldots + \mathcal{O}\Big(\mathbb{E}[\|\widetilde{\mathbf{Z}}_t^{(1)} - \mathbf{L}\mathbf{X}\mathbf{W}_t^{(1)}\|_F^2]\Big)
\tag{183}
$$

The upper-bound for term $(D)$ is similar to one we have in the proof of Lemma 4.

$$
\begin{aligned}
&\|\mathbf{L}\widetilde{\mathbf{H}}_t^{(\ell-1)}\mathbf{W}_t^{(\ell)} - \widetilde{\mathbf{Z}}_t^{(\ell)}\|_F^2 \\
&= \|[\mathbf{L}\widetilde{\mathbf{H}}_t^{(\ell-1)}\mathbf{W}_t^{(\ell)} - \mathbf{L}\widetilde{\mathbf{H}}_{t-1}^{(\ell-1)}\mathbf{W}_{t-1}^{(\ell)}] + [\mathbf{L}\widetilde{\mathbf{H}}_{t-1}^{(\ell-1)}\mathbf{W}_{t-1}^{(\ell)} - \widetilde{\mathbf{Z}}_{t-1}^{(\ell)}] - [\widetilde{\mathbf{Z}}_t^{(\ell)} - \widetilde{\mathbf{Z}}_{t-1}^{(\ell)}]\|_F^2 \\
&= \|\mathbf{L}\widetilde{\mathbf{H}}_t^{(\ell-1)}\mathbf{W}_t^{(\ell)} - \mathbf{L}\widetilde{\mathbf{H}}_{t-1}^{(\ell-1)}\mathbf{W}_{t-1}^{(\ell)}\|_F^2 + \|\mathbf{L}\widetilde{\mathbf{H}}_{t-1}^{(\ell-1)}\mathbf{W}_{t-1}^{(\ell)} - \widetilde{\mathbf{Z}}_{t-1}^{(\ell)}\|_F^2 + \|\widetilde{\mathbf{Z}}_t^{(\ell)} - \widetilde{\mathbf{Z}}_{t-1}^{(\ell)}\|_F^2 \\
&\quad + 2\langle\mathbf{L}\widetilde{\mathbf{H}}_t^{(\ell-1)}\mathbf{W}_t^{(\ell)}, \mathbf{L}\widetilde{\mathbf{H}}_{t-1}^{(\ell-1)}\mathbf{W}_{t-1}^{(\ell)}\rangle - 2\langle\mathbf{L}\widetilde{\mathbf{H}}_t^{(\ell-1)}\mathbf{W}_t^{(\ell)}, \widetilde{\mathbf{Z}}_t^{(\ell)} - \widetilde{\mathbf{Z}}_{t-1}^{(\ell)}\rangle \\
&\quad - 2\langle\widetilde{\mathbf{Z}}_t^{(\ell)} - \widetilde{\mathbf{Z}}_{t-1}^{(\ell)}, \mathbf{L}\widetilde{\mathbf{H}}_{t-1}^{(\ell-1)}\mathbf{W}_{t-1}^{(\ell)}\rangle
\end{aligned}
\tag{184}
$$

Taking expectation condition on $\mathcal{F}_t$ and using

$$
\mathbb{E}[\widetilde{\mathbf{Z}}_t^{(\ell)} - \widetilde{\mathbf{Z}}_{t-1}^{(\ell)}|\mathcal{F}_t] = \mathbf{L}\widetilde{\mathbf{H}}_t^{(\ell-1)}\mathbf{W}_t^{(\ell)} - \mathbf{L}\widetilde{\mathbf{H}}_{t-1}^{(\ell-1)}\mathbf{W}_{t-1}^{(\ell)}
\tag{185}
$$

we have

$$\mathbb{E}[\|\mathbf{L}\widetilde{\mathbf{H}}_t^{(\ell-1)}\mathbf{W}_t^{(\ell)} - \widetilde{\mathbf{Z}}_t^{(\ell)}\|_{\mathrm{F}}^2|\mathcal{F}_t]$$
$$= \|\mathbf{L}\widetilde{\mathbf{H}}_{t-1}^{(\ell-1)}\mathbf{W}_{t-1}^{(\ell)} - \widetilde{\mathbf{Z}}_{t-1}^{(\ell)}\|_{\mathrm{F}}^2 + \mathbb{E}[\|\widetilde{\mathbf{Z}}_t^{(\ell)} - \widetilde{\mathbf{Z}}_{t-1}^{(\ell)}\|_{\mathrm{F}}^2|\mathcal{F}_t] - \|\mathbf{L}\widetilde{\mathbf{H}}_t^{(\ell-1)}\mathbf{W}_t^{(\ell)} - \mathbf{L}\widetilde{\mathbf{H}}_{t-1}^{(\ell-1)}\mathbf{W}_{t-1}^{(\ell)}\|_{\mathrm{F}}^2 \tag{186}$$

Take expectation over $\mathcal{F}_t$ we have

$$\mathbb{E}[\|\mathbf{L}\widetilde{\mathbf{H}}_t^{(\ell-1)}\mathbf{W}_t^{(\ell)} - \widetilde{\mathbf{Z}}_t^{(\ell)}\|_{\mathrm{F}}^2]$$
$$= \|\mathbf{L}\widetilde{\mathbf{H}}_{t-1}^{(\ell-1)}\mathbf{W}_{t-1}^{(\ell)} - \widetilde{\mathbf{Z}}_{t-1}^{(\ell)}\|_{\mathrm{F}}^2 + \mathbb{E}[\|\widetilde{\mathbf{Z}}_t^{(\ell)} - \widetilde{\mathbf{Z}}_{t-1}^{(\ell)}\|_{\mathrm{F}}^2] - \|\mathbf{L}\widetilde{\mathbf{H}}_t^{(\ell-1)}\mathbf{W}_t^{(\ell)} - \mathbf{L}\widetilde{\mathbf{H}}_{t-1}^{(\ell-1)}\mathbf{W}_{t-1}^{(\ell)}\|_{\mathrm{F}}^2 \tag{187}$$

Let suppose $t \in \{EK+1, \ldots, EK+K\}$. Then we can denote $t = EK + k$ for some $k \le K$ such that

$$\mathbb{E}[\|\mathbf{L}\widetilde{\mathbf{H}}_t^{(\ell-1)}\mathbf{W}_t^{(\ell)} - \widetilde{\mathbf{Z}}_t^{(\ell)}\|_{\mathrm{F}}^2]$$
$$= \mathbb{E}[\|\mathbf{L}\widetilde{\mathbf{H}}_{EK+k}^{(\ell-1)}\mathbf{W}_{EK+k}^{(\ell)} - \widetilde{\mathbf{Z}}_{EK+k}^{(\ell)}\|_{\mathrm{F}}^2]$$
$$= \|\mathbf{L}\widetilde{\mathbf{H}}_{EK}^{(\ell-1)}\mathbf{W}_{EK}^{(\ell)} - \widetilde{\mathbf{Z}}_{EK}^{(\ell)}\|_{\mathrm{F}}^2 + \sum_{t=EK+1}^{EK+K}\mathbb{E}[\|\widetilde{\mathbf{Z}}_t^{(\ell)} - \widetilde{\mathbf{Z}}_{t-1}^{(\ell)}\|_{\mathrm{F}}^2] - \|\mathbf{L}\widetilde{\mathbf{H}}_t^{(\ell-1)}\mathbf{W}_t^{(\ell)} - \mathbf{L}\widetilde{\mathbf{H}}_{t-1}^{(\ell-1)}\mathbf{W}_{t-1}^{(\ell)}\|_{\mathrm{F}}^2$$
$$\le \sum_{t=EK+1}^{EK+K}\mathcal{O}\Big(|\mathbb{E}[\|\widetilde{\mathbf{L}}^{(\ell)}\|_{\mathrm{F}}^2] - \|\mathbf{L}\|_{\mathrm{F}}^2| \times \mathbb{E}[\|\mathbf{W}_t^{(\ell)} - \mathbf{W}_{t-1}^{(\ell)}\|_{\mathrm{F}}^2]\Big) \tag{188}$$

Plugging $(D)$ to $(C)$ and $(C)$ to $(B)$, we have

$$(B) \le \sum_{j=1}^{\ell}\sum_{t=EK+1}^{EK+K}\mathcal{O}\Big(|\mathbb{E}[\|\widetilde{\mathbf{L}}^{(j)}\|_{\mathrm{F}}^2] - \|\mathbf{L}\|_{\mathrm{F}}^2| \times \mathbb{E}[\|\mathbf{W}_t^{(j)} - \mathbf{W}_{t-1}^{(j)}\|_{\mathrm{F}}^2]\Big) \tag{189}$$

Combing with term $(A)$ we have

$$\mathbb{E}[\|\nabla_H\widetilde{f}^{(\ell)}(\mathbf{D}_t^{(\ell+1)}, \widetilde{\mathbf{H}}_t^{(\ell-1)}, \mathbf{W}_t^{(\ell)}) - \nabla_H f^{(\ell)}(\mathbf{D}_t^{(\ell+1)}, \mathbf{H}_t^{(\ell-1)}, \mathbf{W}_t^{(\ell)})\|_{\mathrm{F}}^2]$$
$$\le \sum_{\ell=1}^{L}\sum_{t=EK+1}^{EK+K}\mathcal{O}\Big(|\mathbb{E}[\|\widetilde{\mathbf{L}}^{(\ell)}\|_{\mathrm{F}}^2] - \|\mathbf{L}\|_{\mathrm{F}}^2| \times \mathbb{E}[\|\mathbf{W}_t^{(\ell)} - \mathbf{W}_{t-1}^{(\ell)}\|_{\mathrm{F}}^2]\Big)$$
$$= \sum_{t=EK+1}^{EK+K}\eta^2\mathcal{O}\Big(|\mathbb{E}[\|\widetilde{\mathbf{L}}^{(\ell)}\|_{\mathrm{F}}^2] - \|\mathbf{L}\|_{\mathrm{F}}^2| \times \mathbb{E}[\|\nabla\widetilde{\mathcal{L}}(\boldsymbol{\theta}_{t-1})\|_{\mathrm{F}}^2]\Big) \tag{190}$$

$$\square$$

In the following lemma, we derive the upper-bound on the difference of the gradient with respect to the weight matrix at each graph convolutional layer. Suppose the input node embedding matrix for the $\ell$th GCN layer is defined as $\widetilde{\mathbf{H}}_t^{(\ell-1)}$, the gradient calculated for the $\ell$th weight matrix in SGCN++ is defined as

$$\nabla_W\widetilde{f}^{(\ell)}(\mathbf{D}_t^{(\ell+1)}, \widetilde{\mathbf{H}}_t^{(\ell-1)}, \mathbf{W}_t^{(\ell)})$$
$$= \widetilde{\mathbf{G}}_{t-1}^{(\ell)} + [\widetilde{\mathbf{L}}^{(\ell)}\widetilde{\mathbf{H}}_t^{(\ell-1)}]^\top(\mathbf{D}_t^{(\ell+1)} \circ \sigma'(\widetilde{\mathbf{Z}}_t^{(\ell)})) - [\widetilde{\mathbf{L}}^{(\ell)}\widetilde{\mathbf{H}}_{t-1}^{(\ell-1)}]^\top(\mathbf{D}_{t-1}^{(\ell+1)} \circ \sigma'(\widetilde{\mathbf{Z}}_{t-1}^{(\ell)})) \tag{191}$$

and the backward propagation for the $\ell$th layer in FullGCN is defined as

$$\nabla_W f^{(\ell)}(\mathbf{D}_t^{(\ell+1)}, \widetilde{\mathbf{H}}_t^{(\ell-1)}, \mathbf{W}_t^{(\ell)}) = [\mathbf{L}\widetilde{\mathbf{H}}_t^{(\ell-1)}]^\top(\mathbf{D}_t^{(\ell+1)} \circ \sigma'(\widetilde{\mathbf{Z}}_t^{(\ell)})) \tag{192}$$

**Lemma 9.** *Let suppose $t \in \{EK+1, \ldots, EK+K\}$. The upper-bound on the difference of the gradient with respect to the $\ell$th graph convolutional layer given the same input $\mathbf{D}_t^{(\ell+1)}$ and $\widetilde{\mathbf{H}}_t^{(\ell-1)}$*

*is defined as*

$$\|\nabla_W \widetilde{f}^{(\ell)}(\mathbf{D}_t^{(\ell+1)}, \widetilde{\mathbf{H}}_t^{(\ell-1)}, \mathbf{W}_t^{(\ell)}) - \nabla_W f^{(\ell)}(\mathbf{D}_t^{(\ell+1)}, \widetilde{\mathbf{H}}_t^{(\ell-1)}, \mathbf{W}_t^{(\ell)})\|_{\mathrm{F}}^2$$
$$\leq \sum_{t=EK+1}^{EK+K} \eta^2 \mathcal{O}\Big(|\mathbb{E}[\|\widetilde{\mathbf{L}}^{(\ell)}\|_{\mathrm{F}}^2] - \|\mathbf{L}\|_{\mathrm{F}}^2| \times \mathbb{E}[\|\nabla\widetilde{\mathcal{L}}(\boldsymbol{\theta}_{t-1})\|_{\mathrm{F}}^2]\Big) \tag{193}$$

*Proof.* To simplify the presentation, let us denote $\widetilde{\mathbf{G}}_t^{(\ell)} = \nabla_W \widetilde{f}^{(\ell)}(\mathbf{D}_t^{(\ell+1)}, \widetilde{\mathbf{H}}_t^{(\ell-1)}, \mathbf{W}_t^{(\ell)})$. Then, by definition, we have

$$\widetilde{\mathbf{G}}_t^{(\ell)} = \widetilde{\mathbf{G}}_{t-1}^{(\ell)} + [\widetilde{\mathbf{L}}^{(\ell)}\widetilde{\mathbf{H}}_t^{(\ell-1)}]^\top(\mathbf{D}_t^{(\ell+1)} \circ \sigma'(\widetilde{\mathbf{Z}}_t^{(\ell)})) - [\widetilde{\mathbf{L}}^{(\ell)}\widetilde{\mathbf{H}}_{t-1}^{(\ell-1)}]^\top(\mathbf{D}_{t-1}^{(\ell+1)} \circ \sigma'(\widetilde{\mathbf{Z}}_{t-1}^{(\ell)})) \tag{194}$$

Therefore, we know that

$$\|[\mathbf{L}\widetilde{\mathbf{H}}_t^{(\ell-1)}]^\top(\mathbf{D}_t^{(\ell+1)} \circ \sigma'(\widetilde{\mathbf{Z}}_t^{(\ell)})) - \widetilde{\mathbf{G}}_t^{(\ell)}\|_{\mathrm{F}}^2$$
$$= \Big\| \Big[[\mathbf{L}\widetilde{\mathbf{H}}_t^{(\ell-1)}]^\top(\mathbf{D}_t^{(\ell+1)} \circ \sigma'(\widetilde{\mathbf{Z}}_t^{(\ell)})) - [\mathbf{L}\widetilde{\mathbf{H}}_{t-1}^{(\ell-1)}]^\top(\mathbf{D}_{t-1}^{(\ell+1)} \circ \sigma'(\widetilde{\mathbf{Z}}_{t-1}^{(\ell)}))\Big]$$
$$+ \Big[[\mathbf{L}\widetilde{\mathbf{H}}_{t-1}^{(\ell-1)}]^\top(\mathbf{D}_{t-1}^{(\ell+1)} \circ \sigma'(\widetilde{\mathbf{Z}}_{t-1}^{(\ell)})) - \widetilde{\mathbf{G}}_{t-1}^{(\ell)}\Big] - \Big[\widetilde{\mathbf{G}}_t^{(\ell)} - \widetilde{\mathbf{G}}_{t-1}^{(\ell)}\Big]\Big\|_{\mathrm{F}}^2$$
$$\leq \Big\| \underbrace{[\mathbf{L}\widetilde{\mathbf{H}}_t^{(\ell-1)}]^\top(\mathbf{D}_t^{(\ell+1)} \circ \sigma'(\widetilde{\mathbf{Z}}_t^{(\ell)})) - [\mathbf{L}\widetilde{\mathbf{H}}_{t-1}^{(\ell-1)}]^\top(\mathbf{D}_{t-1}^{(\ell+1)} \circ \sigma'(\widetilde{\mathbf{Z}}_{t-1}^{(\ell)}))}_{(A_1)} \Big\|_{\mathrm{F}}^2 \tag{195}$$
$$+ \| \underbrace{[\mathbf{L}\widetilde{\mathbf{H}}_{t-1}^{(\ell-1)}]^\top(\mathbf{D}_{t-1}^{(\ell+1)} \circ \sigma'(\widetilde{\mathbf{Z}}_{t-1}^{(\ell)})) - \widetilde{\mathbf{G}}_{t-1}^{(\ell)}}_{A_2} \|_{\mathrm{F}}^2 + \| \underbrace{\widetilde{\mathbf{G}}_t^{(\ell)} - \widetilde{\mathbf{G}}_{t-1}^{(\ell)}}_{A_3} \|_{\mathrm{F}}^2$$
$$+ 2\langle A_1, A_2\rangle - 2\langle A_1, A_3\rangle - 2\langle A_2, A_3\rangle$$

Taking expectation condition on $\mathcal{F}_t$ on both side, and using the fact that

$$\mathbb{E}[\widetilde{\mathbf{G}}_t^{(\ell)} - \widetilde{\mathbf{G}}_{t-1}^{(\ell)}|\mathcal{F}_t] = \mathbb{E}[[\widetilde{\mathbf{L}}^{(\ell)}\widetilde{\mathbf{H}}_t^{(\ell-1)}]^\top(\mathbf{D}_t^{(\ell+1)} \circ \sigma'(\widetilde{\mathbf{Z}}_t^{(\ell)})) - [\widetilde{\mathbf{L}}^{(\ell)}\widetilde{\mathbf{H}}_{t-1}^{(\ell-1)}]^\top(\mathbf{D}_{t-1}^{(\ell+1)} \circ \sigma'(\widetilde{\mathbf{Z}}_{t-1}^{(\ell)}))|\mathcal{F}_t]$$
$$= [\mathbf{L}\widetilde{\mathbf{H}}_t^{(\ell-1)}]^\top(\mathbf{D}_t^{(\ell+1)} \circ \sigma'(\widetilde{\mathbf{Z}}_t^{(\ell)})) - [\mathbf{L}\widetilde{\mathbf{H}}_{t-1}^{(\ell-1)}]^\top(\mathbf{D}_{t-1}^{(\ell+1)} \circ \sigma'(\widetilde{\mathbf{Z}}_{t-1}^{(\ell)})) \tag{196}$$

we have the following inequality holds

$$\mathbb{E}[\|[\mathbf{L}\widetilde{\mathbf{H}}_t^{(\ell)}]^\top(\mathbf{D}_t^{(\ell+1)} \circ \sigma'(\widetilde{\mathbf{Z}}_t^{(\ell-1)})) - \widetilde{\mathbf{G}}_t^{(\ell)}\|_{\mathrm{F}}^2|\mathcal{F}_t]$$
$$\leq \|[\mathbf{L}\widetilde{\mathbf{H}}_{t-1}^{(\ell-1)}]^\top(\mathbf{D}_{t-1}^{(\ell+1)} \circ \sigma'(\widetilde{\mathbf{Z}}_{t-1}^{(\ell)})) - \widetilde{\mathbf{G}}_{t-1}^{(\ell)}\|_{\mathrm{F}}^2 + \mathbb{E}[\|\widetilde{\mathbf{G}}_t^{(\ell)} - \widetilde{\mathbf{G}}_{t-1}^{(\ell)}\|_{\mathrm{F}}^2|\mathcal{F}_t] \tag{197}$$
$$- \|[\mathbf{L}\widetilde{\mathbf{H}}_t^{(\ell-1)}]^\top(\mathbf{D}_t^{(\ell+1)} \circ \sigma'(\widetilde{\mathbf{Z}}_t^{(\ell)})) - [\mathbf{L}\widetilde{\mathbf{H}}_{t-1}^{(\ell-1)}]^\top(\mathbf{D}_{t-1}^{(\ell+1)} \circ \sigma'(\widetilde{\mathbf{Z}}_{t-1}^{(\ell)}))\mathbf{W}_{t-1}^{(\ell)}\|_{\mathrm{F}}^2$$

Then, taking expectation over $\mathcal{F}_t$, we have

$$\mathbb{E}[\|[\mathbf{L}\widetilde{\mathbf{H}}_t^{(\ell-1)}]^\top(\mathbf{D}_t^{(\ell+1)} \circ \sigma'(\widetilde{\mathbf{Z}}_t^{(\ell)})) - \widetilde{\mathbf{G}}_t^{(\ell)}\|_{\mathrm{F}}^2]$$
$$\leq \mathbb{E}[\|[\mathbf{L}\widetilde{\mathbf{H}}_{t-1}^{(\ell-1)}]^\top(\mathbf{D}_{t-1}^{(\ell+1)} \circ \sigma'(\widetilde{\mathbf{Z}}_{t-1}^{(\ell)}))\mathbf{W}_{t-1}^{(\ell)} - \widetilde{\mathbf{G}}_{t-1}^{(\ell)}\|_{\mathrm{F}}^2] + \mathbb{E}[\|\widetilde{\mathbf{G}}_t^{(\ell)} - \widetilde{\mathbf{G}}_{t-1}^{(\ell)}\|_{\mathrm{F}}^2] \tag{198}$$
$$- [\|[\mathbf{L}\widetilde{\mathbf{H}}_t^{(\ell-1)}]^\top(\mathbf{D}_t^{(\ell+1)} \circ \sigma'(\widetilde{\mathbf{Z}}_t^{(\ell)})) - [\mathbf{L}\widetilde{\mathbf{H}}_{t-1}^{(\ell-1)}]^\top(\mathbf{D}_{t-1}^{(\ell+1)} \circ \sigma'(\widetilde{\mathbf{Z}}_{t-1}^{(\ell)}))\|_{\mathrm{F}}^2]$$

Let suppose $t \in \{EK+1, \ldots, EK+K\}$. Then we can denote $t = EK+k$ for some $k \le K$ such that

$$\mathbb{E}[\|[\mathbf{L}\widetilde{\mathbf{H}}_t^{(\ell-1)}]^\top(\mathbf{D}_t^{(\ell+1)} \circ \sigma'(\widetilde{\mathbf{Z}}_t^{(\ell)})) - \widetilde{\mathbf{G}}_t^{(\ell)}\|_F^2]$$

$$\mathbb{E}[\|[\mathbf{L}\widetilde{\mathbf{H}}_{EK+k}^{(\ell-1)}]^\top(\mathbf{D}_{EK+k}^{(\ell+1)} \circ \sigma'(\widetilde{\mathbf{Z}}_{EK+k}^{(\ell)})) - \widetilde{\mathbf{G}}_{EK+k}^{(\ell)}\|_F^2]$$

$$\le \mathbb{E}[\|[\mathbf{L}\widetilde{\mathbf{H}}_{EK}^{(\ell-1)}]^\top(\mathbf{D}_{EK}^{(\ell+1)} \circ \sigma'(\widetilde{\mathbf{Z}}_{EK}^{(\ell)})) - \widetilde{\mathbf{G}}_{EK}^{(\ell)}\|_F^2] + \sum_{t=EK+1}^{EK+K}\left(\mathbb{E}[\|\widetilde{\mathbf{G}}_t^{(\ell)} - \widetilde{\mathbf{G}}_{t-1}^{(\ell)}\|_F^2]\right.$$

$$\left. - [\|[\mathbf{L}\widetilde{\mathbf{H}}_t^{(\ell-1)}]^\top(\mathbf{D}_t^{(\ell+1)} \circ \sigma'(\widetilde{\mathbf{Z}}_t^{(\ell)})) - [\mathbf{L}\widetilde{\mathbf{H}}_{t-1}^{(\ell-1)}]^\top(\mathbf{D}_{t-1}^{(\ell+1)} \circ \sigma'(\widetilde{\mathbf{Z}}_{t-1}^{(\ell)}))\|_F^2]\right)$$

$$\le \mathbb{E}[\|\mathbf{L}^\top(\mathbf{D}_{EK}^{(\ell+1)} \circ \sigma'(\widetilde{\mathbf{Z}}_{EK}^{(\ell)}))\mathbf{W}_{EK}^{(\ell)} - \widetilde{\mathbf{D}}_{EK}^{(\ell)}\|_F^2]$$

$$+ \sum_{t=EK+1}^{EK+K}\left(\mathbb{E}[\|[\widetilde{\mathbf{L}}^{(\ell)}\widetilde{\mathbf{H}}_t^{(\ell-1)}]^\top(\mathbf{D}_t^{(\ell+1)} \circ \sigma'(\widetilde{\mathbf{Z}}_t^{(\ell)})) - [\widetilde{\mathbf{L}}^{(\ell)}\widetilde{\mathbf{H}}_{t-1}^{(\ell-1)}]^\top(\mathbf{D}_{t-1}^{(\ell+1)} \circ \sigma'(\widetilde{\mathbf{Z}}_{t-1}^{(\ell)}))\|_F^2]\right.$$

$$\left. - \mathbb{E}[\|[\mathbf{L}\widetilde{\mathbf{H}}_t^{(\ell-1)}]^\top(\mathbf{D}_t^{(\ell+1)} \circ \sigma'(\widetilde{\mathbf{Z}}_t^{(\ell)})) - [\mathbf{L}\widetilde{\mathbf{H}}_{t-1}^{(\ell-1)}]^\top(\mathbf{D}_{t-1}^{(\ell+1)} \circ \sigma'(\widetilde{\mathbf{Z}}_{t-1}^{(\ell)}))\|_F^2]\right)$$

$$\tag{199}$$

Knowing that we are taking full-batch gradient descent when $(t \mod K) = 0$, we have

$$\mathbb{E}[\|\mathbf{L}^\top(\mathbf{D}_t^{(\ell+1)} \circ \sigma'(\widetilde{\mathbf{Z}}_t^{(\ell)}))\mathbf{W}_t^{(\ell)} - \widetilde{\mathbf{D}}_t^{(\ell)}\|_F^2]$$

$$\le \sum_{t=EK+1}^{EK+K}\left(\left|\mathbb{E}[\|\widetilde{\mathbf{L}}^{(\ell)}\|_F^2] - \|\mathbf{L}\|_F^2\right|\right) \times \underbrace{\mathbb{E}[\|[\widetilde{\mathbf{H}}_t^{(\ell-1)}]^\top(\mathbf{D}_t^{(\ell+1)} \circ \sigma'(\widetilde{\mathbf{Z}}_t^{(\ell)})) - [\widetilde{\mathbf{H}}_{t-1}^{(\ell-1)}]^\top(\mathbf{D}_{t-1}^{(\ell+1)} \circ \sigma'(\widetilde{\mathbf{Z}}_{t-1}^{(\ell)}))\|_F^2]}_{(B)}$$

$$\tag{200}$$

Let take closer look at term $(B)$.

$$\mathbb{E}[\|[\widetilde{\mathbf{H}}_t^{(\ell-1)}]^\top(\mathbf{D}_t^{(\ell+1)} \circ \sigma'(\widetilde{\mathbf{Z}}_t^{(\ell)})) - [\widetilde{\mathbf{H}}_{t-1}^{(\ell-1)}]^\top(\mathbf{D}_{t-1}^{(\ell+1)} \circ \sigma'(\widetilde{\mathbf{Z}}_{t-1}^{(\ell)}))\|_F^2]$$

$$\le 3(B_D^2 C_\sigma^4 + B_H^2 B_D^2 L_\sigma^2)\underbrace{\mathbb{E}[\|\widetilde{\mathbf{Z}}_t^{(\ell-1)} - \widetilde{\mathbf{Z}}_{t-1}^{(\ell-1)}\|_F^2]}_{(C_1)} + 3B_H^2 C_\sigma^2 \underbrace{\mathbb{E}[\|\mathbf{D}_t^{(\ell+1)} - \mathbf{D}_{t-1}^{(\ell+1)}\|_F^2]}_{(C_2)} \tag{201}$$

For term $(C_1)$ by definition we know

$$\|\mathbf{D}_t^{(\ell+1)} - \mathbf{D}_{t-1}^{(\ell+1)}\|_F^2 = \|\left(\mathbf{L}^\top(\mathbf{D}_t^{(\ell+2)} \circ \sigma'(\mathbf{Z}_t^{(\ell+1)}))\mathbf{W}_t^{(\ell+1)}\right) - \left(\mathbf{L}^\top(\mathbf{D}_{t-1}^{(\ell+2)} \circ \sigma'(\mathbf{Z}_{t-1}^{(\ell+1)}))\mathbf{W}_{t-1}^{(\ell+1)}\right)\|_F^2$$

$$\le 3\|\left(\mathbf{L}^\top(\mathbf{D}_t^{(\ell+2)} \circ \sigma'(\mathbf{Z}_t^{(\ell+1)}))\mathbf{W}_t^{(\ell+1)}\right) - \left(\mathbf{L}^\top(\mathbf{D}_{t-1}^{(\ell+2)} \circ \sigma'(\mathbf{Z}_t^{(\ell+1)}))\mathbf{W}_t^{(\ell+1)}\right)\|_F^2$$

$$+ 3\|\left(\mathbf{L}^\top(\mathbf{D}_{t-1}^{(\ell+2)} \circ \sigma'(\mathbf{Z}_t^{(\ell+1)}))\mathbf{W}_t^{(\ell+1)}\right) - \left(\mathbf{L}^\top(\mathbf{D}_{t-1}^{(\ell+2)} \circ \sigma'(\mathbf{Z}_{t-1}^{(\ell+1)}))\mathbf{W}_t^{(\ell+1)}\right)\|_F^2$$

$$+ 3\|\left(\mathbf{L}^\top(\mathbf{D}_{t-1}^{(\ell+2)} \circ \sigma'(\mathbf{Z}_{t-1}^{(\ell+1)}))\mathbf{W}_t^{(\ell+1)}\right) - \left(\mathbf{L}^\top(\mathbf{D}_{t-1}^{(\ell+2)} \circ \sigma'(\mathbf{Z}_{t-1}^{(\ell+1)}))\mathbf{W}_{t-1}^{(\ell+1)}\right)\|_F^2$$

$$\le \mathcal{O}(\|\mathbf{D}_t^{(\ell+2)} - \mathbf{D}_{t-1}^{(\ell+2)}\|_F^2) + \mathcal{O}(\|\mathbf{Z}_t^{(\ell+1)} - \mathbf{Z}_{t-1}^{(\ell+1)}\|_F^2) + \mathcal{O}(\|\mathbf{W}_t^{(\ell+1)} - \mathbf{W}_{t-1}^{(\ell+1)}\|_F^2) \tag{202}$$

By induction, we have

$$\|\mathbf{D}_t^{(\ell+1)} - \mathbf{D}_{t-1}^{(\ell+1)}\|_F^2 \le \underbrace{\mathcal{O}(\|\mathbf{D}_t^{(L+1)} - \mathbf{D}_{t-1}^{(L+1)}\|_F^2)}_{(D_1)} + \underbrace{\mathcal{O}(\|\mathbf{Z}_t^{(\ell+1)} - \mathbf{Z}_{t-1}^{(\ell+1)}\|_F^2)}_{(D_2)} + \ldots + \mathcal{O}(\|\mathbf{Z}_t^{(L)} - \mathbf{Z}_{t-1}^{(L)}\|_F^2)$$

$$+ \mathcal{O}(\|\mathbf{W}_t^{(\ell+1)} - \mathbf{W}_{t-1}^{(\ell+1)}\|_F^2) + \ldots + \mathcal{O}(\|\mathbf{W}_t^{(L)} - \mathbf{W}_{t-1}^{(L)}\|_F^2) \tag{203}$$

For term $(D_1)$ we have

$$\|\mathbf{D}_t^{(L+1)} - \mathbf{D}_{t-1}^{(L+1)}\|_F^2 = \|\frac{\partial\mathcal{L}(\boldsymbol{\theta}_t)}{\partial\mathbf{W}_t^{(L)}} - \frac{\partial\mathcal{L}(\boldsymbol{\theta}_{t-1})}{\partial\mathbf{W}_{t-1}^{(L)}}\|_F^2$$

$$\le L_{\text{loss}}^2 C_\sigma^2 \|\mathbf{Z}_t^{(L)} - \mathbf{Z}_{t-1}^{(L)}\|_F^2 \tag{204}$$

For term $(D_2)$ we have

$$
\begin{aligned}
\|\mathbf{Z}_t^{(\ell+1)} - \mathbf{Z}_{t-1}^{(\ell+1)}\|_F^2 &\leq C_\sigma^2 \|\mathbf{LH}_t^{(\ell)}\mathbf{W}_t^{(\ell+1)} - \mathbf{LH}_{t-1}^{(\ell)}\mathbf{W}_{t-1}^{(\ell+1)}\|_F^2 \\
&\leq C_\sigma^2 B_{LA}^2 \|\mathbf{H}_t^{(\ell)}\mathbf{W}_t^{(\ell+1)} - \mathbf{H}_{t-1}^{(\ell)}\mathbf{W}_t^{(\ell+1)} + \mathbf{H}_{t-1}^{(\ell)}\mathbf{W}_t^{(\ell+1)} - \mathbf{H}_{t-1}^{(\ell)}\mathbf{W}_{t-1}^{(\ell+1)}\|_F^2 \\
&\leq 2C_\sigma^2 B_{LA}^2 \|\mathbf{H}_t^{(\ell)}\mathbf{W}_t^{(\ell+1)} - \mathbf{H}_{t-1}^{(\ell)}\mathbf{W}_t^{(\ell+1)}\|_F^2 + 2C_\sigma^2 B_{LA}^2 \|\mathbf{H}_{t-1}^{(\ell)}\mathbf{W}_t^{(\ell+1)} - \mathbf{H}_{t-1}^{(\ell)}\mathbf{W}_{t-1}^{(\ell+1)}\|_F^2 \\
&\leq 2C_\sigma^4 B_{LA}^2 B_W^2 \|\mathbf{Z}_t^{(\ell)} - \mathbf{Z}_{t-1}^{(\ell)}\|_F^2 + 2C_\sigma^2 B_{LA}^2 B_H^2 \|\mathbf{W}_t^{(\ell+1)} - \mathbf{W}_{t-1}^{(\ell+1)}\|_F^2
\end{aligned}
\tag{205}
$$

By induction we have

$$
\|\mathbf{Z}_t^{(\ell+1)} - \mathbf{Z}_{t-1}^{(\ell+1)}\|_F^2 \leq \mathcal{O}(\|\mathbf{W}_t^{(\ell+1)} - \mathbf{W}_{t-1}^{(\ell+1)}\|_F^2) + \ldots + \mathcal{O}(\|\mathbf{W}_t^{(1)} - \mathbf{W}_{t-1}^{(1)}\|_F^2)
\tag{206}
$$

For term $(C_2)$ by definition we have

$$
\begin{aligned}
\mathbb{E}[\|\widetilde{\mathbf{Z}}_t^{(\ell)} - \widetilde{\mathbf{Z}}_{t-1}^{(\ell)}\|_F^2] &= \mathbb{E}[\|\widetilde{\mathbf{L}}^{(\ell)}\widetilde{\mathbf{H}}_t^{(\ell-1)}\mathbf{W}_t^{(\ell)} - \widetilde{\mathbf{L}}^{(\ell)}\widetilde{\mathbf{H}}_{t-1}^{(\ell-1)}\mathbf{W}_{t-1}^{(\ell)}\|_F^2] \\
&\leq 2B_{LA}^2 B_{W\ell}^2 C_\sigma^2 \mathbb{E}[\|\widetilde{\mathbf{Z}}_t^{(\ell-1)} - \widetilde{\mathbf{Z}}_{t-1}^{(\ell-1)}\|_F^2] + 2B_{LA}^2 B_H^2 \mathbb{E}[\|\mathbf{W}_t^{(\ell)} - \mathbf{W}_{t-1}^{(\ell)}\|_F^2]
\end{aligned}
\tag{207}
$$

By induction we have

$$
\mathbb{E}[\|\widetilde{\mathbf{Z}}_t^{(\ell)} - \widetilde{\mathbf{Z}}_{t-1}^{(\ell)}\|_F^2] \leq \mathcal{O}(\mathbb{E}[\|\mathbf{W}_t^{(\ell)} - \mathbf{W}_{t-1}^{(\ell)}\|_F^2]) + \ldots + \mathcal{O}(\mathbb{E}[\|\mathbf{W}_t^{(1)} - \mathbf{W}_{t-1}^{(1)}\|_F^2])
\tag{208}
$$

Plugging $(D_1), (D_2)$ to $(C_2)$ and $C_1, (C_2)$ to $(B)$, we have

$$
\mathbb{E}[\|[\mathbf{L}\widetilde{\mathbf{H}}_t^{(\ell-1)}]^\top (\mathbf{D}_t^{(\ell+1)} \circ \sigma'(\widetilde{\mathbf{Z}}_t^{(\ell)})) - \widetilde{\mathbf{G}}_t^{(\ell)}\|_F^2] \leq \sum_{t=EK+1}^{EK+K} \eta^2 \mathcal{O}\Big(|\mathbb{E}[\|\widetilde{\mathbf{L}}^{(\ell)}\|_F^2] - \|\mathbf{L}\|_F^2| \times \mathbb{E}[\|\nabla\widetilde{\mathcal{L}}(\boldsymbol{\theta}_{t-1})\|_F^2]\Big)
\tag{209}
$$

$$\square$$

Using the previous lemma, we provide the upper-bound of Eq. 155, which is one of the three key factors that affect the mean-square error of stochastic gradient at the $\ell$th layer.

**Lemma 10.** *Let suppose $t \in \{EK + 1, \ldots, EK + K\}$. The upper-bound on the difference of the gradient with respect to the weight of the $\ell$th graph convolutional layer given the same input $\mathbf{D}_t^{(\ell+1)}$ but different input $\widetilde{\mathbf{H}}_t^{(\ell-1)}, \mathbf{H}_t^{(\ell-1)}$ is defined as*

$$
\begin{aligned}
&\mathbb{E}[\|\nabla_W \widetilde{f}^{(\ell)}(\mathbf{D}_t^{(\ell+1)}, \widetilde{\mathbf{H}}_t^{(\ell-1)}, \mathbf{W}_t^{(\ell)}) - \nabla_W f^{(\ell)}(\mathbf{D}_t^{(\ell+1)}, \mathbf{H}_t^{(\ell-1)}, \mathbf{W}_t^{(\ell)})\|_F^2] \\
&\leq \sum_{t=EK+1}^{EK+K} \eta^2 \mathcal{O}\Big(|\mathbb{E}[\|\widetilde{\mathbf{L}}^{(\ell)}\|_F^2] - \|\mathbf{L}\|_F^2| \times \mathbb{E}[\|\nabla\widetilde{\mathcal{L}}(\boldsymbol{\theta}_{t-1})\|_F^2]\Big)
\end{aligned}
\tag{210}
$$

*Proof.* For the gradient w.r.t. the weight matrices, we have

$$
\mathbb{E}[\|\nabla_W \widetilde{f}^{(\ell)}(\mathbf{D}_t^{(\ell+1)}, \widetilde{\mathbf{H}}_t^{(\ell-1)}, \mathbf{W}_t^{(\ell)}) - \nabla_W f^{(\ell)}(\mathbf{D}_t^{(\ell+1)}, \mathbf{H}_t^{(\ell-1)}, \mathbf{W}_t^{(\ell)})\|_{\mathrm{F}}^2]
$$

$$
= \mathbb{E}[\|\Big(\widetilde{\mathbf{G}}_{t-1}^{(\ell)} + [\widetilde{\mathbf{L}}^{(\ell)}\widetilde{\mathbf{H}}_t^{(\ell-1)}]^\top (\mathbf{D}_t^{(\ell+1)} \circ \sigma'(\widetilde{\mathbf{Z}}_t^{(\ell)})) - [\widetilde{\mathbf{L}}^{(\ell)}\widetilde{\mathbf{H}}_{t-1}^{(\ell-1)}]^\top (\mathbf{D}_{t-1}^{(\ell+1)} \circ \sigma'(\widetilde{\mathbf{Z}}_{t-1}^{(\ell)}))\Big)
$$

$$
- \Big([\mathbf{L}\mathbf{H}_t^{(\ell-1)}]^\top (\mathbf{D}_t^{(\ell+1)} \circ \sigma'(\mathbf{Z}_t^{(\ell)}))\Big)\|_{\mathrm{F}}^2]
$$

$$
\leq 2\,\mathbb{E}[\|\underbrace{\Big(\widetilde{\mathbf{G}}_{t-1}^{(\ell)} + [\widetilde{\mathbf{L}}^{(\ell)}\widetilde{\mathbf{H}}_t^{(\ell-1)}]^\top (\mathbf{D}_t^{(\ell+1)} \circ \sigma'(\widetilde{\mathbf{Z}}_t^{(\ell)})) - [\widetilde{\mathbf{L}}^{(\ell)}\widetilde{\mathbf{H}}_t^{(\ell-1)}]^\top (\mathbf{D}_{t-1}^{(\ell+1)} \circ \sigma'(\widetilde{\mathbf{Z}}_{t-1}^{(\ell)}))\Big)}_{(A)}
$$

$$
\underbrace{- \Big([\mathbf{L}\widetilde{\mathbf{H}}_t^{(\ell-1)}]^\top (\mathbf{D}_t^{(\ell+1)} \circ \sigma'(\widetilde{\mathbf{Z}}_t^{(\ell)}))\Big)}_{(A)}\|_{\mathrm{F}}^2]
$$

$$
+ 2\,\mathbb{E}[\|\underbrace{\Big([\mathbf{L}\widetilde{\mathbf{H}}_t^{(\ell-1)}]^\top (\mathbf{D}_t^{(\ell+1)} \circ \sigma'(\widetilde{\mathbf{Z}}_t^{(\ell)}))\Big) - \Big([\mathbf{L}\mathbf{H}_t^{(\ell-1)}]^\top (\mathbf{D}_t^{(\ell+1)} \circ \sigma'(\mathbf{Z}_t^{(\ell)}))\Big)}_{(B)}\|_{\mathrm{F}}^2]
$$

$$(211)$$

Let first take a closer look at term $(A)$. Let suppose $t \in \{EK+1, \ldots, EK+K\}$, where $E = t \mod K$ is the current epoch number and $K$ is the inner-loop size. By the previous lemma, term $(A)$ can be bounded by

$$
(A) \leq \sum_{t=EK+1}^{EK+K} \Big(\mathcal{O}(\|\mathbf{W}_t^{(1)} - \mathbf{W}_{t-1}^{(1)}\|_{\mathrm{F}}^2) + \ldots + \mathcal{O}(\|\mathbf{W}_t^{(L)} - \mathbf{W}_{t-1}^{(L)}\|_{\mathrm{F}}^2)\Big) \qquad (212)
$$

Then we take a closer look at term $(B)$.

$$
\mathbb{E}[\|\Big([\mathbf{L}\widetilde{\mathbf{H}}_t^{(\ell-1)}]^\top (\mathbf{D}_t^{(\ell+1)} \circ \sigma'(\widetilde{\mathbf{Z}}_t^{(\ell)}))\Big) - \Big([\mathbf{L}\mathbf{H}_t^{(\ell-1)}]^\top (\mathbf{D}_t^{(\ell+1)} \circ \sigma'(\mathbf{Z}_t^{(\ell)}))\Big)\|_{\mathrm{F}}^2]
$$

$$
\leq 2(B_{LA}^2 B_D^2 C_\sigma^4 + B_{LA}^2 B_H^2 B_D^2) \underbrace{\mathbb{E}[\|\widetilde{\mathbf{Z}}_t^{(\ell)} - \mathbf{Z}_t^{(\ell)}\|_{\mathrm{F}}^2]}_{(C)} \qquad (213)
$$

Let suppose $t \in \{EK+1, \ldots, EK+K\}$. Then we can denote $t = EK+k$ for some $k \leq K$. From Eq. 220, we know that

$$
(C) = \mathbb{E}[\|\mathbf{L}\widetilde{\mathbf{H}}_t^{(\ell-1)}\mathbf{W}_t^{(\ell)} - \widetilde{\mathbf{Z}}_t^{(\ell)}\|_{\mathrm{F}}^2]
$$

$$
\leq \sum_{t=EK+1}^{EK+K} \mathcal{O}\Big(|\mathbb{E}[\|\widetilde{\mathbf{L}}^{(\ell)}\|_{\mathrm{F}}^2] - \|\mathbf{L}\|_{\mathrm{F}}^2| \times \mathbb{E}[\|\mathbf{W}_t^{(\ell)} - \mathbf{W}_{t-1}^{(\ell)}\|_{\mathrm{F}}^2]\Big) \qquad (214)
$$

Plugging $(C)$ back to $(B)$, combing with $(A)$ we have

$$
\mathbb{E}[\|\nabla_W \widetilde{f}^{(\ell)}(\mathbf{D}_t^{(\ell+1)}, \widetilde{\mathbf{H}}_t^{(\ell-1)}, \mathbf{W}_t^{(\ell)}) - \nabla_W f^{(\ell)}(\mathbf{D}_t^{(\ell+1)}, \mathbf{H}_t^{(\ell-1)}, \mathbf{W}_t^{(\ell)})\|_{\mathrm{F}}^2]
$$

$$
\leq \sum_{t=EK+1}^{EK+K} \eta^2 \mathcal{O}\Big(|\mathbb{E}[\|\widetilde{\mathbf{L}}^{(\ell)}\|_{\mathrm{F}}^2] - \|\mathbf{L}\|_{\mathrm{F}}^2| \times \mathbb{E}[\|\nabla\widetilde{\mathcal{L}}(\boldsymbol{\theta}_t)\|_{\mathrm{F}}^2]\Big) \qquad (215)
$$

$\square$

In the following lemma, we provide the upper-bound of Eq. 153, which is one of the three key factors that affect the mean-square error of stochastic gradient at the $\ell$th layer.

**Lemma 11.** *Let suppose $t \in \{EK+1, \ldots, EK+K\}$. Then the upper bound of*

$$
\mathbb{E}[\|\widetilde{\mathbf{D}}_t^{(L+1)} - \mathbf{D}_t^{(L+1)}\|_{\mathrm{F}}^2] \leq \sum_{t=EK+1}^{EK+K} \eta^2 \times \mathcal{O}\Big(|\mathbb{E}[\|\widetilde{\mathbf{L}}^{(\ell)}\|_{\mathrm{F}}^2] - \|\mathbf{L}\|_{\mathrm{F}}^2| \times \mathbb{E}[\|\nabla\widetilde{\mathcal{L}}(\boldsymbol{\theta}_{t-1})\|_{\mathrm{F}}^2]\Big) \quad (216)
$$

*Proof.* By definition we have

$$\|\widetilde{\mathbf{D}}_t^{(L+1)} - \mathbf{D}_t^{(L+1)}\|_{\mathrm{F}}^2 \leq \left\|\frac{\partial \widetilde{\mathcal{L}}(\boldsymbol{\theta}_t)}{\partial \widetilde{\mathbf{H}}_t^{(L)}} - \frac{\partial \mathcal{L}(\boldsymbol{\theta}_t)}{\partial \mathbf{H}_{t-1}^{(L)}}\right\|_{\mathrm{F}}^2$$

$$\leq L_{\mathrm{loss}}^2 \|\widetilde{\mathbf{H}}_t^{(L)} - \mathbf{H}_t^{(L)}\|_{\mathrm{F}}^2 \qquad (217)$$

$$\leq L_{\mathrm{loss}}^2 C_\sigma^2 \underbrace{\|\widetilde{\mathbf{Z}}_t^{(L)} - \mathbf{Z}_t^{(L)}\|_{\mathrm{F}}^2}_{(A)}$$

Let take closer look at term $(A)$:

$$\|\widetilde{\mathbf{Z}}_t^{(L)} - \mathbf{Z}_t^{(L)}\|_{\mathrm{F}}^2 = \|\widetilde{f}^{(L)}(\widetilde{\mathbf{H}}_t^{(L-1)}, \mathbf{W}^{(L)}) - f^{(L)}(\mathbf{H}_t^{(L-1)}, \mathbf{W}^{(L)})\|_{\mathrm{F}}^2$$

$$\leq 2\|\widetilde{f}^{(L)}(\widetilde{\mathbf{H}}_t^{(L-1)}, \mathbf{W}^{(L)}) - f^{(L)}(\widetilde{\mathbf{H}}_t^{(L-1)}, \mathbf{W}^{(L)})\|_{\mathrm{F}}^2$$

$$\qquad + 2\|f^{(L)}(\widetilde{\mathbf{H}}_t^{(L-1)}, \mathbf{W}^{(L)}) - f^{(L)}(\mathbf{H}_t^{(L-1)}, \mathbf{W}^{(L)})\|_{\mathrm{F}}^2$$

$$= 2\|\widetilde{f}^{(L)}(\widetilde{\mathbf{H}}_t^{(L-1)}, \mathbf{W}^{(L)}) - f^{(L)}(\widetilde{\mathbf{H}}_t^{(L-1)}, \mathbf{W}^{(L)})\|_{\mathrm{F}}^2 \qquad (218)$$

$$\qquad + 2\|\sigma(\mathbf{L}\widetilde{\mathbf{H}}_t^{(L-1)}\mathbf{W}^{(L)}) - \sigma(\mathbf{L}\mathbf{H}_t^{(L-1)}\mathbf{W}^{(L)})\|_{\mathrm{F}}^2$$

$$\leq 2\|\widetilde{f}^{(L)}(\widetilde{\mathbf{H}}_t^{(L-1)}, \mathbf{W}^{(L)}) - f^{(L)}(\widetilde{\mathbf{H}}_t^{(L-1)}, \mathbf{W}^{(L)})\|_{\mathrm{F}}^2$$

$$\qquad + 2C_\sigma^4 B_{LA}^2 B_W^2 \|\widetilde{\mathbf{Z}}_t^{(L-1)} - \mathbf{Z}_t^{(L-1)}\|_{\mathrm{F}}^2$$

By induction, we have

$$\|\widetilde{\mathbf{Z}}_t^{(L)} - \mathbf{Z}_t^{(L)}\|_{\mathrm{F}}^2 \leq \sum_{\ell=1}^L \mathcal{O}(\|\widetilde{f}^{(\ell)}(\widetilde{\mathbf{H}}_t^{(\ell-1)}, \mathbf{W}^{(\ell)}) - f^{(\ell)}(\widetilde{\mathbf{H}}_t^{(\ell-1)}, \mathbf{W}^{(\ell)})\|_{\mathrm{F}}^2) \qquad (219)$$

Let suppose $t \in \{EK+1, \ldots, EK+K\}$. Then we can denote $t = EK+k$ for some $k \leq K$. From Eq. 220, we know that

$$\|\widetilde{f}^{(\ell)}(\widetilde{\mathbf{H}}_t^{(\ell-1)}, \mathbf{W}^{(\ell)}) - f^{(\ell)}(\widetilde{\mathbf{H}}_t^{(\ell-1)}, \mathbf{W}^{(\ell)})\|_{\mathrm{F}}^2$$

$$\leq \sum_{t=EK+1}^{EK+K} \mathcal{O}\left(|\mathbb{E}[\|\widetilde{\mathbf{L}}^{(\ell)}\|_{\mathrm{F}}^2] - \|\mathbf{L}\|_{\mathrm{F}}^2| \times \mathbb{E}[\|\mathbf{W}_t^{(\ell)} - \mathbf{W}_{t-1}^{(\ell)}\|_{\mathrm{F}}^2]\right) \qquad (220)$$

Therefore, we know

$$\|\widetilde{\mathbf{D}}_t^{(L+1)} - \mathbf{D}_t^{(L+1)}\|_{\mathrm{F}}^2 \leq L_{\mathrm{loss}}^2 C_\sigma^2 \|\widetilde{\mathbf{Z}}_t^{(L)} - \mathbf{Z}_t^{(L)}\|_{\mathrm{F}}^2$$

$$\leq \sum_{\ell=1}^L \sum_{t=EK+1}^{EK+K} \mathcal{O}\left(|\mathbb{E}[\|\widetilde{\mathbf{L}}^{(\ell)}\|_{\mathrm{F}}^2] - \|\mathbf{L}\|_{\mathrm{F}}^2| \times \mathbb{E}[\|\mathbf{W}_t^{(\ell)} - \mathbf{W}_{t-1}^{(\ell)}\|_{\mathrm{F}}^2]\right) \qquad (221)$$

$$= \sum_{t=EK+1}^{EK+K} \eta^2 \times \mathcal{O}\left(|\mathbb{E}[\|\widetilde{\mathbf{L}}^{(\ell)}\|_{\mathrm{F}}^2] - \|\mathbf{L}\|_{\mathrm{F}}^2| \times \mathbb{E}[\|\nabla\widetilde{\mathcal{L}}(\boldsymbol{\theta}_{t-1})\|_{\mathrm{F}}^2]\right)$$

which conclude the proof. $\qquad\square$

Combing the upper-bound of Eq. 153, 154, 155, we provide the upper-bound of mean-suqare error of stochastic gradient in `SGCN++`.

**Lemma 12.** *Let suppose* $t \in \{EK+1, \ldots, EK+K\}$. *Then we can denote* $t = EK+k$ *for some* $k \leq K$ *such that*

$$\mathbb{E}[\|\nabla\widetilde{\mathcal{L}}(\boldsymbol{\theta}_t) - \nabla\mathcal{L}(\boldsymbol{\theta}_t)\|_{\mathrm{F}}^2] \leq \sum_{t=EK+1}^{EK+K} \eta^2 \mathcal{O}\left(|\mathbb{E}[\|\widetilde{\mathbf{L}}^{(\ell)}\|_{\mathrm{F}}^2] - \|\mathbf{L}\|_{\mathrm{F}}^2| \times \mathbb{E}[\|\nabla\widetilde{\mathcal{L}}(\boldsymbol{\theta}_{t-1})\|_{\mathrm{F}}^2]\right) \qquad (222)$$

*Proof.* From Lemma 6, we have

$$\mathbb{E}[\|\widetilde{\mathbf{G}}^{(\ell)} - \mathbf{G}^{(\ell)}\|F^2]$$

$$\leq \mathcal{O}(\mathbb{E}[\|\widetilde{\mathbf{D}}^{(L+1)} - \mathbf{D}^{(L+1)}\|_{\mathrm{F}}^2])$$

$$\qquad + \mathcal{O}(\mathbb{E}[\|\nabla_H \widetilde{f}^{(L)}(\mathbf{D}^{(L+1)}, \widetilde{\mathbf{H}}^{(L-1)}\mathbf{W}^{(L)}) - \nabla_H f^{(L)}(\mathbf{D}^{(L+1)}, \mathbf{H}^{(L-1)}\mathbf{W}^{(L)})\|_{\mathrm{F}}^2]) + \cdots$$

$$\qquad + \mathcal{O}(\mathbb{E}[\|\nabla_H \widetilde{f}^{(\ell+1)}(\mathbf{D}^{(\ell+2)}, \widetilde{\mathbf{H}}^{(\ell)}, \mathbf{W}^{(\ell+1)}) - \nabla_H f^{(\ell+1)}(\mathbf{D}^{(\ell+2)}, \mathbf{H}^{(\ell)}, \mathbf{W}^{(\ell+1)})\|_{\mathrm{F}}^2])$$

$$\qquad + \mathcal{O}(\mathbb{E}[\|\nabla_W \widetilde{f}^{(\ell)}(\mathbf{D}^{(\ell+1)}, \widetilde{\mathbf{H}}^{(\ell-1)}, \mathbf{W}^{(\ell)}) - \nabla_W f^{(\ell)}(\mathbf{D}^{(\ell+1)}, \mathbf{H}^{(\ell-1)}, \mathbf{W}^{(\ell)})\|_{\mathrm{F}}^2])$$

$$\tag{223}$$

Plugging the result from support lemmas, i.e., Lemma 8, 10, 11 and using the definition of stochastic gradient for all model parameters $\nabla\widetilde{\mathcal{L}}(\boldsymbol{\theta}_t) = \{\widetilde{\mathbf{G}}_t^{(\ell)}\}_{\ell=1}^L$ we have

$$\mathbb{E}[\|\nabla\widetilde{\mathcal{L}}(\boldsymbol{\theta}_t) - \nabla\mathcal{L}(\boldsymbol{\theta}_t)\|_{\mathrm{F}}^2] \leq \sum_{t=EK+1}^{EK+K} \eta^2 \mathcal{O}\Big( \sum_{\ell=1}^{L} |\mathbb{E}[\|\widetilde{\mathbf{L}}^{(\ell)}\|_{\mathrm{F}}^2] - \|\mathbf{L}\|_{\mathrm{F}}^2| \times \mathbb{E}[\|\nabla\widetilde{\mathcal{L}}(\boldsymbol{\theta}_{t-1})\|_{\mathrm{F}}^2]\Big) \tag{224}$$

$$\square$$

## I.2 Remaining steps toward Theorem 3

By the smoothness of $\mathcal{L}(\boldsymbol{\theta}_t)$, we have

$$\mathcal{L}(\boldsymbol{\theta}_{T+1}) \leq \mathcal{L}(\boldsymbol{\theta}_t) + \langle\nabla\mathcal{L}(\boldsymbol{\theta}_t), \boldsymbol{\theta}_{t+1} - \boldsymbol{\theta}_t\rangle + \frac{L_f}{2}\|\boldsymbol{\theta}_{t+1} - \boldsymbol{\theta}_t\|_{\mathrm{F}}^2$$

$$= \mathcal{L}(\boldsymbol{\theta}_t) - \eta\langle\nabla\mathcal{L}(\boldsymbol{\theta}_t), \nabla\widetilde{\mathcal{L}}(\boldsymbol{\theta}_t)\rangle + \frac{\eta^2 L_f}{2}\|\nabla\widetilde{\mathcal{L}}(\boldsymbol{\theta}_t)\|_{\mathrm{F}}^2$$

$$\overset{(a)}{=} \mathcal{L}(\boldsymbol{\theta}_t) - \frac{\eta}{2}\|\nabla\mathcal{L}(\boldsymbol{\theta}_t)\|_{\mathrm{F}}^2 + \frac{\eta}{2}\|\nabla\mathcal{L}(\boldsymbol{\theta}_t) - \nabla\widetilde{\mathcal{L}}(\boldsymbol{\theta}_t)\|_{\mathrm{F}}^2 - \Big(\frac{\eta}{2} - \frac{L_f\eta^2}{2}\Big)\|\nabla\widetilde{\mathcal{L}}(\boldsymbol{\theta}_t)\|_{\mathrm{F}}^2$$

$$\tag{225}$$

where equality $(a)$ is due to the fact $2\langle\boldsymbol{x}, \boldsymbol{y}\rangle = \|\boldsymbol{x}\|_{\mathrm{F}}^2 + \|\boldsymbol{y}\|_{\mathrm{F}}^2 - \|\boldsymbol{x} - \boldsymbol{y}\|_{\mathrm{F}}^2$ for any $\boldsymbol{x}, \boldsymbol{y}$.

Take expectation on both sides, we have

$$\mathbb{E}[\mathcal{L}(\boldsymbol{\theta}_{T+1})] \leq \mathbb{E}[\mathcal{L}(\boldsymbol{\theta}_t)] - \frac{\eta}{2}\mathbb{E}[\|\nabla\mathcal{L}(\boldsymbol{\theta}_t)\|_{\mathrm{F}}^2] + \frac{\eta}{2}\mathbb{E}[\|\nabla\mathcal{L}(\boldsymbol{\theta}_t) - \nabla\widetilde{\mathcal{L}}(\boldsymbol{\theta}_t)\|_{\mathrm{F}}^2] - \Big(\frac{\eta}{2} - \frac{L_f\eta^2}{2}\Big)\mathbb{E}[\|\nabla\widetilde{\mathcal{L}}(\boldsymbol{\theta}_t)\|_{\mathrm{F}}^2]$$

$$\tag{226}$$

By summing over $t = 1, \ldots, T$ where $T$ is the inner-loop size, we have

$$\sum_{t=1}^{T} \mathbb{E}[\|\nabla\mathcal{L}(\boldsymbol{\theta}_t)\|_{\mathrm{F}}^2]$$

$$\leq \frac{2}{\eta}\Big(\mathbb{E}[\mathcal{L}(\boldsymbol{\theta}_1)] - \mathbb{E}[\mathcal{L}(\boldsymbol{\theta}_{T+1})]\Big) + \sum_{t=1}^{T}[\mathbb{E}[\|\nabla\mathcal{L}(\boldsymbol{\theta}_t) - \nabla\widetilde{\mathcal{L}}(\boldsymbol{\theta}_t)\|_{\mathrm{F}}^2] - \Big(1 - L_f\eta\Big)\mathbb{E}[\|\nabla\widetilde{\mathcal{L}}(\boldsymbol{\theta}_t)\|_{\mathrm{F}}^2]]$$

$$\leq \frac{2}{\eta}\Big(\mathbb{E}[\mathcal{L}(\boldsymbol{\theta}_1)] - \mathbb{E}[\mathcal{L}(\boldsymbol{\theta}^\star)]\Big) + \sum_{t=1}^{T}[\mathbb{E}[\|\nabla\mathcal{L}(\boldsymbol{\theta}_t) - \nabla\widetilde{\mathcal{L}}(\boldsymbol{\theta}_t)\|_{\mathrm{F}}^2] - \Big(1 - L_f\eta\Big)\mathbb{E}[\|\nabla\widetilde{\mathcal{L}}(\boldsymbol{\theta}_t)\|_{\mathrm{F}}^2]]$$

$$\tag{227}$$

where $\mathcal{L}(\boldsymbol{\theta}^\star)$ is the global optimal solution.

Let us consider each inner-loop with $E \in [0, T/K - 1]$ and $t \in \{EK + 1, \ldots, EK + K\}$. Using Lemma 12 we have

$$\sum_{E=0}^{T/K-1} \sum_{t=EK+1}^{EK+K} \mathbb{E}[\|\mathcal{L}(\boldsymbol{\theta}_t) - \nabla\widetilde{\mathcal{L}}(\boldsymbol{\theta}_t)\|_{\mathrm{F}}^2] - \left(1 - L_f\eta\right) \sum_{E=0}^{T/K-1} \sum_{t=EK+1}^{EK+K} \mathbb{E}[\|\nabla\widetilde{\mathcal{L}}(\boldsymbol{\theta}_t)\|_{\mathrm{F}}^2]$$

$$\leq \mathcal{O}\Big( \sum_{\ell=1}^{L} |\mathbb{E}[\|\widetilde{\mathbf{L}}^{(\ell)}\|_{\mathrm{F}}^2] - \|\mathbf{L}\|_{\mathrm{F}}^2| \Big) \Big( \sum_{E=0}^{T/K-1} \sum_{t=EK+1}^{EK+K} \sum_{j=2}^{t-EK} \eta^2 \mathbb{E}[\|\nabla\widetilde{\mathcal{L}}(\boldsymbol{\theta})_{j-1}\|_{\mathrm{F}}^2] \Big)$$

$$- \left(1 - L_f\eta\right) \sum_{E=0}^{T/K-1} \sum_{t=EK+1}^{EK+K} \mathbb{E}[\|\nabla\widetilde{\mathcal{L}}(\boldsymbol{\theta}_t)\|_{\mathrm{F}}^2] \tag{228}$$

$$\leq [\eta^2 K \times \mathcal{O}\Big( \sum_{\ell=1}^{L} |\mathbb{E}[\|\widetilde{\mathbf{L}}^{(\ell)}\|_{\mathrm{F}}^2] - \|\mathbf{L}\|_{\mathrm{F}}^2| \Big) - \left(1 - L_f\eta\right)] \sum_{E=0}^{T/K-1} \sum_{t=EK+1}^{EK+K} \mathbb{E}[\|\nabla\widetilde{\mathcal{L}}(\boldsymbol{\theta}_t)\|_{\mathrm{F}}^2]$$

$$= [\eta^2 \Delta_{\mathbf{b+n}}^{++'} - (1 - L_f\eta)] \sum_{E=0}^{T/K-1} \sum_{t=EK+1}^{EK+K} \mathbb{E}[\|\nabla\widetilde{\mathcal{L}}(\boldsymbol{\theta}_t)\|_{\mathrm{F}}^2]$$

Notice that $\eta = \frac{2}{L_f + \sqrt{L_f^2 + 4\Delta_{\mathbf{b+n}}^{++'}}}$ is a root of equation $\eta^2 \Delta_{\mathbf{b+n}}^{++'} - (1 - L_f\eta) = 0$. Therefore we have

$$\sum_{t=1}^{T} \mathbb{E}[\|\nabla\mathcal{L}(\boldsymbol{\theta}_t)\|_{\mathrm{F}}^2] \leq \frac{2}{\eta}\Big( \mathbb{E}[\mathcal{L}(\boldsymbol{\theta}_1)] - \mathbb{E}[\mathcal{L}(\boldsymbol{\theta}^\star)] \Big) \tag{229}$$

which implies

$$\frac{1}{T}\sum_{t=1}^{T} \mathbb{E}[\|\nabla\mathcal{L}(\boldsymbol{\theta}_t)\|_{\mathrm{F}}^2] \leq \frac{1}{T}\Big( L_f + \sqrt{L_f^2 + 4\Delta_{\mathbf{b+n}}^{++'}} \Big)\Big( \mathbb{E}[\mathcal{L}(\boldsymbol{\theta}_1)] - \mathbb{E}[\mathcal{L}(\boldsymbol{\theta}^\star)] \Big) \tag{230}$$

## J  REPRODUCING EXPERIMENT RESULTS

To reproduce the results reported in the paper, we provide link for dataset download, the bash script to reproduce the experiment results, and a jupyter notebook file for a quick visualization and GPU utilization calculation. It is worth noting that due to the existence of randomness, the obtained results (e.g., loss curve) may be slightly different. However, it is not difficult to find that the overall trend of loss curves and conclusions will remains the same.

This implementation is based on [4]PyTorch using Python 3. We notice that Python 2 might results in a wrong gradient update, even for vanilla SGCNs.

Install dependencies:

```
# create virtual environment
$ virtualenv env
$ source env/bin/activate
# install dependencies
$ pip install -r requirements.txt
```

Experiments are produced on PPI, PPI-Large, Flickr, Reddit, and Yelp datasets. The utilized datasets can be downloaded from [5]Google drive.

```
# create folders that save experiment results and datasets
$ mkdir ./results
$ mkdir ./data # please download the dataset and put them inside this folder
```

---

[4]https://pytorch.org/
[5]https://drive.google.com/drive/folders/15eP7OHiHQUnDrHKYh1YPxXkiqGoJhbis?usp=sharing

To reproduce the results, please run the following commands:

```
$ python train.py --sample_method 'ladies' --dataset 'reddit'
$ python train.py --sample_method 'fastgcn' --dataset 'reddit'
$ python train.py --sample_method 'graphsage' --dataset 'reddit'
$ python train.py --sample_method 'vrgcn' --dataset 'reddit'
$ python train.py --sample_method 'graphsaint' --dataset 'reddit'
$ python train.py --sample_method 'exact' --dataset 'reddit'

$ python train.py --sample_method 'ladies' --dataset 'ppi'
$ python train.py --sample_method 'fastgcn' --dataset 'ppi'
$ python train.py --sample_method 'graphsage' --dataset 'ppi'
$ python train.py --sample_method 'vrgcn' --dataset 'ppi'
$ python train.py --sample_method 'graphsaint' --dataset 'ppi'
$ python train.py --sample_method 'exact' --dataset 'ppi'

$ python train.py --sample_method 'ladies' --dataset 'flickr'
$ python train.py --sample_method 'fastgcn' --dataset 'flickr'
$ python train.py --sample_method 'graphsage' --dataset 'flickr'
$ python train.py --sample_method 'vrgcn' --dataset 'flickr'
$ python train.py --sample_method 'graphsaint' --dataset 'flickr'
$ python train.py --sample_method 'exact' --dataset 'flickr'

$ python train.py --sample_method 'ladies' --dataset 'ppi-large'
$ python train.py --sample_method 'fastgcn' --dataset 'ppi-large'
$ python train.py --sample_method 'graphsage' --dataset 'ppi-large'
$ python train.py --sample_method 'vrgcn' --dataset 'ppi-large'
$ python train.py --sample_method 'graphsaint' --dataset 'ppi-large'
$ python train.py --sample_method 'exact' --dataset 'ppi-large'

$ python train.py --sample_method 'ladies' --dataset 'yelp'
$ python train.py --sample_method 'fastgcn' --dataset 'yelp'
$ python train.py --sample_method 'graphsage' --dataset 'yelp'
$ python train.py --sample_method 'vrgcn' --dataset 'yelp'
$ python train.py --sample_method 'graphsaint' --dataset 'yelp'
$ python train.py --sample_method 'exact' --dataset 'yelp'
```

