# OpenReview forum: "On the Importance of Sampling in Training GCNs: Convergence Analysis and Variance Reduction"
_ICLR.cc/2021/Conference — Reject_

### Official Review · AnonReviewer4 · 2020-10-26
**need to clarify contribution**

**Rating:** 4
**Confidence:** 4

**Review:**

The paper provides a convergence analysis for GCN training and proposes two variance reduction algorithms to ensure convergence.

The authors first draw a connection between sampling-based GCN training and compositional optimization, and they divide the error of gradient into a bias term and a variance term. This is actually first established in [Cong et al. Minimal Variance Sampling with Provable Guarantees for Fast Training of Graph Neural Networks]. Although there is some inaccuracy in their formulation, the authors should clarify that this observation is first made by previous work.

Based on the variance and bias decoupling, the authors then establish a convergence result for biased SGD (Theorem 1). The analysis itself is trivial and is not new. The authors should consider adding some references.

The authors then propose an algorithm to reduce the bias term (Algorithm 1). Again, this is not new. It is exactly the nested SPIDER method for solving multi-level stochastic compositional optimization [Zhang et al. Multi-Level Composite Stochastic Optimization via Nested Variance Reduction]. Although the paper is the first to apply the algorithm to GCN training (as far as I know), it is important to have correct references to clarify the contribution. Also, it is worth noting that the algorithm is based on the assumption that the full neighbor aggregation result is available and the storage of Z and H is possible. I doubt the practicality of the algorithm for training on large graphs.

The experimental results validate that the algorithm works for GCN training. However, I notice that some values in Table 1 for the baseline (SGCN) are not the best results reported in other papers. For example, GraphSAINT achieves 96% on Reddit graph, but the authors report 93.68% as the baseline. This makes me suspect that the good results reported in the paper may be due to some parameter tuning instead of the algorithm itself.


A minor point:
The paper says “the data points in these works are sampled independently, but the data points (nodes) in SGCN are sampled node- or layer-dependent according to the graph structure.”
The data samples in compositional optimization do not need to be independent. The only requirement is that the stochastic function with the data samples is unbiased.

---

> ### Author Response · Authors · 2020-11-20
> **Response to Reviewer 4**
>
> We would like to thank the reviewer's informative and helpful comments which indeed help us to improve the quality and readability of this paper. We will address your concerns as follows.
>
> **1. Question: Novelty of bias-variance decomposition in GCNs.**
>
> Both [5] and ours are inspired by bias-variance decomposition, with a key difference in formulation.
> Let denote $\boldsymbol{g}$ as the stochastic gradient and $\nabla F(\boldsymbol{\theta})$ as the full-batch gradient. Our formulation follows the standard definition of mean-square error of stochastic gradient, which is $\mathbb{E}[\| \boldsymbol{g} - \nabla F(\boldsymbol{\theta})\|\_F^2] = \mathbb{E}[\| \boldsymbol{g} - \mathbb{E}[\boldsymbol{g}]\|\_F^2] + \mathbb{E}[\| \mathbb{E}[\boldsymbol{g}] - \nabla F(\boldsymbol{\theta})\|\_F^2]$, where $\mathbb{E}[\| \boldsymbol{g} - \mathbb{E}[\boldsymbol{g}]\|\_F^2]$ is known as the variance and $\mathbb{E}[\| \mathbb{E}[\boldsymbol{g}] - \nabla F(\boldsymbol{\theta})\|_F^2]$ is known as the bias. [5] treats $\mathbb{E}[\boldsymbol{g}]$ as the gradient computed using all neighbors with mini-batch sampling (Eq.~2 in [5]), which is different from ours. This difference results in a different analysis and theoretical results. Furthermore, [5] does not provide any further analysis (e.g., convergence) based on their observation of bias-variance decomposition.
>
> **2. Question: Novelty of analysis of biased SGD and variance reduced algorithm.**
>
> Please refer to the general comment on the novelty of analysis, algorithms, and theoretical results.
>
> **3. Question: Full-batch information and the storage of historical hidden embeddings**
>
> We agree with the reviewer that computing the full-batch node embeddings and gradient will hinder the scalability of variance reduced algorithm for large real-world graphs. Therefore, at the end of Section 5 (both original submission and updated version), we propose to approximate the full-batch gradient with the gradient computed on the large-batch using all neighbors. The intuition stems from matrix Bernstein inequality, details can be found at the end of Appendix E. In addition, we analyze the effectiveness of large-batch size to the performance, detailed can be found in Appendix C.
>
> Besides, we also agree with reviewer that storing the historical hidden embeddings can impede the scalability. However, since the dimension for hidden embeddings are usually small (no more than $512$), this is not a concern. For example, our algorithm can load the adjacency matrix, node features, historical embeddings, and models on a machine with $32$GB of memory when training on Yelp datasets ($0.7\times10^6$ nodes and $7\times 10^6$ edges). Furthermore, when using the large-batch approximation, we only need to save the embedding of the sampled large-batch nodes in memory, where a GPU with only $8$G memory can handle all the aforementioned information for variance reduction.

---

> > ### Author Response · Authors · 2020-11-20
> > **Response to Reviewer 4 (cont'd)**
> >
> > **4. Question: Performance gap between reported accuracy in this paper to original paper.**
> >
> > Please note that we are using the same hyper-parameters (i.e., number of layers as $2$, learning rate $0.01$, hidden dimension size as $256$, batch size as $512$) for all $5$ datasets, $6$ baseline algorithms, $5$ first-order variance reduction, and $5$ doubly variance reduction algorithms. Besides, in Appendix C, we also explore the effectiveness of mini-batch size (mini-batch size as $512, 2048, 4096$) on GraphSAINT and GraphSAINT++. These are general and default configurations that widely used by existing framework implementations [5,12]. All implementations have exactly the same model structure. The only difference between each baseline model is their data sampler.
> >
> > As discussed in Appendix B, the difference in performance are mainly due to the difference in the model model structure. For example, different graph convolution operations (e.g., mean aggregation, concatenate aggregation, graph attention aggregation), different model structures (e.g., skip-connected, jumping knowledge, initial residual connection), different number of layers (different layers for different datasets). These inconsistent configurations make it hard to truly understand the performance between different methods. Therefore, we implement a unified framework that compares existing methods fairly. We also provided detailed instructions on reproducing the results in Appendix J.
> >
> > GraphSAINT is a subgraph sampling algorithm, where the subgraphs are sampled by choosing nodes independently under some importance sampling distribution. When the dataset is large, the sampled subgraph could be extremely sparse, which makes the training data distribution diverge from testing data distribution (stochastic gradient different from full-batch gradient). Intuitively, by using the same model structure and hyper-parameters, subgraph sampling cannot perform better than nodewise sampling or layerwise sampling, where the connectivity of the computation graph can be guaranteed. More importantly, in Theorem 1, the vanilla sampling-based GCN suffers a residual error during training, and this residual error is proportional to bias of gradient and $\mathcal{O}(\| \widetilde{\mathbf{L}}^{(\ell)} - \mathbf{P}^{(\ell)} \|_F^2)$. This theorem also implies that GCNs will suffer a performance decay if the sampled training Laplacian matrix $\widetilde{\mathbf{L}}^{(\ell)}$ is too different from the full Laplacian.
> >
> > **5. Question: Data point sampling dependency.**
> >
> > We agree with the reviewer that not all composite optimization methods consider an independent sampling.
> > However, dependent sampling makes the unbiased sampling hard to guarantee, and could be another interesting topic to explore. For example, the Eq.1 in [11] it is assumed that the random variances are independent. This is different in most sampling-based methods, where the nodes  are first sampled from the $L$th layer to $1$st layer, then inference from the $1$st layer to $L$th layer, with layerwise dependency taken into considerations. This type of dependency are used in our analysis to analyze the relation between embedding approximation variance to $\mathcal{O}(\| \widetilde{\mathbf{L}}^{(\ell)} - \mathbf{P}^{(\ell)}\|\_F^2 )$, and stochastic gradient variance to $\mathcal{O}(\| \mathbf{P}^{(\ell)} - \mathbf{L}\|_F^2)$, which is one of the key take home messages we provided in theorems.

---

### Official Review · AnonReviewer1 · 2020-10-27
**The theory is not consistent with experiments**

**Rating:** 4
**Confidence:** 5

**Review:**

This paper studies the convergence of stochastic training methods for graph neural networks. Here, this paper views GNN as a compositional optimization problem. Then, to reduce the variance incurred by the neighbor sampling, this paper uses SPIDER to reduce the variance to accelerate the convergence speed. It provides theoretical convergence analysis for SPIDER used on GNNs, showing that the proposed method has a better convergence rate compared with the traditional gradient descent method. At last, this paper conducts experiments to verify the proposed algorithm.

1. Overall, this is a new application of SPIDER on GNNs. It shows how to bound the variance under the setting of GNNs, which is then used for the convergence proof. However, some assumptions are too strong, such as assumption 3. With these strong assumptions, the theoretical analysis is simplified too much.

2. Some parts are not very clear. In eq.(3), this paper claims that the bias term is mainly caused by node embedding approximation in the forward pass while the variance term is caused by gradient varinace in the backward pass. It is very confusing. The node embedding approximation affects both the forward and backward pass. Why does it only affect the forward pass?

3. For figure 3, the exact sampling method uses all neighbors. Therefore, it is actually the full gradient desenct method. Why does there exist variance?

4. The most weak point of this paper is that the theory is NOT consistent with experiments. In detail, the theorems study the convergence rate of (variance-reduced) SGD. However, in experiments, this paper uses Adam. Thus, the experimental results cannot support the claim of theories.

---

> ### Author Response · Authors · 2020-11-20
> **Response to Reviewer 1**
>
>
> Thanks for your careful and valuable comments. We address your concerns as follows.
>
> **1. Question: This paper is an another application of SPDIER on GCN and Assumption 3 is too strong.**
>
> Please refer to the general comment for clarifications about this work and SPIDER (Appendix A in revised manuscript)
>
> In Assumption 3, we assume that the Frobenius norm of weight matrices $\{\mathbf{W}^{(\ell)}\}\_{\ell=1}^L$, Laplacian matrices $\\\{\widetilde{\mathbf{L}}^{(\ell)}, \mathbf{L}^{(\ell)}\\\}_{\ell=1}^L$, and input feature matrix $\mathbf{X}$ are bounded.
> Next we will explain why these almost surely hold in GCN training:
>
> - **Weight matrices**. Random initialization gives weight matrices a small norm and lazy training [4] guarantees the weight is close to the initial value, such that it still stays in a low norm status, unless over-fit happens. For example, we compute the Frobenius norm of all weight parameters of a 2-layer FullGCN on Cora dataset with dimension 16. By repeat computing $1000$ times, we get mean value as $52.718$, minimum value as $49.581$, maximum value as $54.464$, and variance as $0.436$.
> - **Laplacian matrices**. By definition we know that the spectral norm of Laplacian matrices are equals to $1$. Using the relation between spectral norm and Frobenius norm, we can easily derive the upper bound for Laplacian matrices.
> - **Feature matrix**. Input feature matrix are usually pre-normalized, which has a fixed Frobenius norm.
>
> Note that these assumptions have been previously used in the generalization analysis of GCNs [6]. Since our main focus is on the sampling and variance reduction, these assumptions can help us focus on the key idea proposed in the paper.
>
>
> **2. Question: Definition of two types of variance is not clear.**
>
> We agree with the reviewer about the confusing definition and we believe that is mainly due to our definition of embedding approximation variance and stochastic gradient variance. Next, we recall our definition and provide a detailed reason behind our decision.
>
> In this paper, we decouple the inner-layer sampling from mini-batch sampling, such that we can analyze their effect on training GCNs, respectively. Therefore, we define the node embedding approximation variance as the variance that happens during forward propagation when computing $\\\{ \widetilde{\mathbf{L}}^{(\ell)}\widetilde{\mathbf{H}}^{(\ell-1)}\\\}\_{\ell=1}^L$. After computation, these value are fixed and saved into memory for the use of backward-propagation. By doing so, we do not need to re-compute these values during backward-propagation. In addition, we define the stochastic gradient variance as the variance that happens due to mini-batch sampling. The stochastic gradient variance is propagated from $\mathbf{D}^{(L+1)}$ to each inner layers  when we compute the stochastic gradient using the information $\\\{ \widetilde{\mathbf{L}}^{(\ell)}\widetilde{\mathbf{H}}^{(\ell-1)}\\\}\_{\ell=1}^L$ that is computed during forward-propagation.
>
> By using our definition of variances, we can make a connection between embedding approximation variance to $\mathcal{O}(\| \widetilde{\mathbf{L}}^{(\ell)} - \mathbf{P}^{(\ell)}\|\_F^2 )$ and stochastic gradient variance to $\mathcal{O}(\| \mathbf{P}^{(\ell)} - \mathbf{L} \|_F^2)$, which can be extremely helpful to give novel insights on why first-order and doubly variance reduction enjoy faster convergence rate in sampling-based GCN training.
>
> **3. Question: Why exact sampling has variance?**
>
> In this paper, *exact method* is defined as mini-batching with all neighbors, which is different from full-batch where the entire graph is used. Therefore, it suffers stochastic gradient variance, and applying first-order variance reduction can reduce its stochastic gradient variance.
>
>
> **4. Question: Using Adam in empirical valuation but SGD in theory.**
>
> We use Adam for empirical evaluation for the following reasons. We added discussion and results comparing Adam and SGD optimizer in the updated version (Page 14, Figure 3).
>
> - Baseline methods training with SGD cannot converge when using a constant learning rate due to the bias and variance in stochastic gradient (Adam has some implicit variance reduction effect, which can alleviate the issue). The empirical result of SGD trained baseline models has a huge performance gap to the one trained with Adam, which makes the comparison meaningless. Please refer to Figure 3 in the updated version for detail.
> - Most public implementation of GCNs, including all implementations in Pytorch Geometric and DGL packages, use Adam optimizer instead of SGD optimizer.
> - In this paper, we mainly focus on how to estimate a stabilized stochastic gradient, instead of how to take the existing gradient for weight update. We employ Adam optimizer for all algorithms during experiment, which lead to a fair comparing. Moreover, to the best of our knowledge, the convergence analysis of Adam is still an open problem [7], and is not the focus of this paper.

---

### Official Review · AnonReviewer2 · 2020-10-29
**Experiments on discussing the version without full-batch computation are needed.**

**Rating:** 7
**Confidence:** 4

**Review:**

Node sampling is a crucial point in making GCNs efficient. While several sampling methods have been proposed previously, the theorectical convergence analysis is still lacking. This paper finds that the convergence speed is related to not only the function approximation error but also the layer-gradient error. Based on this finding, the authors suggest to take historical hidden features and historical gradients to do doubly variance reduction. Experiments are done on 5 datasets for 7 baseline sampling-based GCNs.

Pros:

1. The core contribution of this paper lies in Theorem 1, which reveals the relationship between convergence speed and the function approximation error and the layer-gradient error.

2. The idea of doubly variance reduction is reasonable.

Cons:

The biggest weakness of the proposed method is that it requires to compute snapshot features and gradients over all nodes (Line 5, Alg. 1 & Line 5 Alg. 2) before the sampling process. As this paper aims at enhancing sampling based GCNs, we should assumes no computation access/memory of performing full GCN. The authors have provided the related analyses in the appendix. It will be better if the experiments without full-batch shapshot are added in Table 1, as such we can check how it influences the final performance and if certain approximation will work well.

---

> ### Author Response · Authors · 2020-11-20
> **Response to Reviewer 2**
>
> Thanks for your careful review and valuable suggestion. We discuss this concern in the following.
>
> **1. Question: Requirement of full-batch snapshot in Table 1.**
>
> We agree with the reviewer about the use of full-batch, however note that the experiments on Reddit and Yelp datasets in Table 1 are performed using the proposed large-batch approximation (from **Algorithm 6** in **Appendix E.5**), such that the snapshot step can be computed on a single $8$ Gigabyte GPU. More specifically, we choose $50\\%$ of training nodes for Reddit dataset and $15\\%$ nodes for Yelp dataset on large-batch snapshot approximation by default.  As a result, while the theoretical convergence relies on computing full-batch snapshot, in practice we can replace it with a sufficiently large mini-batch as in Algorithm 6.
>
> Furthermore, the effectiveness of large-batch sizes instead of full-batch is evaluated in **Appendix C** (Figure 6-9 with related discussion). We clarified this matter in the revised version in Section 7.

---

### Official Review · AnonReviewer3 · 2020-10-30
**a novel method to reduce variance with theoretical guarantee, but the algorithmic novelty is somewhat weak**

**Rating:** 7
**Confidence:** 3

**Review:**

##########################################################################

Summary:
This paper presents a novel variance reduction method which can adapt to any sampling-based GCN methods (inductive GCNs). The paper draws the idea from VRGCN that integrates the historical latent representations of nodes computed with full Laplacian to approximate the that computed with sampled sparse Laplacian. The variance reduction is implemented on both node embedding approximation, as well as layer-wise gradient computation in back-propagation. The resulting algorithms lead to faster convergence rate.

##########################################################################

Reasons for score:
Overall, I vote for accepting. The proposed variance reduction techniques can successfully accelerate convergence of any sampling method, according to the experiments, while also enjoys theoretical guarantee. Yet the novelty of the proposed SGCN+/SGCN++ algorithms themselves is a little limited to some extent.

##########################################################################

Pros:
1. The authors introduced a doubly variance reduction which can effectively reduce the node approximation variance the layer-wise gradient variance of the existing sampling based GCN methods and accelerate convergence.
2. This paper also provides thorough theoretical analysis and convergence guarantee of the proposed algorithms.
3. The  authors have conducted comprehensive experiments using a variety of sampling-based GCN methods as building blocks. The quantitative results clearly demonstrate the effectiveness of the proposed algorithms. The authors also provide detailed empirical analysis on the training time / GPU memory usage of the proposed method.

##########################################################################

Cons:
1. To better illustrate the idea the variance reduction, the authors could compare the proposed algorithms with a vanilla full-batch GCN (instead of the mini-batch training version Exact used in this paper, and it could be evaluated on smaller datasets like Cora). This baseline may serve as a theoretical upper bound of SGCN+/SGCN++.
2. The proposed SGCN+ and SGCN++ requires a full-batch forward/backward computation every k step. As the authors suggest, this might hinder the scalability of SGCN++ on extremely large graphs. The authors hence propose a variant of SGCN++ which applies a large-batch approximation. The authors could also provide  an alternative version of SGCN+ without full-batch that only reduces the zeroth-order variance, and evaluate how SGCN+ without full-batch snapshot computation would impact on the zeroth-order and first-order variance.
3. Since the snapshot gap K serve as a budge hyper parameter balances between training speed and quality of variance reduction. As the model converges w.r.t. increasing number of epochs, I would like to know whether we can dynamically increase K during the training process to obtain some speed boost.
4. The paper is well-written in general. However, there is still some typos. For example, in Eq. (1) and Eq. (2), in the computation of gradient G_t^(l), the superscript in D_t should be (l+1) instead of (l), since we require the gradient of the loss w.r.t. the upper layer.

#########################################################################

Questions during rebuttal period:
Please address and clarify the cons above

#########################################################################

---

> ### Author Response · Authors · 2020-11-20
> **Response to Reviewer 3**
>
> We would like to thank the reviewer for the valuable comments and suggestions. We address the typo in the main text and addressed the following questions.
>
> **1. Question: Compare the proposed algorithm with a vanilla Full-batch GCN.**
>
> We run FullGCN using the same configuration on $5$ different datasets and update Table 1. We obtain PPI $82.14\\%$, PPI-large $90.62\\%$, Flickr $52.99\\%$, Reddit $95.15\\%$, Yelp $62.77\\%$. We observe that due to the implicit regularization of stochastic gradient descent [9], the F1-score of VRGCN++ and Exact++ slightly outperform FullGCN on PPI and Reddit datasets. In addition, we add the training loss and validation loss curve for FullGCN to Figure 2. Please see the updated version for the details. As shown in the updated Figure 2, FullGCNs enjoys the best convergence speed since all sampling algorithms are trying to approximate the behavior of FullGCNs.
>
> **2. Question: Evaluate how would SGCN+ without full-batch snapshot computation impact the variance.**
>
> Both SGCN+ and SGCN++ on Reddit and Yelp datasets are using large-batch snapshot approximation, where we select $50\\%$ of the nodes in Reddit and $15\\%$ of all training nodes for Yelp as the large-batch size. In order to further clarify this, we added more discussion in the text of experimental setup (Section 7). In addition, we added further experiments on the effectiveness of large-batch size and snapshot gap on the convergence for SGCN+ in the updated version. Please refer to Figure 7 and 9 in the updated version.
>
> **3. Suggestion: Dynamically increasing $K$ during training.**
>
> Many thanks for pointing this out. Per your suggestion, in Figure 12, we show the comparison of validation loss for fixed snapshot gap $K=10$ and gradually increasing the snapshot gap with $K=10+0.1\times s$. Discussion on the setup can be found on the page 18. Recall that the key bottleneck for SGCN++ is the memory budget and sampling complexity, instead of snapshot computing. Although dynamically increasing the snapshot gap can reduce the number of snapshot steps, we have to admit that increasing $K$ during training cannot significantly reduce the training time, unless an efficient snapshot gap increasing strategy is founded or an advanced hardware system is used for sampling.

---

### Author Response · Authors · 2020-11-20
**References**

### References
1. Jie Chen and Ronny Luss. Stochastic gradient descent with biased but consistent gradient estimators. arXiv preprint arXiv:1807.11880, 2018.
2. Chen, Jianfei, Jun Zhu, and Le Song. Stochastic training of graph convolutional networks with variance reduction. arXiv preprint arXiv:1710.10568 (2017).
3. Tianyi Chen, Yuejiao Sun, and Wotao Yin. Solving stochastic compositional optimization is nearly as easy as solving stochastic optimization. arXiv preprint arXiv:2008.10847, 2020.
4. Lenaic Chizat, Edouard Oyallon, and Francis Bach. On lazy training in differentiable programming. In Advances in Neural Information Processing Systems, pages 2937–2947, 2019.
5. Weilin Cong,  Rana Forsati,  Mahmut Kandemir,  and Mehrdad Mahdavi. Minimal variance sampling with provable guarantees for fast training of graph neural networks. In Proceedings of the 26th ACM SIGKDD International Conference on Knowledge Discovery \& Data Mining, pages 1393–1403, 2020.
6. Vikas K Garg,  Stefanie Jegelka,  and Tommi Jaakkola. Generalization and representational limits of graph neural networks. arXiv preprint arXiv:2002.06157, 2020.
7. SJ Reddi, S Kale, and S Kumar. On the convergence of adam and beyond. arxiv 2019.arXivpreprint arXiv:1904.09237.
8. Mengdi Wang, Ethan X Fang, and Han Liu. Stochastic compositional gradient descent: algorithms for minimizing compositions of expected-value functions. Mathematical Programming, 161(1-2):419–449, 2017.
9. Lei Wu, Zhanxing Zhu, et al. Towards understanding generalization of deep learning: Perspective of loss landscapes. arXiv preprint arXiv:1706.10239, 2017.
10. Shuoguang Yang, Mengdi Wang, and Ethan X Fang. Multilevel stochastic gradient methods for nested composition optimization. SIAM Journal on Optimization, 29(1):616–659, 2019.
11. Junyu Zhang and Lin Xiao. Multi-level composite stochastic optimization via nested variance reduction. arXiv preprint arXiv:1908.11468, 2019.
12. Difan  Zou,  Ziniu  Hu,  Yewen  Wang,  Song  Jiang,  Yizhou  Sun,  and  Quanquan  Gu. Layer-dependent importance sampling for training deep and large graph convolutional networks. In Advances in Neural Information Processing Systems, pages 11249–11259, 2019.

---

### Author Response · Authors · 2020-11-20
**Novelty and contributions**

Thanks for the comment. We note that while at first glance the objective function and algorithmic ideas in this paper look similar to the  variance reduced composite optimization [11], however, it is worth noting that there are significant differences between empirical loss of GCN and  classical composite optimization that necessitate developing novel ideas  to accelerate optimization as  briefly highlighted below (please see **Appendix A** in revised version for a through discussion of differences that distinguish the present work and makes it novel):

- Different objective function which makes the GCN analysis challenging
- Different gradient computation, analysis, and algorithm, which make directly applying multi-level SPIDER nontrivial
- Different theoretical results and novel intuition for sampling-based GCN training.

**Different objective function.** In composite optimization, the output of the lower-level function is treated as the *parameter* of the outer-level function. However in GCN, the output of the lower-level function is used as the *input* of the outer-level function, and the parameter of the outer-level function is independent of the output of the inner-layer result. More specifically, a two-level composite optimization problem can be formulated as
$$
F(\boldsymbol{\theta}) = \frac{1}{N}\sum\_{i=1}^N f\_i\Big(\frac{1}{M}\sum\_{j=1}^M g\_j(\boldsymbol{w}) \Big), \boldsymbol{\theta}=\\\{\boldsymbol{w}\\\}, \qquad\qquad \text{(Eq.1)}
$$
where $f\_i(\cdot)$ is the outer-level function computed on the $i$th data point, $g\_j(\cdot)$ is the inner-level function computed on the $j$th data point, and $\boldsymbol{w}$ is the parameter.
We denote $\nabla f\_i(\cdot)$ and $\nabla g\_j(\cdot)$ as the gradient.
Then, the gradient for Eq.1 is computed as
$$
\nabla F(\boldsymbol{\theta}) = \Big[ \frac{1}{N}\sum\_{i=1}^N \nabla f\_i \Big( \frac{1}{M}\sum\_{j=1}^M g\_j(\boldsymbol{w}) \Big) \Big] \Big( \frac{1}{M}\sum\_{j=1}^M \nabla g\_j(\boldsymbol{w}) \Big),\boldsymbol{\theta}=\\\{\boldsymbol{w}\\\}, \qquad\qquad \text{(Eq.2)}
$$
where the dependency between inner- and outer-level sampling are not considered.
One can independently sample inner layer data to estimate $\widetilde{g}\approx \frac{1}{M}\sum\_{j=1}^M g\_j(\boldsymbol{w})$ and $\nabla \widetilde{g} \approx \frac{1}{M}\sum\_{j=1}^M \nabla g\_j (\boldsymbol{w})$, sample outer layer data to estimate $\nabla \widetilde{f} \approx \frac{1}{N}\sum\_{i=1}^N \nabla f\_i(\widetilde{g})$, then estimate $\nabla F(\boldsymbol{\theta})$ by using $[\nabla \widetilde{f}]^\top \nabla \widetilde{g}$.

By casting the optimization problem in GCN as composite optimization problem in Eq.1, we have
$$
\mathcal{L}(\boldsymbol{\theta}) = \frac{1}{B}\sum\_{i\in\mathcal{V}\_\mathcal{B}} \text{Loss}(\boldsymbol{h}\_i^{(L)}, y_i), \boldsymbol{\theta}=\\\{\mathbf{W}^{(1)}\\\}, \qquad\qquad \text{(Eq.3)}
$$
$$
\mathbf{H}^{(L)} = \sigma(\widetilde{\mathbf{L}}^{(L)} \mathbf{X} \widetilde{\mathbf{W}}^{(L)})
$$
$$\widetilde{\mathbf{W}}^{(L)} = \sigma\Big( \widetilde{\mathbf{L}}^{(L-1)} \mathbf{X}\sigma\Big( \widetilde{\mathbf{L}}^{(L-2)}  \mathbf{X} \ldots \underbrace{\sigma\Big( \widetilde{\mathbf{L}}^{(1)} \mathbf{X} \mathbf{W}^{(1)} \Big)}_{\widetilde{\mathbf{W}}^{(2)}} \ldots \Big)
$$
which is different from the vanilla GCN model. To see this, we note that in vanilla GCNs, since the sampled nodes at the $\ell$th layer are dependent from the nodes sampled at the $(\ell+1)$th layer, we have $\mathbb{E}[\widetilde{\mathbf{L}}^{(\ell)}] = \mathbf{\mathbf{P}}^{(\ell)} \neq \mathbf{L}$.  However, in Eq.3, since the sampled nodes have no dependency on the weight matrices or nodes sampled at other layers, we can easily obtain $\mathbb{E}[\widetilde{\mathbf{L}}^{(\ell)}] = \mathbf{L}$.  These key differences make the analysis more involved and are reflected in all three theorems, that gives us different results.

---

> ### Author Response · Authors · 2020-11-20
> **Novelty and contributions (cont'd)**
>
> **Different gradient computation and algorithm.** The stochastic gradients to update the parameters in Eq.3 are computed as $$
> \frac{\partial \mathcal{L}(\boldsymbol{\theta})}{\partial \widetilde{\mathbf{W}}^{(\ell)}} = \frac{\partial \mathcal{L}(\boldsymbol{\theta})}{\partial \widetilde{\mathbf{W}}^{(L)}} \Big( \prod_{j=\ell+1}^L \frac{\partial \widetilde{\mathbf{W}}^{(j)} }{\partial \widetilde{\mathbf{W}}^{(j-1)}}\Big).
> $$
> However in GCN, there are two types of gradient at each layer (i.e., $\widetilde{\mathbf{D}}^{(\ell)}$ and $\widetilde{\mathbf{G}}^{(\ell)}$) that are fused with each other (i.e., $\widetilde{\mathbf{D}}^{(\ell)}$ is a part of $\widetilde{\mathbf{G}}^{(\ell-1)}$ and $\widetilde{\mathbf{D}}^{(\ell)}$ is a part of $\widetilde{\mathbf{D}}^{(\ell-1)}$) but with different functionality.
> $\widetilde{\mathbf{D}}^{(\ell)}$ is passing gradient between different layers, $\widetilde{\mathbf{G}}^{(\ell)}$ is passing gradient to weight matrices.
>
> These two types of gradient and their coupled relation make both algorithm and analysis different from [11]. For example in multi-level SPIDER, the zeroth-order variance reduction is applied to $\widetilde{\mathbf{W}}\_t^{(\ell)}$ in Eq.~2 (refer to Algorithm 3 in [11]), where $\widetilde{\mathbf{W}}\_{t-1}^{(\ell)}$ is used as a control variant to reduce the variance of $\widetilde{\mathbf{W}}\_{t}^{(\ell)}$, i.e.,
> $$
>     \widetilde{\mathbf{W}}\_t^{(\ell+1)} = \widetilde{\mathbf{W}}\_{t-1}^{(\ell+1)} + \sigma(\widetilde{\mathbf{L}}\_t^{(\ell)} \mathbf{X} \widetilde{\mathbf{W}}\_t^{(\ell)}) - \sigma(\widetilde{\mathbf{L}}\_t^{(\ell)} \mathbf{X} \widetilde{\mathbf{W}}\_{t-1}^{(\ell)})
> $$
> However in *SGCN++*, the zeroth-order variance reduction is applied to $\widetilde{\mathbf{H}}\_t^{(\ell)}$. Because the nodes sampled at the $t$th and $(t-1)$th iterations are unlikely to be the same, we cannot directly use $\mathbf{H}^{(\ell)}\_{t-1}$ to reduce the variance of $\mathbf{H}^{(\ell)}\_{t}$. Instead, the control variant in *SGCN++* is computed by applying historical weight $\mathbf{W}^{(\ell)}\_{t-1}$ on the historical node embedding from previous layer $\mathbf{H}^{(\ell-1)}\_{t-1}$, i.e.,
> $$
>     \widetilde{\mathbf{H}}\_t^{(\ell)} = \widetilde{\mathbf{H}}\_{t-1}^{(\ell)} + \sigma(\widetilde{\mathbf{L}}\_t^{(\ell)} \mathbf{H}^{(\ell-1)}\_{t} \mathbf{W}\_{t}^{(\ell)}) - \sigma(\widetilde{\mathbf{L}}\_t^{(\ell)} \mathbf{H}^{(\ell-1)}\_{t-1} \mathbf{W}\_{t-1}^{(\ell)})
> $$
> These changes are not simply heuristic modifications, but all reflected in the analysis and the result.
>
>
> **Different theoretical results and intuition.** The aforementioned differences further result in a novel analysis of Theorem 1, where we show that the vanilla sampling-based GCNs suffer a residual error $\Delta_\mathbf{b}$ that is not decreasing as the number of iterations $T$ increases. Also, this residual error is strongly related to the difference between sampled and full Laplacian matrices. This is one of our novel observations for GCNs, when compared to (1) multi-level composite optimization with layerwise changing learning rate [10,3], (2) variance reduction based methods [11], and (3) the previous analysis on the convergence of GCNs [1,2].
>
> Our observations can be used as a theoretical motivation on using first-order and doubly variance reduction, and mathematically explain why VRGCN outperform GraphSAGE, even with fewer nodes during training. Furthermore, as the algorithm and gradient computation are different, the theoretical results in Theorem 2 and 3 are also different.

---

### Author Response · Authors · 2020-11-20
**Changes to the original submission**

We appreciate the careful reviews and valuable feedback from all our reviewers. To address the concerns and improve the clarity, we made the following changes to the main body and supplementary materials. Note that we marked all these changes in blue color in the revised version.

 - Added a section to discuss the key differences between this paper and classical multi-level composite optimization and to illustrate why the extension of variance reduction methods for composite optimization to GCNs is not straightforward.

 - Updated the code with FullGCN training, SGD training, and increasing-$K$ training. Besides, a Jupyter notebook is provided to help reproducing the additional experiments.

 - Added more related works per reviewers suggestions and concrete discussion on the connection between each related work to our proposed methods.

---

### Decision · Program_Chairs · 2021-01-07
**Final Decision**

**Decision:**

Reject

**Comment:**

The paper provides variance reduction techniques for GCN training. When training a GCN it is common to sample nodes as in SGD, but also subsample the nodes’ neighbors, due to computational reasons. The entire mechanism introduces both bias and variance to the gradient estimation. The authors decompose the gradient estimate into its variance and bias error, allowing them to apply more targeted variance (and bias) reduction techniques.

The results and improvement over existing GCN methods seem to be solid. The main weakness of the paper is its novelty. As pointed out in the reviews the techniques seem to be quite close to papers [5],[11] (referring to the authors posted list).
It therefore boils down to the question of whether the authors simply applied existing techniques, achieving a better implementation than previous art, or did they develop a truly new algorithm that will encourage further research and deepen the understanding of GCNs. Given the decisive opinions of reviewers 1 and 4, that remained after taking the response into account, I tend to believe that the improvement provided here is either too incremental or not stated in a crisp enough manner in order to be published in its current form